# COPA: Certifying Robust Policies for Offline Reinforcement Learning against Poisoning Attacks

**Fan Wu**[1][*] **Linyi Li**[1][*] **Chejian Xu**[1] **Huan Zhang**[2] **Bhavya Kailkhura**[3]
**Krishnaram Kenthapadi**[4] **Ding Zhao**[2] **Bo Li**[1]
[1]University of Illinois at Urbana-Champaign   [2]Carnegie Mellon University
[3]Lawrence Livermore National Laboratory   [4]Amazon AWS AI
{fanw6,linyi2,chejian2,lbo}@illinois.edu   huan@huan-zhang.com
kailkhura1@llnl.gov   kkenthapadi@gmail.com   dingzhao@andrew.cmu.edu
* Equal contribution.

## Abstract

As reinforcement learning (RL) has achieved near human-level performance in a variety of tasks, its robustness has raised great attention. While a vast body of research has explored test-time (evasion) attacks in RL and corresponding defenses, its robustness against training-time (poisoning) attacks remains largely unanswered. In this work, we focus on *certifying* the robustness of offline RL in the presence of poisoning attacks, where a subset of training trajectories could be *arbitrarily* manipulated. We propose the *first* certification framework, COPA, to certify the number of poisoning trajectories that can be tolerated regarding different certification criteria. Given the complex structure of RL, we propose two certification criteria: *per-state action stability* and *cumulative reward bound*. To further improve the certification, we propose new partition and aggregation protocols to train robust policies. We further prove that some of the proposed certification methods are theoretically tight and some are NP-Complete problems. We leverage COPA to certify three RL environments trained with different algorithms and conclude: (1) The proposed robust aggregation protocols such as *temporal aggregation* can significantly improve the certifications; (2) Our certifications for both per-state action stability and cumulative reward bound are efficient and tight; (3) The certification for different training algorithms and environments are different, implying their intrinsic robustness properties. All experimental results are available at https://copa-leaderboard.github.io.

## 1 Introduction

Reinforcement learning (RL) has been widely applied to a range of applications, including robotics (Kober et al., 2013; Deisenroth et al., 2013; Polydoros & Nalpantidis, 2017) and autonomous vehicles (Shalev-Shwartz et al., 2016; Sallab et al., 2017). In particular, offline RL (Levine et al., 2020) is proposed to leverage previously observed data to train the policy without requiring expensive interaction with environments or online data collection, and enables the reuse of training data (Agarwal et al., 2020). However, offline RL also raises a great safety concern on poisoning attacks (Kiourti et al., 2020; Wang et al., 2020; 2021). A recent survey of industry reports that data poisoning is significantly more concerning than other threats (Kumar et al., 2020). For offline RL, the situation is even worse, and a recent theoretical study shows that the robust offline RL against poisoning attacks is a strictly harder problem than online RL (Zhang et al., 2021).

Although there are several empirical and certified defenses for classification tasks against poisoning attacks (Peri et al., 2020; Levine & Feizi, 2021; Weber et al., 2020), it is challenging and shown ineffective to directly apply them to RL given its complex structure (Kiourti et al., 2020). Thus, robust offline RL against poisoning attacks remains largely unexplored with no mention of robustness certification. In addition, though some theoretical analyses provide general robustness bounds for RL, they either assume a bounded distance between the learned and Bellman optimal policies or are limited to linear MDPs (Zhang et al., 2021). To the best of our knowledge, there is no robust RL

method that is able to provide *practically computable certified robustness* against poisoning attacks. In this paper, we tackle this problem by proposing the first framework of Certifying robust policies for general offline RL against poisoning attacks (**COPA**).

**Certification Criteria.** One critical challenge in certifying robustness for offline RL is the certification criteria, since the prediction consistency is no longer the only goal as in classification. We propose two criteria based on the properties of RL: per-state action stability and cumulative reward bound. The former guarantees that at a specific time, the policy learned with COPA will predict the same action before and after attacks under certain conditions. This is important for guaranteeing the safety of the policy at critical states, *e.g.*, braking when seeing pedestrians. For cumulative reward bound, a lower bound of the cumulative reward for the policy learned with COPA is guaranteed under certain poisoning conditions. This directly guarantees the worst-case overall performance.

**COPA Framework.** COPA is composed of two components: policy partition and aggregation protocol and robustness certification method. We propose three policy partition aggregation protocols: PARL (Per-State Partition Aggregation), TPARL (Temporal Partition Aggregation), and DPARL (Dynamic Temporal Partition Aggregation), and propose certification methods for each of them corresponding to both proposed certification criteria. In addition, for *per-state action stability*, we prove that our certifications for PARL and TPARL are theoretically tight. For *cumulative reward bound*, we propose an adaptive search algorithm, where we compute the possible action set for each state under certain poisoning conditions. Concretely, we propose a novel method to compute the precise action set for PARL and efficient algorithms to compute a superset of the possible action set which leads to sound certification for TPARL and DPARL. We further prove that for PARL our certification is theoretically tight, for TPARL the theoretically tight certification is NP-complete, and for DPARL it is open whether theoretically tight certification exists.

**Technical Contributions.** We take the first step towards certifying the robustness of offline RL against poisoning attacks, and we make contributions on both theoretical and practical fronts.

- We abstract and formulate the robustness certification for offline RL against poisoning attacks, and we propose two certification criteria: per-state action stability and cumulative reward bound.
- We propose the first framework COPA for certifying robustness of offline RL against poisoning attacks. COPA includes novel policy aggregation protocols and certification methods.
- We prove the tightness of the proposed certification methods for the aggregation protocol PARL. We also prove the computational hardness of the certification for TPARL.
- We conduct thorough experimental evaluation for COPA on different RL environments with three offline RL algorithms, demonstrating the effectiveness of COPA, together with several interesting findings.

## 2 RELATED WORK

Poisoning attacks (Nelson et al., 2008; Diakonikolas et al., 2016) are critical threats in machine learning, which are claimed to be more concerning than other threats (Kumar et al., 2020). Poisoning attacks widely exist in classification (Schwarzschild et al., 2021), and both empirical defenses (Liu et al., 2018; Chacon et al., 2019; Peri et al., 2020; Steinhardt et al., 2017) and certified defenses (Weber et al., 2020; Jia et al., 2020; Levine & Feizi, 2021) have been proposed.

After Kiourti et al. (2020) show the existence of effective backdoor poisoning attacks in RL, a recent work theoretically and empirically validates the existence of reward poisoning in online RL (Zhang et al., 2020b). Furthermore, Zhang et al. (2021) theoretically prove that the offline RL is more difficult to be robustified against poisoning than online RL considering linear MDP. From the defense side, Zhang et al. (2021) propose robust variants of the Least-Square Value Iteration algorithm that provides probabilistic robustness guarantees under linear MDP assumption. In addition, Robust RL against reward poisoning is studied in Banihashem et al. (2021), but robust RL against general poisoning is less explored. In this background, we aim to provide the certified robustness for general offline RL algorithms against poisoning attacks, which is the first work that achieves the goal. We discuss broader related work in Appendix I.

## 3 CERTIFICATION CRITERIA OF COPA

In this section, we propose two robustness certification criteria for offline RL against general poisoning attacks: per-state action stability and cumulative reward bound.

**Offline RL.** We model the RL environment by an episodic finite-horizon Markov decision process (MDP) $\mathcal{E} = (\mathcal{S}, \mathcal{A}, R, P, H, d_0)$, where $\mathcal{S}$ is the set of states, $\mathcal{A}$ is the set of discrete actions, $R : \mathcal{S} \times \mathcal{A} \to \mathbb{R}$ is the reward function, $P : \mathcal{S} \times \mathcal{A} \to \mathcal{P}(\mathcal{S})$ is the stochastic transition function with $\mathcal{P}(\cdot)$ defining the set of probability measures, $H$ is the time horizon, and $d_0 \in \mathcal{P}(\mathcal{S})$ is the distribution of the initial state. At time step $t$, the RL agent is at state $s_t \in \mathcal{S}$. After choosing action $a_t \in \mathcal{A}$, the agent transitions to the next state $s_{t+1} \sim P(s_t, a_t)$ and receives reward $r_t = R(s_t, a_t)$. After $H$ time steps, the cumulative reward $J = \sum_{t=0}^{H-1} r_t$. We denote a consecutive sequence of all states between time step $l$ and $r$ as $s_{l:r} := [s_l, s_{l+1}, \ldots, s_r]$.

Here we focus on offline RL, for which the threat of poisoning attacks is practical and more challenging to deal with (Zhang et al., 2021). Concretely, in offline RL, a training dataset $D = \{\tau_i\}_{i=1}^{N}$ consists of logged trajectories, where each trajectory $\tau = \{(s_j, r_j, a_j, s'_j)\}_{j=1}^{l} \in (\mathcal{S} \times \mathcal{A} \times \mathbb{R} \times \mathcal{S})^l$ consists of multiple tuples denoting the transitions (*i.e.*, starting from state $s_j$, taking the action $a_j$, receiving reward $r_j$, and transitioning to the next state $s'_j$).

**Poisoning Attacks.** Training dataset $D$ can be poisoned in the following manner. For each trajectory $\tau \in D$, the adversary is allowed to replace it with an arbitrary trajectory $\widetilde{\tau}$, generating a manipulated dataset $\widetilde{D}$. We denote $D \ominus \widetilde{D} = (D \backslash \widetilde{D}) \bigcup (\widetilde{D} \backslash D)$ as the *symmetric difference* between two datasets $D$ and $\widetilde{D}$. For instance, adding or removing one trajectory causes a symmetric difference of magnitude 1, while replacing one trajectory with a new one leads to a symmetric difference of magnitude 2. We refer to the size of the symmetric difference as the *poisoning size*.

**Certification Goal.** To provide the robustness certification against poisoning attacks introduced above, we aim to certify the *test-time* performance of the trained policy in a clean environment. Specifically, in the *training phase*, the RL training algorithm and our aggregation protocol can be jointly modeled by $\mathcal{M} : \mathcal{D} \to (\mathcal{S}^{\star} \to \mathcal{A})$ which provides an aggregated policy, where $\mathcal{S}^{\star}$ denotes the set of all consecutive state sequences. Our **goal** is to provide robustness certification for the poisoned aggregated policy $\tilde{\pi} = \mathcal{M}(\widetilde{D})$, given bounded poisoning size (*i.e.*, $|D \ominus \widetilde{D}| \le \overline{K}$).

**Robustness Certification Criteria: Per-State Action Stability.** We first aim to certify the robustness of the poisoned policy in terms of the stability of per-state action during test time.

**Definition 1** (Robustness Certification for Per-State Action Stability). Given a clean dataset $D$, we define the robustness certification for *per-state action stability* as that for *any* $\widetilde{D}$ satisfying $|D \ominus \widetilde{D}| \le \overline{K}$, the action predictions of the poisoned and clean policies for the state (or state sequence) $s$ are the same, *i.e.*, $\tilde{\pi} = \mathcal{M}(\widetilde{D}), \pi = \mathcal{M}(D), \tilde{\pi}(s) = \pi(s)$, under the the tolerable poisoning threshold $\overline{K}$. In an episode, we denote the tolerable poisoning threshold for the state at step $t$ by $\overline{K}_t$.

The definition encodes the requirement that, for a particular state, any poisoned policy will always give the same action prediction as the clean one, as long as the poisoning size is within $\overline{K}$ ($\overline{K}$ computed in Section 4). In this definition, $s$ could be either a state or a state sequence, since our aggregated policy (defined in Section 3) may aggregate multiple recent states to make an action choice.

**Robustness Certification Criteria: Lower Bound of Cumulative Reward.** We also aim to certify poisoned policy's overall performance in addition to the prediction at a particular state. Here we measure the overall performance by the cumulative reward $J(\pi)$ (formally defined in Appendix A).

Now we are ready to define the robustness certification for cumulative reward bound.

**Definition 2** (Robustness Certification for Cumulative Reward Bound). Robustness certification for cumulative reward bound is the *lower bound* of *cumulative reward* $\underline{J}_K$ such that $\underline{J}_K \le J(\tilde{\pi})$ for any $\tilde{\pi} = \mathcal{M}(\widetilde{D})$ where $|D \ominus \widetilde{D}| \le K$, *i.e.*, $\tilde{\pi}$ is trained on poisoned dataset $\widetilde{D}$ within poisoning size $K$.

## 4 CERTIFICATION PROCESS OF COPA

In this section, we introduce our framework COPA, which is composed of *training protocols*, *aggregation protocols*, and *certification methods*. The training protocol combined with an offline RL training algorithm provides subpolicies. The aggregation protocol aggregates the subpolicies as an aggregated policy. The certification method certifies the robustness of the aggregated policy against poisoning attacks corresponding to different certification criteria provided in Section 3.

Table 1: Overview of theoretical results in Section 4. "Certification" columns entail our certification theorems. "Analysis" columns entail the analyses of our certification bounds, where "tight" means our certification COPA is theoretically tight, "NP-complete" means the tight certification problem is NP-complete, and "open" means the tight certification problem is still open. Theorems 8 and 13 and proposition 9 are in Appendices C and F.6.2.

| Certification Criteria | Proposed Aggregation Protocol | | | | | |
| --- | --- | --- | --- | --- | --- | --- |
| | PARL ($\pi_P$, Definition 7) | | TPARL ($\pi_T$, Definition 8) | | DPARL ($\pi_D$, Definition 3) | |
| | Certification | Analysis | Certification | Analysis | Certification | Analysis |
| Per-State Action | Theorem 8 | tight (Proposition 9) | Theorem 1 | tight (Proposition 2) | Theorem 3 | open |
| Cumulative Reward | Theorem 4 | tight (Theorem 13) | Theorem 5 | NP-complete (Theorem 6) | Theorem 7 | open |

**Overview of Theoretical Results.** In Table 1, we present an overview of our theoretical results: For each proposed aggregation protocol and certification criteria, we provide the corresponding certification method and core theorems, and we also provide the tightness analysis for each certification.

## 4.1 PARTITION-BASED TRAINING PROTOCOL

COPA's training protocol contains two stages: partitioning and training. We denote $D$ as the entire offline RL training dataset. We abstract an offline RL training algorithm (*e.g.*, DQN) by $\mathcal{M}_0 : 2^D \to \Pi$, where $2^D$ is the power set of $D$, and $\Pi = \{\pi : \mathcal{S} \to \mathcal{A}\}$ is the set of trained policies. Each trained policy in $\Pi$ is a function mapping a given state to the predicted action.

**Partitioning Stage.** In this stage, we separate the training dataset $D$ into $u$ partitions $\{D_i\}_{i=0}^{u-1}$ that satisfy $\bigcup_{i=0}^{u-1} D_i = D$ and $\forall i \neq j, D_i \cap D_j = \emptyset$. Concretely, when performing partitioning, for each trajectory $\tau \in D$, we *deterministically* assign it to one unique partition. The assignment is only dependent on the trajectory $\tau$ itself, and not impacted by any modification to other parts of the training set. One design choice of such a deterministic assignment is using a deterministic hash function $h$ to compute the assignment, *i.e.*, $D_i = \{\tau \in D \mid h(\tau) \equiv i \pmod{u}\}, \forall i \in [u]$.

**Training Stage.** In this stage, for each training data partition $D_i$, we independently apply an RL algorithm $\mathcal{M}_0$ to train a policy $\pi_i = \mathcal{M}_0(D_i)$. Hereinafter, we call these trained polices as *subpolicies* to distinguish from the aggregated policies. Concretely, let $[u] := \{0, 1, \ldots, u-1\}$. For these subpolicies, the *policy indicator* $\mathbf{1}_{i,a} : \mathcal{S} \to \{0, 1\}$ is defined by $\mathbf{1}_{i,a}(s) := \mathbf{1}[\pi_i(s) = a]$, indicating whether subpolicy $\pi_i$ chooses action $a$ at state $s$. The *aggregated action count* $n_a : \mathcal{S} \to \mathbb{N}_{\geq 0}$ is the number of votes across all the subpolicies for action $a$ given state $s$: $n_a(s) := |\{i|\pi_i(s) = a, i \in [u]\}| = \sum_{i=0}^{u-1} \mathbf{1}_{i,a}(s)$. Specifically, we denote $n_a(s_{l:r})$ for $\sum_{j=l}^{r} n_a(s_j)$, *i.e.*, the sum of votes for states between time step $l$ and $r$. A detailed algorithm of the training protocol is in Appendix E.1.

Now we are ready to introduce the proposed aggregation protocols in COPA (PARL, TPARL, DPARL) that generate *aggregated policies* based on subpolicies, and corresponding certification.

## 4.2 AGGREGATION PROTOCOLS: PARL, TPARL, DPARL

With $u$ learned subpolicies $\{\pi_i\}_{i=0}^{u-1}$, we propose three different aggregation protocols in COPA to form three types of aggregated policies for each certification criteria: PARL, TPARL, and DPARL.

**Per-State Partition Aggregation (PARL).** Inspired by aggregation in classification (Levine & Feizi, 2021), PARL aggregates subpolicies by choosing actions with the highest votes. We denote the PARL aggregated policy by $\pi_P : \mathcal{S} \to \mathcal{A}$. When there are multiple highest voting actions, we break ties deterministically by returning the "smaller" ($<$) action, which can be defined by numerical order, lexicographical order, etc. Throughout the paper, we assume $\arg\max$ over $\mathcal{A}$ always uses $<$ operator to break ties. The formal definition of the protocol is in Appendix A.

The intuition behind PARL is that the poisoning attack within size $K$ can change at most $K$ subpolicies. Therefore, as long as the margin between the votes for top and runner-up actions is larger than $2K$ for the given state, after poisoning, we can guarantee that the aggregated PARL policy will not change its action choice. We will formally state the robustness guarantee in Section 4.3.

**Temporal Partition Aggregation (TPARL).** In the sequential decision making process of RL, it is likely that certain important states are much more vulnerable to poisoning attacks, which we refer to as *bottleneck states*. Therefore, the attacker may just change the action predictions for these bottleneck states to deteriorate the overall performance, say, the cumulative reward. For example, in Pong game, we may lose the round when choosing an immediate bad action when the ball is closely approaching the paddle. Thus, to improve the overall certified robustness, we need to focus

on improving the tolerable poisoning threshold for these bottleneck states. Given such intuition and goal, we propose Temporal Partition Aggregation (TPARL) and the aggregated policy is denoted as $\pi_{\mathrm{T}}$, which is formally defined in Definition 8 in Appendix A.

TPARL is based on two insights: (1) Bottleneck states have lower tolerable poisoning threshold, which is because the vote margin between the top and runner-up actions is smaller at such state; (2) Some RL tasks satisfy temporal continuity (Legenstein et al., 2010; Veerapaneni et al., 2020), indicating that good action choices are usually similar across states of adjacent time steps, *i.e.*, adjacent states. Hence, we leverage the subpolicies' votes from adjacent states to enlarge the vote margin, and thus increase the tolerable poisoning threshold. To this end, in TPARL, we predetermine a *window size $W$*, and choose the action with the highest votes across recent $W$ states.

**Dynamic Temporal Partition Aggregation (DPARL).** The TPARL uses a fixed window size $W$ across all states. Since the specification of the window size $W$ requires certain prior knowledge, plus that the same fixed window size $W$ may not be suitable for all states, it is preferable to perform *dynamic temporal aggregation* by using a flexible window size. Therefore, we propose Dynamic Temporal Partition Aggregation (DPARL), which dynamically selects the window size $W$ towards maximizing the tolerable poisoning threshold *per step*. Intuitively, DPARL selects the window size $W$ such that the average vote margin over selected states is maximized. To guarantee that only recent states are chosen, we further constrain the maximum window size ($W_{\max}$).

**Definition 3** (Dynamic Temporal Partition Aggregation). *Given subpolicies $\{\pi_i\}_{i=0}^{u-1}$ and maximum window size $W_{\max}$, at time step $t$, the Dynamic Temporal Partition Aggregation (DPARL) defines an aggregated policy $\pi_{\mathrm{D}} : \mathcal{S}^{\min\{t+1, W_{\max}\}} \to \mathcal{A}$ such that*

$$\pi_{\mathrm{D}}(s_{\max\{t-W_{\max}+1,0\}:t}) := \arg\max_{a \in \mathcal{A}} n_a(s_{t-W'+1:t}), \text{ where } W' = \arg\max_{1 \le W \le \min\{W_{\max}, t+1\}} \Delta_t^W. \quad (1)$$

*In the above equation, $n_a$ is defined in Section 3 and $\Delta_t^W$ is given by*

$$\Delta_t^W := \frac{1}{W} \left( n_{a_1}(s_{t-W+1:t}) - n_{a_2}(s_{t-W+1:t}) \right),$$

$$\text{where} \quad a_1 = \arg\max_{a \in \mathcal{A}} n_a(s_{t-W+1:t}), a_2 = \arg\max_{a \in \mathcal{A}, a \ne a_1} n_a(s_{t-W+1:t}). \quad (2)$$

In the above definition, $\Delta_t^W$ encodes the average vote margin between top action $a_1$ and runner-up action $a_2$ if choosing window size $W$. Thus, $W'$ locates the window size with maximum average vote margin, and its corresponding action is selected. Again, we use the mechanism described in PARL to break ties. Robustness certification methods for DPARL are in Sections 4.3 and 4.4.

In Appendix B, we present a concrete example to demonstrate how different aggregation protocols induce different tolerable poisoning thresholds, and illustrate bottleneck and non-bottleneck states.

### 4.3 CERTIFICATION OF PER-STATE ACTION STABILITY

In this section, we present our robustness certification theorems and methods for *per-state action*. For each of the aggregation protocols (PARL, TPARL, and DPARL), at each time step $t$, we will compute a valid *tolerable poisoning threshold $\overline{K}$* as defined in Definition 1, such that the chosen action at step $t$ does not change as long as the poisoning size $K \le \overline{K}$.

**Certification for PARL.** Due to the space limit, we defer the robustness certification method for PARL to Appendix C. The certification method is based on Theorem 8. We further show the theoretical tightness of the certification in Proposition 9. All the theorem statements are in Appendix C.

**Certification for TPARL.** We certify the robustness of TPARL following Theorem 1.

**Theorem 1.** *Let $D$ be the clean training dataset; let $\pi_i = \mathcal{M}_0(D_i), 0 \le i \le u-1$ be the learned subpolicies according to Section 4.1 from which we define $n_a$ (Section 3); and let $\pi_{\mathrm{T}}$ be the Temporal Partition Aggregation policy: $\pi_{\mathrm{T}} = \mathcal{M}(D)$ where $\mathcal{M}$ abstracts the whole training-aggregation process. $\widetilde{D}$ is a poisoned dataset and $\widetilde{\pi_{\mathrm{T}}}$ is the poisoned policy: $\widetilde{\pi_{\mathrm{T}}} = \mathcal{M}(\widetilde{D})$.*

*For a given state $s_t$ encountered at time step $t$ during test time, let $a := \pi_{\mathrm{T}}(s_{\max\{t-W+1,0\}:t})$, then at time step $t$ the tolerable poisoning threshold (see Definition 1)*

$$\overline{K}_t = \min_{a' \ne a, a' \in \mathcal{A}} \max \left\{ p \,\Big|\, \sum_{i=1}^{p} h_{a,a'}^{(i)} \le \delta_{a,a'} \right\} \quad (3)$$

*where* $\{h_{a,a'}^{(i)}\}_{i=1}^{u}$ *is a nonincreasing permutation of*

$$\left\{ \sum_{j=0}^{\min\{W-1,t\}} \mathbf{1}_{i,a}(s_{t-j}) + \min\{W, t+1\} - \sum_{j=0}^{\min\{W-1,t\}} \mathbf{1}_{i,a'}(s_{t-j}) \right\}_{i=0}^{u-1} =: \{h_{i,a,a'}\}_{i=0}^{u-1}, \quad (4)$$

*and* $\delta_{a,a'} := n_a(s_{\max\{t-W+1,0\}:t}) - (n_{a'}(s_{\max\{t-W+1,0\}:t}) + \mathbf{1}[a' < a])$. *Here,* $\mathbf{1}_{i,a}(s) = \mathbf{1}[\pi_i(s) = a]$ *(Section 3), and* $W$ *is the window size.*

*Remark.* We defer the detailed proof to Appendix F.2. The theorem provides a per-state action certification for TPARL. The detailed algorithm is in Algorithm 3 (Appendix E.2). The certification time complexity per state is $O(|\mathcal{A}|u(W+\log u))$ and can be further optimized to $O(|\mathcal{A}|u \log u)$ with proper prefix sum caching across time steps. We prove the certification for TPARL is theoretically tight in Proposition 2 (proof in Appendix F.3). We also prove that directly extending Theorem 8 for TPARL (Corollary 10) is loose in Appendix F.4.

**Proposition 2.** *Under the same condition as Theorem 1, for any time step* $t$*, there exists an RL learning algorithm* $\mathcal{M}_0$*, and a poisoned dataset* $\widetilde{D}$*, such that* $|D \ominus \widetilde{D}| = \overline{K}_t + 1$*, and* $\widetilde{\pi}_{\mathrm{T}}(s_{\max\{t-W+1,0\}:t}) \neq \pi_{\mathrm{T}}(s_{\max\{t-W+1,0\}:t})$.

**Certification for DPARL.** Theorem 3 provides certification for DPARL.

**Theorem 3.** *Let* $D$ *be the clean training dataset; let* $\pi_i = \mathcal{M}_0(D_i), 0 \leq i \leq u - 1$ *be the learned subpolicies according to Section 4.1 from which we define* $n_a$ *(see Section 3); and let* $\pi_{\mathrm{D}}$ *be the Dynamic Temporal Partition Aggregation:* $\pi_{\mathrm{D}} = \mathcal{M}(D)$ *where* $\mathcal{M}$ *abstracts the whole training-aggregation process.* $\widetilde{D}$ *is a poisoned dataset and* $\widetilde{\pi}_{\mathrm{D}}$ *is the poisoned policy:* $\widetilde{\pi}_{\mathrm{D}} = \mathcal{M}(\widetilde{D})$.

*For a given state* $s_t$ *encountered at time step* $t$ *during test time, let* $a := \pi_{\mathrm{D}}(s_{\max\{t-W_{\max}+1,0\}:t})$ *and* $W'$ *be the chosen time window (according to Equation* (1)*), then tolerable poisoning threshold*

$$\overline{K}_t^D = \min \left\{ \overline{K}_t, \min_{1 \leq W^* \leq \min\{W_{\max}, t+1\}, W^* \neq W', a' \neq a, a'' \neq a} L_{a',a''}^{W^*,W'} \right\} \quad (5)$$

*where* $\overline{K}_t$ *is defined by Equation* (3) *with* $W$ *as* $W'$ *and* $L_{a,a''}^{W^*,W'}$ *defined by the below Definition 4.*

**Definition 4** ($L$ in Theorem 3)**.** Under the same condition as Theorem 3, for given $W^*, W', a, a', a''$, we let $a^{\#} := \arg\max_{a_0 \neq a', a_0 \in \mathcal{A}} n_{a_0}(s_{t-W^*+1:t})$, then

$$L_{a',a''}^{W^*,W'} := \max \left\{ p \,\Big|\, \sum_{i=1}^{p} g^{(i)} + W'(n_{a'}^{W^*} - n_{a^{\#}}^{W^*}) - W^*(n_a^{W'} - n_{a''}^{W'}) - \mathbf{1}[a' > a] < 0 \right\} \quad (6)$$

where $n_a^w$ is a shorthand of $n_a(s_{t-w+1:t})$ and $\{g^{(i)}\}_{i=1}^{u}$ is a nonincreasing permutation of $\{g_i\}_{i=0}^{u-1}$. Each $g_i$ is defined by

$$g_i := \sum_{w=0}^{\max\{W^*,W'\}} \max_{a_0 \in \mathcal{A}} \sigma^w(a_0) - \sigma^w(\pi_i(s_{t-w})), \text{ where} \quad (7)$$

$$\sigma^w(a_0) := W'\mathbf{1}[a_0 = a', w \leq W^*] - W'\mathbf{1}[a_0 = a^{\#}, w \leq W^*] - W^*\mathbf{1}[a_0 = a, w \leq W'] + W^*\mathbf{1}[a_0 = a'', w \leq W'].$$

*Proof sketch.* A successful poisoning attack should change the chosen action from $a$ to an another $a'$. If after attack, the chosen window size is still $W'$, the poisoning size should be at least larger than $\overline{K}_t$ according to Theorem 1. If the chosen window size is not $W'$, we find out that the poisoning size is at least $\min_{a'' \neq a} L_{a',a''}^{W^*,W'} + 1$ from a greedy-based analysis. Formal proof is in Appendix F.5. $\square$

*Remark.* The theorem provides a valid per-state action certification for DPARL policy. The detailed algorithm is in Algorithm 4 (Appendix E.2). The certification time complexity per state is $O(W_{\max}^2|\mathcal{A}|^2u + W_{\max}|\mathcal{A}|^2u \log u)$, which in practice adds similar overhead compared with TPARL certification (see Appendix H.3). Unlike certification for PARL and TPARL, the certification given by Theorem 3 is not theoretically tight. An interesting future work would be providing a tighter per-state action certification for DPARL.

## 4.4 CERTIFICATION OF CUMULATIVE REWARD BOUND

In this section, we present our robustness certification for *cumulative reward bound*. We assume the deterministic RL environment throughout the cumulative reward certification for convenience, *i.e.*, the transition function is $P : \mathcal{S} \times \mathcal{A} \to \mathcal{S}$ and the initial state is a fixed $s_0 \in \mathcal{S}$. The certification goal, as listed in Definition 6, is to obtain a lower bound of cumulative reward under poisoning attacks, given bounded poisoning size $K$. The cumulative reward certification is based on a novel adaptive search algorithm COPA-SEARCH inspired from Wu et al. (2022); we tailor the algorithm to certify against poisoning attacks. We defer detailed discussions and complexity analysis to Appendix E.3.

**COPA-SEARCH Algorithm Description.** The pseudocode is in Algorithm 5 (Appendix E.3). The method starts from the base case: when the poisoning threshold $K_{\text{cur}} = 0$, the lower bound of cumulative reward $\underline{J}_{K_{\text{cur}}}$ is exactly the reward without poisoning. The method then gradually increases the poisoning threshold $K_{\text{cur}}$, by finding the immediate larger $K' > K_{\text{cur}}$ that can expand the possible action set along the trajectory. With the increase of $K_{\text{cur}} \leftarrow K'$, the attack may cause the poisoned policy $\tilde{\pi}$ to take different actions at some states, thus resulting in new trajectories. We need to figure out *a set of all possible actions* to exhaustively traverse all possible trajectories. We will introduce theorems to compute this set of possible actions. With this set, the method effectively explores these new trajectories by formulating them as expanded branches of a trajectory tree. Once all new trajectories are explored, the method examines all leaf nodes of the tree and figures out the minimum reward among them, which is the new lower bound of cumulative reward $\underline{J}_{K'}$ under new poisoning size $K'$. We then repeat this process of increasing poisoning size from $K'$ and expanding with new trajectories until we reach a predefined threshold for poisoning size $K$.

**Definition 5** (Possible Action Set). Given previous states $s_{0:t}$, the subpolicies $\{\pi_i\}_{i=0}^{u-1}$, the aggregation protocol (PARL, TPARL, or DPARL), and the poisoning size $K$, the *possible action set* $A$ at step $t$ is a subset of action space: $A \subseteq \mathcal{A}$, such that for any poisoned policy $\tilde{\pi}$, as long as the poisoning size is within $K$, the chosen action at step $t$ will always be in $A$, *i.e.*, $a_t = \tilde{\pi}(s_{0:t}) \in A$.

**Possible Action Set for PARL.** The following theorem gives the possible action set for PARL.

**Theorem 4** (Tight PARL Action Set). *Under the condition of Definition 5, suppose the aggregation protocol is PARL as defined in Definition 7, then the* possible action set *at step $t$*

$$A^T(K) = \left\{ a \in \mathcal{A} \,\middle|\, \sum_{a' \in \mathcal{A}} \max\{n_{a'}(s_t) - n_a(s_t) - K + \mathbf{1}[a' < a], 0\} \leq K \right\}. \tag{8}$$

We defer the proof to Appendix F.6.1. Furthermore, in Appendix F.6.2 we show that: 1) The theorem gives theoretically tight possible action set; 2) In contrast, directly extending PARL's per-state certification gives loose certification.

**Possible Action Set for TPARL.** The following theorem gives the possible action set for TPARL.

**Theorem 5.** *Under the condition of Definition 5, suppose the aggregation protocol is TPARL as defined in Definition 8, then the* possible action set *at step $t$*

$$A(K) = \left\{ a \in \mathcal{A} \,\middle|\, \sum_{i=1}^{K} h_{a',a}^{(i)} > \delta_{a',a}, \forall a' \neq a \right\}, \tag{9}$$

*where $h_{a',a}^{(i)}$ and $\delta_{a',a}$ follow the definition in Theorem 1.*

We defer the proof to Appendix F.7. The possible action set here is no longer theoretically tight. Indeed, the problem of computing a possible action set with minimum cardinality for TPARL is NP-complete as we shown in the following theorem (proved in Appendix F.8), where we reduce computing theoretically tight possible action set to the set cover problem (Karp, 1972). This result can be viewed as the hardness of targeted attack. In other words, the optimal *untargeted* attack on TPARL can be found in polynomial time, while the optimal *targeted* attack on TPARL is NP-complete, which indicates the robustness property of proposed TPARL.

**Theorem 6.** *Under the condition of Definition 5, suppose we use TPARL (Definition 8) as the aggregation protocol, then computing a possible action set $A(K)$ such that any possible action set $S$ satisfies $|A(K)| \leq |S|$ is* NP*-complete.*

**Possible Action Set for DPARL.** The following theorem gives the possible action set for DPARL.

**Theorem 7.** *Under the condition of Definition 5, suppose the aggregation protocol is TPARL as defined in Definition 3, then the* possible action set *at step $t$*

$$A(K) = \{a_t\} \cup \left\{ a' \in \mathcal{A} \,\middle|\, \min_{\substack{1 \leq W^* \leq \min\{W_{\max}, t+1\}, \\ W^* \neq W', a'' \neq a_t}} L_{a',a''}^{W^*, W'} \leq K \right\} \cup \left\{ a \in \mathcal{A} \,\middle|\, \sum_{i=1}^{K} h_{a',a}^{(i)} > \delta_{a',a}, \forall a' \neq a \right\} \tag{10}$$

*where $a_t = \pi_{\mathrm{D}}(s_{\max\{t - W_{\max}+1, 0\}:t})$ is the clean policy's chosen action, $W'$ is defined by Equation (1), $L_{a',a''}^{W^*, W'}$ is defined by Definition 4 with $a$ being replaced by $a_t$, and $h_{a',a}^{(i)}, \delta_{a',a}$ is defined in Theorem 1 with $W$ replaced by $W'$.*

We prove the theorem in Appendix F.8. As summarized in Table 1, we further prove the tightness or hardness of certification for PARL and TPARL, while for DPARL it is an interesting open problem on whether theoretical tight certification is possible in polynomial time.

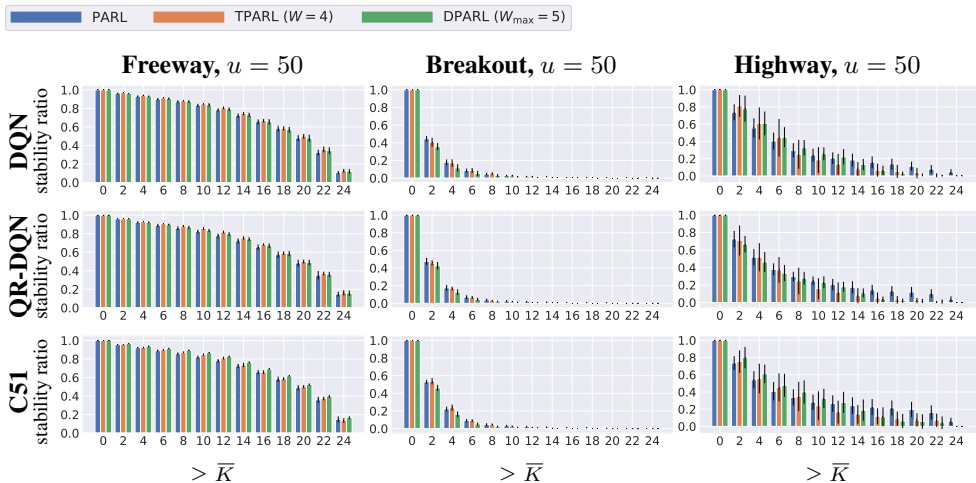

Figure 1: **Robustness certification for per-state action stability.** We plot the cumulative histogram of the *tolerable poisoning size* $\overline{K}$ for all time steps. We provide the certification for different aggregation protocols (PARL, TPARL, DPARL) on three environments and #partitions $u = 50$. The results are averaged over 20 runs with the vertical bar on top denoting the standard deviation.

## 5 EXPERIMENTS

In this section, we present the evaluation for our COPA framework, specifically, the aggregation protocols (Section 4.2) and the certification methods under different certification criteria (Sections 4.3 and 4.4). We defer the description of the offline RL algorithms (DQN (Mnih et al., 2013), QR-DQN (Dabney et al., 2018), and C51 (Bellemare et al., 2017)) used for training the subpolicies to Appendix G.1, and the concrete experimental procedures to Appendix G.2. As a summary, we obtain similar conclusions from per-state action certification and reward certification: 1) QR-DQN and C51 are oftentimes more certifiably robust than DQN; 2) temporal aggregation (TPARL and DPARL) achieves higher certification for environments satisfying temporal continuity, *e.g.*, Freeway; 3) larger partition number improves the certified robustness; 4) Freeway is the most stable and robust environment among the three. More interesting discussions are deferred to Appendix H.

### 5.1 EVALUATION OF ROBUSTNESS CERTIFICATION FOR PER-STATE ACTION STABILITY

We provide the robustness certification for per-state action stability based on Section 4.3.

**Experimental Setup and Metrics.** We evaluate the aggregated policies $\pi_{\mathrm{P}}$, $\pi_{\mathrm{T}}$, and $\pi_{\mathrm{D}}$ following Section 4.2. Basically, in each run, we run one trajectory (of maximum length $H$) using the derived policy, and compute $\overline{K}_t$ at each time step $t$. Given $\{\overline{K}_t\}_{t=0}^{H-1}$, we obtain a cumulative histogram— for each threshold $\overline{K}$, we count the time steps that achieve a threshold no smaller than it and then normalize, *i.e.*, $\sum_{t=0}^{H-1} \mathbf{1}[\overline{K}_t \geq \overline{K}]/H$. We call this quantity *stability ratio* since it reflects the per-state action stability w.r.t. given poisoning thresholds. We also compute an *average tolerable poisoning thresholds* for a trajectory, defined as $\sum_{t=0}^{H-1} \overline{K}_t/H$. More details are deferred to Appendix G.2.

**Evaluation Results.** We present the comparison of per-state action certification for different RL methods and certification methods in Figure 1. We plot partial poisoning thresholds on the $x$-axes here, and omit full results in Appendix H.6, where we also report the average tolerable poisoning thresholds. We additionally report benign empirical reward and the comparisons with standard training in Appendices H.1 and H.2, as well as more analytical statistics in Appendices H.3 and H.4.

The cumulative histograms in Figure 1 can be compared in different levels. Basically, we compare the *stability ratio* at each tolerable poisoning thresholds $\overline{K}$—*higher ratio at larger poisoning size indicates stronger certified robustness*. On the **RL algorithm** level, QR-DQN and C51 consistently outperform the baseline DQN, and C51 has a substantial advantage particularly in Highway. On the **aggregation protocol** level, we observe different behaviors in different environments. On Freeway, methods with temporal aggregation (TPARL and DPARL) achieve higher robustness, and DPARL achieves the highest certified robustness in most cases; while on Breakout and Highway, the single-step aggregation PARL is oftentimes better. This difference is due to the different properties of environments. Our temporal aggregation is developed based on the assumption of consistent action

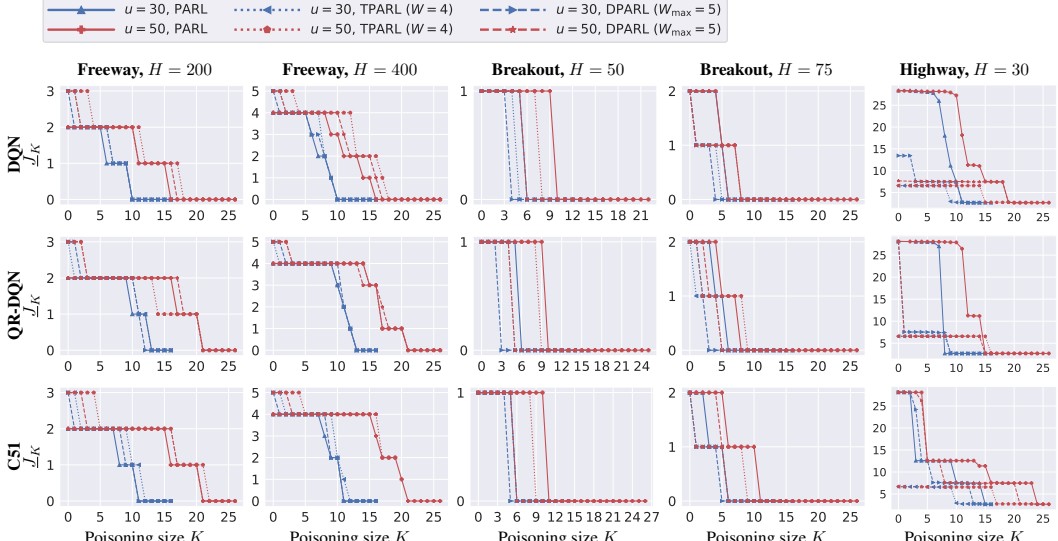

Figure 2: **Robustness certification for cumulative reward.** We plot the *lower bound of cumulative reward bound* $\underline{J}_K$ w.r.t. *poisoning size $K$* under three aggregation protocols (PARL, TPARL ($W = 4$), DPARL ($W_{\max} = 5$)) with two #partitions $u$, evaluated on three environments with different horizon lengths $H$.

selection in adjacent time steps. This assumption is true in Freeway while violated in Breakout and Highway. A more detailed explanation of environment properties is omitted to Appendix H.5. On the **partition number** level, a larger partition number generally allows larger tolerable poisoning thresholds as shown in Appendix H.6. Finally, on the **RL environment** level, Freeway achieves much higher certified robustness for per-state action stability than Highway, followed by Breakout, implying that Freeway is an environment that accommodates more stable and robust policies.

## 5.2 EVALUATION OF ROBUSTNESS CERTIFICATION FOR CUMULATIVE REWARD BOUND

We provide the robustness certification for cumulative reward bound according to Section 4.4.

**Experimental Setup and Metrics.** We evaluate the aggregated policies $\pi_P$, $\pi_T$, and $\pi_D$ following Theorem 4, Theorem 5 and Theorem 7. We compute the lower bounds of the cumulative reward $\underline{J}_K$ w.r.t. the poisoning size $K$ using the COPA-SEARCH algorithm introduced in Section 4.4. We provide details of the evaluated trajectory length along with the rationales in Appendix G.2.

**Evaluation Results.** We present the comparison of reward certification for different RL algorithms and certification methods in Figure 2. Essentially, at each poisoning size $K$, we compare the lower bound of cumulative reward achieved by different RL algorithms and certification methods—*higher value of the lower bound implies stronger certified robustness*. On the **RL algorithm** level, QR-DQN and C51 almost invariably outperform the baseline DQN algorithm. On the **aggregation protocol** level, methods with temporal aggregation consistently surpass the single-step aggregation PARL on Freeway but not the other two, as analyzed in Section 5.1. In addition, we note that DPARL is sometimes not as robust as TPARL. We hypothesize two reasons: 1) the dynamic mechanism is more susceptible to the attack, *e.g.*, the selected optimal window size is prone to be manipulated; 2) the lower bound is looser for DPARL given the difficulty of computing the possible action set in DPARL (discussed in Theorem 7). On the **partition number** level, a larger partition number ($u = 50$) demonstrates higher robustness. On the **horizon length** level, the robustness ranking of different policies is similar under different horizon lengths with slight differences, corresponding to the property of finite-horizon RL. On the **RL environment** level, Freeway can tolerate a larger poisoning size than Breakout and Highway. More results and discussions are in Appendix H.7.

## 6 CONCLUSIONS

In this paper, we proposed **COPA**, the first framework for certifying robust policies for offline RL against poisoning attacks. COPA includes three policy aggregation protocols. For each aggregation protocol, COPA provides a sound certification for both per-state action stability and cumulative reward bound. Experimental evaluations on different environments and different offline RL training algorithms show the effectiveness of our robustness certification in a wide range of scenarios.

ACKNOWLEDGMENTS

This work is partially supported by the NSF grant No.1910100, NSF CNS 20-46726 CAR, Alfred P. Sloan Fellowship, the U.S. Department of Energy by the Lawrence Livermore National Laboratory under Contract No. DE-AC52-07NA27344 and LLNL LDRD Program Project No. 20-ER-014, and Amazon Research Award.

**Ethics Statement.** In this paper, we prove the first framework for certifying robust policies for offline RL against poisoning attacks. On the one hand, such framework provides rigorous guarantees in practice RL tasks and thus significantly alleviates the security vulnerabilities of offline RL algorithms against training-phase poisoning attacks. The evaluation of different RL algorithms also provides a better understanding about the different degrees of security vulnerabilities across different RL algorithms. On the other hand, the robustness guarantee provided by our framework only holds under specific conditions of the attack. Specifically, we require the attack to change only a bounded number of training trajectories. Therefore, users should be aware of such limitations of our framework, and should not blindly trust the robustness guarantee when the attack can change a large number of training instances or modify the training algorithm itself. As a result, we encourage researchers to understand the potential risks, and evaluate whether our constraints on the attack align with their usage scenarios when applying our COPA to real-world applications. We do not expect any ethics issues raised by our work.

**Reproducibility Statement.** All theorem statements are substantiated with rigorous proofs in Appendix F. In Appendix E, we list the pseudocode for all key algorithms. Our experimental evaluation is conducted with publicly available OpenAI Gym toolkit (Brockman et al., 2016). We introduce all experimental details in both Section 5 and Appendix G. Specifically, we build the code upon the open-source code base of Agarwal et al. (2020), and we upload the source code at `https://github.com/AI-secure/COPA` for reproducibility purpose.

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

## A  OMITTED DEFINITIONS

### A.1  CUMULATIVE REWARD

**Definition 6** (Cumulative Reward). Let $P : \mathcal{S} \times \mathcal{A} \to \mathcal{P}(\mathcal{S})$ be the transition function with $\mathcal{P}(\cdot)$ defining the set of probability measures. Let $R, d_0, H$ be the reward function, initial state distribution, and time horizon. We denote $J(\pi)$ as the *cumulative reward* of the given policy $\pi$:

$$J(\pi) := \mathbb{E} \sum_{t=0}^{H-1} R(s_t, a_t), \text{ where } a_t = \pi(s_t), s_{t+1} \sim P(s_t, a_t), s_0 \sim d_0, \tag{11}$$

It is natural to adapt this definition when there is a discount factor. If policy $\pi$ considers recent $n_t$ states instead of only $s_t$ to make action predictions, we can change $a_t = \pi(s_t)$ to $a_t = \pi(s_{t-n_t+1:t})$ in Equation (11).

### A.2  PARL (PER-STATE PARTITION AGGREGATION) PROTOCOL

**Definition 7** (Per-State Partition Aggregation). Given subpolicies $\{\pi_i\}_{i=0}^{u-1}$, the Per-State Partition Aggregation (PARL) protocol defines an aggregated policy $\pi_P : \mathcal{S} \to \mathcal{A}$ such that

$$\pi_P(s) := \underset{a \in \mathcal{A}}{\arg \max}\, n_a(s), \tag{12}$$

where $n_a(s)$ is defined in Section 3.

As mentioned in Section 4, when the $\arg \max$ in Equation (12) returns multiple elements, we break ties deterministically by returning the "smaller" ($<$) action, which can be defined by numerical order, lexicographical order, etc.

### A.3  TPARL (TEMPORAL PARTITION AGGREGATION) PROTOCOL

**Definition 8** (Temporal Partition Aggregation). Given subpolicies $\{\pi_i\}_{i=0}^{u-1}$ and window size $W$, at time step $t$, the Temporal Partition Aggregation (TPARL) defines an aggregated policy $\pi_T : \mathcal{S}^{\min\{t+1,W\}} \to \mathcal{A}$ such that

$$\pi_T(s_{\max\{t-W+1,0\}:t}) = \underset{a \in \mathcal{A}}{\arg \max}\, n_a(s_{\max\{t-W+1,0\}:t}), \tag{13}$$

where $s_{l:r}$ and $n_a$ are defined in Section 3.

In the above definition, at time step $t$, the input of policy $\pi_T$ contains all states between time step $\max\{t - W + 1, 0\}$ and $t$ within window size $W$; and the output is the policy prediction with the highest votes across these states. Recall that in Section 3, we define $n_a(s_{l:r}) = \sum_{j=l}^{r} n_a(s_j)$. We break ties in the same way as in PARL.

## B  ILLUSTRATION OF COPA AGGREGATION PROTOCOLS

We present a concrete example to demonstrate how different aggregation protocols induce different tolerable poisoning thresholds, illustrate bottleneck and non-bottleneck states, and provide additional discussions.

Suppose the action space contains two actions $\mathcal{A} = \{a_1, a_2\}$, and there are six subpolicies $\{\pi_i\}_{i=0}^{u-1}$ where $u = 6$. We are at time step $t = 7$ now. When there is no poisoning, the subpolicies' action predictions for state $s_0$ to $s_7$ are shown by Table 2. After aggregation, all aggregated policies will choose action $a_1$ at $t = 7$.

**PARL (Definition 7).**    The PARL aggregation protocol only uses the current state $s_7$ to aggregate the votes. On $s_7$, three subpolicies choose action $a_1$ and three others choose action $a_2$. By breaking the tie with $a_1 < a_2$, the PARL policy $\pi_P(s_7) = a_1$. The tolerable poisoning threshold $\overline{K}_t = 0$, because one action flipping from $a_1$ to $a_2$ by poisoning only one subpolicy can change the aggregated policy to $a_2$.

**TPARL (Definition 8).**    The TPARL aggregation protocol uses window size $W = 7$. This implies that, we aggregate all states $s_{1:7}$ to count the votes and decide the action. Since $n_{a_1}(s_{1:7}) = 36$ and

Table 2: A concrete example of action predictions, where "1" means action $a_1$ and "2" means action $a_2$. When there is no poisoning attack, the corresponding time window spans by PARL, TPARL, and DPARL are shown by green , blue , and pink respectively. All aggregated policies choose action $a_1$, but have different tolerable poisoning thresholds as shown in the last column.

| Action Prediction for | $s_0$ | $s_1$ | $s_2$ | $s_3$ | $s_4$ | $s_5$ | $s_6$ | $s_7$ | $\overline{K}$ at $s_7$ |
|---|---|---|---|---|---|---|---|---|---|
| $\pi_0$ | 1 | 1 | 1 | 1 | 1 | 1 | 1 | 1 | |
| $\pi_1$ | 1 | 1 | 1 | 1 | 1 | 1 | 1 | 1 | |
| $\pi_2$ | 1 | 1 | 1 | 1 | 1 | 1 | 1 | 1 | |
| $\pi_3$ | 1 | 1 | 1 | 1 | 1 | 1 | 1 | 2 | |
| $\pi_4$ | 1 | 1 | 1 | 1 | 1 | 1 | 1 | 2 | |
| $\pi_5$ | 1 | 1 | 1 | 2 | 1 | 2 | 2 | 2 | |
| PARL ($\pi_\mathrm{P}$, Definition 7) | | | | | | | | 1 | 0 |
| TPARL with $W = 7$ ($\pi_\mathrm{T}$, Definition 8) | | | | | | | | 1 | 2 |
| DPARL with $W_{\max} = 8$ ($\pi_\mathrm{D}$, Definition 3) | | | | | | | | 1 | 1 |

$n_{a_2}(s_{1:7}) = 6$, after poisoning two subpolicies, we still $\widetilde{n}_{a_1}(s_{1:7}) \geq \widetilde{n}_{a_2}(s_{1:7})$. Thus, we achieve an tolerable poisoning threshold $\overline{K}_t = 2$.

Compared with PARL, the temporal aggregation in TPARL increases the vote margin between $a_1$ and $a_2$ and thus improve the tolerable poisoning threshold at current state.

**DPARL (Definition 3).** The DPARL aggregation protocol chooses the window size $W' = 8$ since this window size gives the largest average vote margin $\Delta_t^{W'}$ (defined in Equation (1)). We can easily find out that with poisoning size $K = 1$, we will still choose $\widetilde{W'} = 8$ as the window size and the resulting votes for $a_1$ can be kept higher than votes for $a_2$. However, when it comes to $K = 2$, if we totally flip subpolicies $\pi_0$ and $\pi_1$ to let them always predict action $a_2$, then the DPARL aggregated policy will turn to choose $W' = 1$ as the window size and then choose action $a_2$ as the action output. Thus, we achieve a tolerable poisoning threshold $\overline{K}_t = 1$ this time.

**Illustrations of Bottleneck and Non-Bottleneck States.** According to our illustration in Section 4.2, bottleneck states are those where subpolicies vote for diverse actions but the best action is the same as previous states, and non-bottleneck states are those where subpolicies mostly vote for the same action. As we can see, $s_0$ to $s_6$ are non-bottleneck states since the subpolicies almost unanimously vote for one action. In contrast, $s_7$ is a bottleneck state.

From the perspectives of different *aggregation protocols*, we observe that for the bottleneck state $s_7$, both TPARL and DPARL exploit temporal aggregation to boost the certified robust poisoning threshold (from 0 to 2 and from 0 to 1 respectively), thus demonstrating the effectiveness of TPARL and DPARL on improving certified robustness. On the other hand, for non-bottleneck states $s_0$ to $s_6$, we can easily see that different aggregation protocols do not differ much in terms of provided certified robustness levels.

**Explanations for Bottleneck and Non-Bottleneck States.** We provide additional discussions regarding *why bottleneck states are vulnerable* and *why our temporal aggregation strategy can effectively alleviate the problem.* We first explain the existence of the bottleneck states. Given the property of the bottleneck states that there is high disagreement among different subpolicies on such states, they may lie close to the decision boundary. This may be a result of poisoning, or simply due to the intrinsic difficulty of the state. On the other hand, such states naturally exist in the rollouts (like natural adversarial examples (Hendrycks et al., 2021)), and may induce high instability of the rollouts during testing. We next discuss additional intuitions for our temporal aggregation. Essentially, temporal aggregation effectively leverages the adjacent non-bottleneck states to rectify the decisions at the bottleneck states, based on the assumption of temporal continuity which has been revealed and utilized in several previous works (Legenstein et al., 2010; Veerapaneni et al., 2020; Ye et al., 2021).

## C CERTIFICATION OF PER-STATE ACTION STABILITY FOR PARL

We certify the robustness of PARL following Theorem 8.

**Theorem 8.** *Let $D$ be the clean training dataset; let $\pi_i = \mathcal{M}_0(D_i), 0 \leq i \leq u - 1$ be the learned subpolicies according to Section 4.1 from which we define $n_a$ (see Section 3); and let $\pi_P$ be the Per-State Partition Aggregation policy: $\pi_P = \mathcal{M}(D)$ where $\mathcal{M}$ abstracts the whole training-aggregation process. $\widetilde{D}$ is a poisoned dataset and $\widetilde{\pi_P}$ is the poisoned policy: $\widetilde{\pi_P} = \mathcal{M}(\widetilde{D})$.*

*For a given state $s_t$ encountered at time step $t$ during test time, let $a := \pi_P(s_t)$, then*

$$\overline{K}_t = \left\lfloor \frac{n_a(s_t) - \max_{a' \neq a} (n_{a'}(s_t) + \mathbf{1}[a' < a])}{2} \right\rfloor. \tag{14}$$

*Remark.* The theorem provides a valid per-state action certification for PARL, since according to Definition 1, as long as the poisoning size is smaller or equal to $\overline{K}_t$, *i.e.*, $|D \ominus \widetilde{D}| \leq \overline{K}_t$, the action of the poisoned policy is the same as the clean policy: $\widetilde{\pi_P}(s_t) = \pi_P(s_t) = a$. To compute $\overline{K}_t$, according to Equation (14), we rely on the aggregated action count $n_a$ for state $s_t$ from subpolicies. The $a' < a$ is the "smaller-than" operator introduced following Definition 7.

*Proof of Theorem 8.* Suppose the poisoning size is within $K$, then after poisoning, the aggregated action count $\widetilde{n}_a(s_t) \in [n_a(s_t) - K, n_a(s_t) + K]$ where $n_a(s_t)$ is such count before poisoning. Thus, to ensure the chosen action does not change, a necessary condition is that $\widetilde{n}_a(s_t) \geq \widetilde{n}_{a'}(s_t) + \mathbf{1}[a' < a]$ for any other $a' < a$. Solving $K$ yields Equation (14). $\qquad\square$

We use the theorem to compute the per-state action certification for each time step $t$ along the trajectory, and the detailed algorithm is in Algorithm 2 (Appendix E). Furthermore, we prove that this certification is theoretically tight as the following proposition shows. The proof is in Appendix F.1.

**Proposition 9.** *Under the same condition as Theorem 8, for any time step $t$, there exists an RL learning algorithm $\mathcal{M}_0$, and a poisoned dataset $\widetilde{D}$, such that $|D \ominus \widetilde{D}| = \overline{K}_t + 1$, and $\widetilde{\pi_P}(s_t) \neq \pi_P(s_t)$.*

## D A TABLE OF POSSIBLE ACTION SET

Table 3: Expressions of possible action set $A(K)$ given poisoning threshold $K$ for different aggregation protocols. Full theorem statements are in Theorems 4, 5 and 7 (in Section 4.4).

| Protocol | $A(K)$ |
|---|---|
| PARL | $\left\{ a \in \mathcal{A} \;\middle\vert\; \sum_{a' \in \mathcal{A}} \max\{n_{a'}(s_t) - n_a(s_t) - K + \mathbf{1}[a' < a], 0\} \leq K \right\}$ |
| TPARL | $\left\{ a \in \mathcal{A} \;\middle\vert\; \sum_{i=1}^{K} h_{a',a}^{(i)} > \delta_{a',a}, \forall a' \neq a \right\}$ |
| DPARL | $\{a_t\} \cup \left\{ a' \in \mathcal{A} \;\middle\vert\; \min_{1 \leq W^* \leq \min\{W_{\max}, t+1\}, W^* \neq W', a'' \neq a_t} L_{a', a''}^{W^*, W'} \leq K \right\} \cup \left\{ a \in \mathcal{A} \;\middle\vert\; \sum_{i=1}^{K} h_{a',a}^{(i)} > \delta_{a',a}, \forall a' \neq a \right\}$ |

For all three aggregation protocols, we summarize the expressions used for computing the possible action set $A(K)$ given tolerable poisoning threshold $K$ in Table 3.

## E ALGORITHM PSEUDOCODE AND DISCUSSION

### E.1 COPA TRAINING PROTOCOL

---
**Algorithm 1:** COPA training protocol.

---
**Input:** training dataset $D$, number of partitions $k$, deterministic hash function $h$
**Output:** COPA subpolicies $\{\pi_i\}_{i=0}^{k-1}$

1 **for** $i \in [k]$ **do**
2 $\quad\lfloor\; D_i \leftarrow \{\tau \in D \mid h(\tau) \equiv i \pmod{k}\}$ $\qquad\qquad$ ▷ Separate the training data $D$ into $k$ partitions
3 **for** *each partition $D_i$* **do**
4 $\quad\lfloor\; \pi_i \leftarrow \mathcal{M}_0(D_i)$ $\qquad$ ▷ Subpolicy trained on partition $D_i$ with offline RL algorithm $\mathcal{M}_0$
5 **return** $\{\pi_i\}_{i=0}^{k-1}$

---

E.2  COPA PER-STATE ACTION CERTIFICATION

**Per-State Partition Aggregation (PARL).**

---

**Algorithm 2:** COPA per-state certification algorithm for Per-State Partition Aggregation (PARL).

---

**Input:** environment $\mathcal{E} = (\mathcal{S}, \mathcal{A}, R, P, H, d_0)$, subpolicies $\{\pi_i\}_{i=0}^{k-1}$
**Output:** COPA robust size at each time step $\{\overline{K}_t\}_{t=0}^{H-1}$

1   $s_0 \sim d_0$                ▷ sample initial state
2   **for** $t$ *from* 0 *to* $H - 1$ **do**
3      **for** *each* $a \in \mathcal{A}$ **do**
4         Compute $n_a(s_t)$ from subpolicies' $\{\pi_i\}_{i=0}^{k-1}$ decisions     ▷ $n_a$ is defined in Section 3
5      Determine the chosen action $a_t \leftarrow \pi_{\mathrm{P}}(s_t)$ according to PARL (Definition 7)
6      Compute $\overline{K}_t$ according to Equation (14) in Theorem 8
7      $s_{t+1} \sim P(s_t, a_t)$
8   **return** $\{\overline{K}_t\}_{t=0}^{H-1}$

---

**Temporal Partition Aggregation (TPARL).**

---

**Algorithm 3:** COPA per-state certification algorithm for Temporal Partition Aggregation (TPARL).

---

**Input:** environment $\mathcal{E} = (\mathcal{S}, \mathcal{A}, R, P, H, d_0)$, subpolicies $\{\pi_i\}_{i=0}^{k-1}$, window size $W$
**Output:** COPA robust size at each time step $\{\overline{K}_t\}_{t=0}^{H-1}$

1   $s_0 \sim d_0$                ▷ sample initial state
2   **for** $t$ *from* 0 *to* $H - 1$ **do**
3      **for** *each* $a \in \mathcal{A}$ **do**
4         Compute $n_a(s_t)$ from subpolicies' $\{\pi_i\}_{i=0}^{k-1}$ decisions     ▷ $n_a$ is defined in Section 3
5      Determine the chosen action $a_t \leftarrow \pi_{\mathrm{T}}(s_t)$ according to TPARL (Definition 8)
6      $p_{\min} \leftarrow \infty$
7      **for** *each* $a' \in \mathcal{A}, a' \neq a_t$ **do**
8         **for** $i$ *from* 0 *to* $k - 1$ **do**
9            Compute $h_{i,a_t,a'}$ according to Equation (4)
10        $\{h_{a_t,a'}^{(i)}\}_{i=1}^{k} \leftarrow \mathtt{sorted}(\{h_{i,a_t,a'}\}_{i=0}^{k-1}, \mathtt{reverse} = \mathtt{True})$
11        Compute $\delta_{a_t,a'}$
12        $\mathtt{sum} \leftarrow 0, p \leftarrow 0$
13        **for** $j$ *from* 1 *to* $k$ **do**
14           **if** $\mathtt{sum} + h_{a_t,a'}^{(j)} > \delta_{a_t,a'}$ **then**
15              $p \leftarrow j - 1$
16              **break**
17           $p \leftarrow j, \mathtt{sum} \leftarrow \mathtt{sum} + h_{a_t,a'}^{(j)}$
18        $p_{\min} \leftarrow \min\{p_{\min}, p\}$
19      $\overline{K}_t \leftarrow p_{\min}$
20      $s_{t+1} \sim P(s_t, a_t)$
21   **return** $\{\overline{K}_t\}_{t=0}^{H-1}$

---

**Dynamic Temporal Partition Aggregation (DPARL).**

---

**Algorithm 4:** COPA per-state certification algorithm for Dynamic Temporal Partition Aggregation (DPARL).

---

**Input:** environment $\mathcal{E} = (\mathcal{S}, \mathcal{A}, R, P, H, d_0)$, subpolicies $\{\pi_i\}_{i=0}^{k-1}$, maximum window size $W_{\max}$
**Output:** COPA robust size at each time step $\{\overline{K}_t\}_{t=0}^{H-1}$

1   $s_0 \sim d_0$                                                 ▷ `sample initial state`
2   **for** $t$ *from* $0$ *to* $H - 1$ **do**
3      **for** *each* $a \in \mathcal{A}$ **do**
4         Compute $n_a(s_t)$ from subpolicies' $\{\pi_i\}_{i=0}^{k-1}$ decisions given $s_t$ ▷ `n_a is defined in Section 3`
5      Determine the chosen action $a_t \leftarrow \pi_D(s_t)$ and chosen window size $W'$ according to DPARL (Definition 3)
6      $p_{\min} \leftarrow \infty$
7      $\overline{K}_t \leftarrow$ Algorithm 3             ▷ `use Algorithm 3 with W replaced by W' to compute` $\overline{K}_t$
8      $\overline{K}_t^D \leftarrow \overline{K}_t$
9      **for** *each* $a' \in \mathcal{A}, a' \neq a_t$ **do**
10         **for** *each* $a'' \in \mathcal{A}, a'' \neq a_t$ **do**
11            **for** $W^*$ *from* $1$ *to* $\min\{W_{\max}, t + 1\}$ **do**
12               $a^{\#} \leftarrow \arg\max_{a_0 \neq a', a_0 \in \mathcal{A}} n_{a_0}(s_{t-W^*+1:t})$
13               **for** $w$ *from* $1$ *to* $\max\{W', W^*\}$ **do**
14                  Compute $\max_{a_0 \in \mathcal{A}} \sigma^w(a_0)$ according to Equation (7)
15               **for** $i$ *from* $0$ *to* $k - 1$ **do**
16                  **for** $w$ *from* $1$ *to* $\max\{W', W^*\}$ **do**
17                     Compute $\sigma^w(\pi_i(s_{t-w}))$ according to Equation (7)
18                  Compute $g_i$ according to Equation (7)
19               $\{g^{(i)}\}_{i=1}^k \leftarrow \texttt{sorted}(\{g_i\}_{i=0}^{k-1}, \texttt{reverse = True})$
20               Compute $n_{a'}^{W^*}, n_{a\#}^{W^*}, n_{a_t}^{W'}, n_{a''}^{W'}$
21               $\texttt{tmp} \leftarrow W'(n_{a'}^{W^*} - n_{a\#}^{W^*}) - W^*(n_{a_t}^{W'} - n_{a''}^{W'}) - \mathbf{1}[a' > a_t]$
22               $\texttt{sum} \leftarrow 0, p \leftarrow 0$
23               **for** $j$ *from* $1$ *to* $k$ **do**
24                  **if** $\texttt{sum} + \texttt{tmp} \geq 0$ **then**
25                     $p \leftarrow j - 1$
26                     **break**
27                  $p \leftarrow j, \texttt{sum} \leftarrow \texttt{sum} + g^{(j)}$
28               $L_{a',a''}^{W^*,W'} \leftarrow p, \overline{K}_t^D \leftarrow \min\{\overline{K}_t^D, L_{a',a''}^{W^*,W'}\}$
29      $s_{t+1} \sim P(s_t, a_t)$
30 **return** $\{\overline{K}_t^D\}_{t=0}^{H-1}$

---

### E.3   COPA CUMULATIVE REWARD CERTIFICATION

COPA-SEARCH alternately executes the procedure of *trajectory exploration and expansion* and *poisoning threshold growth*. In trajectory exploration and expansion, COPA-SEARCH organizes all possible trajectories in the form of a search tree and progressively grows it. For each node (representing a state), we leverage Theorems 4, 5 and 7 to compute the Possible Action Set. We then expand the tree branches corresponding to the actions in the derived set. In poisoning threshold growth, when all trajectories for the current poisoning threshold are explored, we increase $K'$ to seek certification under larger poisoning sizes, via maintaining a priority queue of all poisoning sizes during the expansion of the tree. The iterative procedures end when the priority queue becomes empty.

The highlighted line in the following algorithm needs to inject different algorithms based on the aggregation protocol: for PARL ($\pi_P$), use Theorem 4; for TPARL ($\pi_T$), use Theorem 5; for DPARL ($\pi_D$), use Theorem 7.

---

**Algorithm 5:** COPA-SEARCH: adaptive tree search for cumulative reward certification.

---

**Input:** environment $\mathcal{E} = (\mathcal{S}, \mathcal{A}, R, P, H, d_0)$, subpolicies $\{\pi_i\}_{i=0}^{k-1}$, aggregated policy $\pi_P$ or $\pi_T$ (with $W$) or $\pi_T$ (with $W_{\max}$)

**Output:** a map $M$ that maps poisoning size $K$ to corresponding certified lower bound of cumulative reward $\underline{J}_K$

    ▷ Initialize global variables

1   p_que $\leftarrow \emptyset$              ▷ initialize an empty priority queue containing tuples of (state history$s_{0:t}$, action $a$, poisoning size $K$, reward $J$) sorted by increasing $K$

2   $J_{\text{global}} \leftarrow \infty$              ▷ initialize global minimum reward

3   **Function** GETACTIONS $(s_{0:t}, K_{\lim}, J_{\text{cur}})$:

4      $A \leftarrow$ possibleActions$(s_{0:t}, \{\pi_i\}_{i=1}^{k-1}, \pi_*, K_{\lim})$    ▷ compute the possible action set given state history, subpolicies, aggregation policy $\pi_*$, and poisoning size according to theorems in Section 4.4

5      a_list $\leftarrow \emptyset$

6      **for** *each action* $a \in \mathcal{A}$ **do**

7          **if** $P(s, a) = \perp$ **then**

8              **continue**     ▷ game terminate here, no possible larger or lower cumulative reward to search

9          a_list $\leftarrow$ a_list $\cup \{a\}$            ▷ record possible actions to expand

10      **if** $A \neq \mathcal{A}$ **then**

11          $K' \leftarrow \min_{\text{possibleActions}(s_{0:t}, \{\pi_i\}_{i=0}^{k-1}, \pi_*, K) \neq A} K$

12          $A_{\text{new}} \leftarrow$ possibleActions$(s_{0:t}, \{\pi_i\}_{i=1}^{k-1}, \pi_*, K') \setminus A$ ▷ compute the immediate actions that will possibly be chosen if enlarging the poisoning size

13          **for** *each action* $a \in A_{\text{new}}$ **do**

14              p_que.push$((s_{0:t}, a, K', J_{\text{cur}}))$

15      **return** a_list

16   **Procedure** EXPAND $(s_{0:t}, K_{\lim}, J_{\text{cur}})$:

17      **if** $J_{\text{cur}} \geq J_{\text{global}} \land$ (*step reward is non-negative*) **then**

18          **return**                             ▷ pruning

19      a_list $\leftarrow$ GETACTIONS $(s_{0:t}, \{\pi_i\}_{i=0}^{k-1}, \pi_*, K_{\lim})$      ▷ compute according to theorems in Section 4.4

20      **if** a_list $= \emptyset$ **then**

21          $J_{\text{global}} \leftarrow \min\{J_{\text{global}}, J_{\text{cur}}\}$

22          **return**

23      **for** $a \in$ a_list **do**

24          $s_{t+1} \leftarrow P(s_t, a)$

25          **if** $s_{t+1} = \perp$ **then**

26              $J_{\text{global}} \leftarrow \min\{J_{\text{global}}, J_{\text{cur}}\}$

27          **else**

28              EXPAND $(s_{0:t+1}, K_{\lim}, J_{\text{cur}} + R(s_t, a))$

29   $M \leftarrow \emptyset$

30   $s_0 \leftarrow d_0$

                                     ▷ initial state is $s_0$

31   EXPAND $(s_0, K_{\lim} = 0, J_{\text{cur}} = 0)$           ▷ expand initial trajectory

32   **while** True **do**

33      **if** p_que $= \emptyset$ **then**

34          **break**                          ▷ no state to expand

35      $(s_{0:t}, a, K, J) \leftarrow$ p_que.pop()           ▷ pop out the first element

36      $(\_, \_, K', \_) \leftarrow$ p_que.top()            ▷ examine the next element

37      $M(K) \leftarrow J_{\text{global}}$          ▷ obtain one new point of certification result

38      $s_{t+1} \leftarrow P(s_t, a)$

39      EXPAND $(s_{0:t+1}, K', J + R(s_t, a))$          ▷ expand the tree from the new node

40   **return** $M$      ▷ indeed the algorithm can terminate at any time within the while loop

---

**Time Complexity.** The complexity of COPA-SEARCH is $O(H|\mathcal{S}_{\text{explored}}|(\log|\mathcal{S}_{\text{explored}}| + |\mathcal{A}|T))$, where $|\mathcal{S}_{\text{explored}}|$ is the number of explored states throughout the search procedure, which is no larger than cardinality of state set $\mathcal{S}$, $H$ is the horizon length, $|\mathcal{A}|$ is the cardinality of action set, and $T$ is the time complexity of per-state action certification. The main bottleneck of the algorithm is the large number of possible states, which is in the worse case exponential to state dimension. However, to provide a deterministic worst-case certification agnostic to environment properties, exploring all possible states may be inevitable.

**Relation with Wu et al. (2022).** The COPA-SEARCH algorithm is inspired from Wu et al. (2022, Algorithm 3), which also leverages tree search to explore all possible states and thus derive the robustness guarantee. However, the major distinction is that their algorithm tries to derive a probabilistic guarantee of RL robustness against state perturbations, while COPA-SEARCH derives deterministic guarantee of RL robustness against poisoning attacks. We think this general tree search methodology can be further extended to provide certification beyond evasion attack (Wu et al., 2022) and poisoning attack (our work) and we leave it as future work.

**Extension to stochastic MDPs.** In the current version of COPA-SEARCH, the exhaustive search is enabled by the deterministic MDP assumption. However, we foresee the potential of conveniently extending COPA-SEARCH to stochastic MDPs and will discuss one concrete method below. In contrast to interacting with the environment for one time to obtain the *deterministic* next state transition in the deterministic MDP case, in the case of stochastic MDPs, we can leverage *sampling* (*i.e.*, by repeatedly taking the same action at the same state) to obtain the set of high probability next state transitions with high confidence. In this way, our COPA-SEARCH will be able to yield a *probabilistic bound*, in comparison with the *deterministic bound* achieved in this paper enabled by the deterministic MDP assumption.

## F PROOFS

### F.1 TIGHTNESS OF PER-STATE ACTION CERTIFICATION IN PARL

*Proof of Proposition 9.* We prove by construction. Given the state $s_t$, we first locate the subpolicies whose chosen action is $a$, and denote the set of them as
$$B = \{i \in [u] \mid \pi_i(s_t) = a = \pi_{\text{P}}(s_t)\}.$$
We also denote $a' = \arg\max_{a' \neq a} n_{a'}(s_t) + \mathbf{1}[a' < a]$. According to $\overline{K}_t$'s definition (Equation (14)),
$$|B| = n_a(s_t) > n_a(s_t)/2 \geq \overline{K}_t.$$
We now pick an arbitrary subset $B' \subseteq B$ such that $|B'| = \overline{K}_t + 1$. For each $i \in B'$, we locate its corresponding parititioned dataset $D_i$ for training subpolicy $\pi_i$. We insert one point $p_i$ to $D_i$, such that our chosen learning algorithm $\mathcal{M}_0$ can train a subpolicy $\tilde{\pi}_i = \mathcal{M}_0(D_i \cup \{p_i\})$ that satisfies $\tilde{\pi}_i(s_t) = a'$. For example, the point could be $p_i = \{(s_t, a', s', \infty)\}$ and $\mathcal{M}_0$ learns the action with maximum reward for the memorized nearest state, where $s'$ can be adjusted to make sure the point is hashed to parition $i$.[1] Then, we construct the poisoned dataset
$$\widetilde{D} = \left(\bigcup_{i \in B'} D_i \cup \{p_i\}\right) \cup \left(\bigcup_{i \in [u] \setminus B'} D_i\right).$$
Therefore, we have $\widetilde{D} \ominus D = \cup_{i \in B'} p_i = |B'| = \overline{K}_t + 1$.

On this poisoned dataset, we train subpolicies $\{\tilde{\pi}_i\}_{i=0}^{u-1}$ and get the aggregated policy $\widetilde{\pi_{\text{P}}}$. To study $\widetilde{\pi_{\text{P}}}(s_t)$, we compute the aggregated action count $\tilde{n}_a$ on these poisoned subpolicies. We found that

---

[1]Strictly speaking, we need to choose a determinisitc hash function $h$ for dataset partitioning such that adjustment on $s'$ and reward (currently $\infty$, but can be an arbitrary large enough number) can make the point being partitioned to $i$, *i.e.*, $h(p_i) \equiv i \pmod u$. Since our adjustment space is infinite due to infinite number of large enough reward, such assumption can be easily achieved. Same applies to other attack constructions in the following proofs.

$\widetilde{n}_a(s_t) = n_a(s_t) - |B'|$ and $\widetilde{n}_{a'}(s_t) = n_{a'}(s_t) + |B'|$. Therefore,

$$\widetilde{n}_a(s_t) - \widetilde{n}_{a'}(s_t) = n_a(s_t) - n_{a'}(s_t) - 2(\overline{K}_t + 1)$$

$$= n_a(s_t) - n_{a'}(s_t) - 2\left(\left\lfloor \frac{n_a(s_t) - n_{a'}(s_t) - \mathbf{1}[a' < a]}{2} \right\rfloor + 1\right)$$

$$< n_a(s_t) - n_{a'}(s_t) - (n_a(s_t) - n_{a'}(s_t) - \mathbf{1}[a' < a]) = \mathbf{1}[a' < a].$$

Therefore, if $a' < a$, $\widetilde{n}_a(s_t) \le \widetilde{n}_{a'}(s_t)$, and $a'$ has higher priority to be chosen than $a$; if $a' > a$, $\widetilde{n}_a(s_t) \le \widetilde{n}_{a'}(s_t) - 1$, and $a'$ still has higher priority to be chosen than $a$. Hence, $\widetilde{\pi}_\mathrm{P}(s_t) \ne a = \pi_\mathrm{P}(s_t)$. To this point, we conclude the proof with a feasible construction. $\qquad\square$

### F.2 PER-STATE ACTION CERTIFICATION IN TPARL

*Proof of Theorem 1.* For ease of notation, we let $w = \min\{W, t+1\}$ so $w$ is the actual window size used at step $t$. We let $t_0 = t - w + 1$, *i.e.*, $t_0$ is the actual start time step for TPARL aggregation at step $t$. Now we can write the chosen action at step $t$ without poisoning as $a = \pi_\mathrm{T}(s_{t_0:t})$.

We prove the theorem by contradiction: We assume that there is a poisoning attack whose poisoning size $K \le \overline{K}_t$ where $\overline{K}_t$ is defined by Equation (3) in the theorem, and after poisoning, the chosen action is $a^T \ne a$. We denote $\{\widetilde{\pi}_i\}_{i=0}^{u-1}$ to the poisoned subpolicies and $\widetilde{\pi}_\mathrm{T}$ to the poisoned TPARL aggregated policy. From the definition of $\overline{K}_t$, we have

$$\sum_{i=1}^{K} h_{a,a^T}^{(i)} \le \sum_{i=1}^{\overline{K}_t} h_{a,a^T}^{(i)} \le \delta_{a,a^T}, \tag{15}$$

since $K \le \overline{K}_t$, and each $h_{a,a^T}^{(i)}$ is an element of $h_{i',a,a^T}$ for some $i'$ where

$$h_{i',a,a^T} = \sum_{j=0}^{w-1} \mathbf{1}_{i',a}(s_{t-j}) + w - \sum_{j=0}^{w-1} \mathbf{1}_{i',a^T}(s_{t-j}) \ge 0$$

by Equation (4) in the theorem. Since the poisoning attack within threshold $K$ can at most affect $K$ subpolicies, we let $B$ be the set of affected policies and assume $|B| = K$ without loss of generality. Formally, $B = \{i \in [u] \mid \exists t' \in [t_0, t], \widetilde{\pi}_i(s_{t'}) \ne \pi_i(s_{t'})\}$. Therefore, according to the monotonicity of $h_{a,a^T}^{(i)}$, from Equation (15),

$$\sum_{i \in B} h_{i,a,a^T} \le \sum_{i=1}^{K} h_{a,a^T}^{(i)} \le \delta_{a,a^T}. \tag{16}$$

According to the assumption of successfully poisoning attack, after attack, the sum of aggregated vote for $a^T$ plus $\mathbf{1}[a^T < a]$ should be larger then that of $a$. Formally, after the poisoning,

$$\widetilde{n}_{a^T}(s_{t_0:t}) + \mathbf{1}[a^T < a] > \widetilde{n}_a(s_{t_0:t})$$

or equivalently

$$\widetilde{n}_a(s_{t_0:t}) - (\widetilde{n}_{a^T}(s_{t_0:t}) + \mathbf{1}[a^T < a]) < 0.$$

From the statement of Theorem 1,

$$\delta_{a,a^T} = n_a(s_{t_0:t}) - (n_{a^T}(s_{t_0:t}) + \mathbf{1}[a^T < a]).$$

Take the difference of the above two equations, we know that the attack satisfies

$$n_a(s_{t_0:t}) - \widetilde{n}_a(s_{t_0:t}) - (n_{a^T}(s_{t_0:t}) - \widetilde{n}_{a^T}(s_{t_0:t})) > \delta_{a,a^T}. \tag{17}$$

Since the attack only changes the subpolicies in $B$, we have

$$n_a(s_{t_0:t}) - \widetilde{n}_a(s_{t_0:t}) = \sum_{i \in B} \left( \sum_{j=0}^{w-1} \mathbf{1}[\pi_i(s_{t-j}) = a] - \sum_{j=0}^{w-1} \mathbf{1}[\widetilde{\pi}_i(s_{t-j}) = a] \right)$$

$$\le \sum_{i \in B} \sum_{j=0}^{w-1} \mathbf{1}_{i,a}(s_{t-j}),$$

$$n_{a^T}(s_{t_0:t}) - \widetilde{n}_{a^T}(s_{t_0:t}) = \sum_{i \in B} \left( \sum_{j=0}^{w-1} \mathbf{1}[\pi_i(s_{t-j}) = a^T] - \sum_{j=0}^{w-1} \mathbf{1}[\widetilde{\pi}_i(s_{t-j}) = a^T] \right)$$

$$\ge \sum_{i \in B} \left( \sum_{j=0}^{w-1} \mathbf{1}_{i,a^T}(s_{t_j}) - w \right).$$

Inject them into Equation (17) yields

$$\sum_{i \in B} \underbrace{\left( \sum_{j=0}^{w-1} \mathbf{1}_{i,a}(s_{t-j}) + w - \sum_{j=0}^{w-1} \mathbf{1}_{i,a^T}(s_{t-j}) \right)}_{h_{i,a,a^T}} > \delta_{a,a^T}$$

which contradicts Equation (16) and thus concludes the proof. $\qquad \square$

### F.3 TIGHTNESS OF PER-STATE ACTION CERTIFICATION IN TPARL

*Proof of Proposition 2.* For ease of notation, we let $w = \min\{W, t+1\}$ so $w$ is the actual window size used at step $t$. We let $t_0 = t - w + 1$, *i.e.*, $t_0$ is the actual start time step for TPARL aggregation at step $t$. Now we can write the chosen action at step $t$ without poisoning as $a = \pi_T(s_{t_0:t})$.

We prove by construction, *i.e.*, we construct an poisoning attack with poisoning size $\overline{K}_t + 1$ that deviates the prediction of the poisoned policy. Specifically, we aim to craft a poisoned dataset $\widetilde{D}$ with poisoning size $|D \ominus \widetilde{D}| = \overline{K}_t + 1$, such that for certain learning algorithm $\mathcal{M}_0$, after partitioning and learning on the poisoned dataset, the poisoned subpolicies $\tilde{\pi}_i = \mathcal{M}_0(\widetilde{D}_i)$ can be aggregated to produce different action prediction: $\widetilde{\pi_T}(s_{t_0:t}) \neq a$.

Before construction, we first show that $\overline{K}_t$ given by Equation (3) satisfies $\overline{K}_t < u$. If $\overline{K}_t = u$, it means that for arbitrary $a' \neq a$,

$$\sum_{i=1}^{u} h_{a,a'}^{(i)} = \sum_{i=0}^{u-1} h_{i,a,a'} = \sum_{i=0}^{u-1} \left( \sum_{j=0}^{w-1} \mathbf{1}_{i,a}(s_{t-j}) + w - \sum_{j=0}^{w-1} \mathbf{1}_{i,a'}(s_{t-j}) \right) = n_a(s_{t_0:t}) + uw - n_{a'}(s_{t_0:t})$$

$$\leq \delta_{a,a'} = n_a(s_{t_0:t}) - n_{a'}(s_{t_0:t}) - \mathbf{1}[a' < a],$$

which implies $uw \leq 0$ contracting $uw > 0$. Now, we know $\overline{K}_t < u$, so $\overline{K}_t + 1$, our poisoning size, is smaller or equal to $u$.

We start our construction by choosing an action $a^T$:

$$a^T = \underset{a' \neq a, a' \in \mathcal{A}}{\arg\min} \ \underset{\sum_{i=1}^{p} h_{a,a'}^{(i)} \leq \delta_{a,a'}}{\arg\max} \ p.$$

According to the definition of $\overline{K}_t$ in Equation (3), for $a^T$ we have

$$\sum_{i=1}^{\overline{K}_t + 1} h_{a,a^T}^{(i)} > \delta_{a,a^T}. \tag{18}$$

We locate the subpolicies to poison as

$$B = \{ i \in [u] \mid h_{a,a^T}^i \text{ is } h_{a,a^T}^{(j)} \text{ in the nonincreasing permutation in Equation (4)}, j \leq \overline{K}_t + 1 \} \tag{19}$$

Therefore, $|B| = \overline{K}_t + 1$ and

$$\sum_{i \in B} h_{i,a,a^T} > \delta_{a,a^T} \tag{20}$$

by Equation (18). For each of these subpolicies $i \in B$, we locate its corresponding partitioned dataset $D_i$ for training subpolicy $\pi_i$, and insert one trajectory $p_i$ to $D_i$, such that our chosen learning algorithm $\mathcal{M}_0$ can train a subpolicy $\tilde{\pi}_i = \mathcal{M}(D_i \cup \{p_i\})$ satisfying $\tilde{\pi}_i(s_{t'}) = a^T$ for any $t' \in [t_0, t]$. For example, the trajectory could be $p_i = \{(s_{t'}, a^T, s', \infty)\}_{t'=t_0}^{t}$ and $\mathcal{M}_0$ learns the action with maximum reward for the memorized nearest state, where $s'$ can be adjusted to make sure the trajectory is hashed to partition $i$. Then, we construct the poisoned dataset

$$\widetilde{D} = \left( \bigcup_{i \in B'} D_i \cup \{p_i\} \right) \cup \left( \bigcup_{i \in [u] \setminus B'} D_i \right).$$

Therefore, we have $\widetilde{D} \ominus D = \cup_{i \in B'} p_i = |B'| = \overline{K}_t + 1$. Now, we compare the aggregated votes for $a$ and $a^T$ before and after poisoning:

$$\widetilde{n}_a(s_{t_0:t}) - n_a(s_{t_0:t}) = \sum_{i \in B} \left( \sum_{j=0}^{w-1} \mathbf{1}[\widetilde{\pi}_i(s_{t-j}) = a] - \sum_{j=0}^{w-1} \mathbf{1}[\pi_i(s_{t-j}) = a] \right)$$

$$= -\sum_{i \in B} \sum_{j=0}^{w-1} \mathbf{1}[\pi_i(s_{t-j}) = a],$$

$$\widetilde{n}_{a^T}(s_{t_0:t}) - n_{a^T}(s_{t_0:t}) = \sum_{i \in B} \left( \sum_{j=0}^{w-1} \mathbf{1}[\widetilde{\pi}_i(s_{t-j}) = a^T] - \sum_{j=0}^{w-1} \mathbf{1}[\pi_i(s_{t-j}) = a^T] \right)$$

$$= \sum_{i \in B} \left( w - \sum_{j=0}^{w-1} \mathbf{1}[\pi_i(s_{t-j}) = a^T] \right).$$

Now we compare the margin between aggregated votes for $a$ and $a^T$ after poisoning:

$$\widetilde{n}_{a^T}(s_{t_0:t}) - \widetilde{n}_a(s_{t_0:t}) + \mathbf{1}[a^T < a]$$

$$= n_{a^T}(s_{t_0:t}) - n_a(s_{t_0:t}) + \mathbf{1}[a^T < a] + \widetilde{n}_{a^T}(s_{t_0:t}) - n_{a^T}(s_{t_0:t}) - (\widetilde{n}_a(s_{t_0:t}) - n_a(s_{t_0:t}))$$

$$= -\delta_{a,a^T} + \sum_{i \in B} \left( \sum_{j=0}^{w-1} \mathbf{1}[\pi_i(s_{t-j}) = a] + w - \sum_{j=0}^{w-1} \mathbf{1}[\pi_i(s_{t-j}) = a^T] \right)$$

$$= -\delta_{a,a^T} + \sum_{i \in B} h_{i,a,a^T} > 0.$$

As a result, after poisoning, $a^T$ has higher priority to be chosen than $a$, *i.e.*, $\widetilde{\pi}_{\mathrm{T}}(s_{t_0:t}) \neq a = \pi_{\mathrm{T}}(s_{t_0:t})$. To this point, we conclude the proof with a feasible attack construction. $\square$

### F.4 Loose Per-State Action Certification in TPARL and Comparision with Tight One

The following corollary of Theorem 8 states a loose per-state action certification.

**Corollary 10.** *Under the same condition as Theorem 1,*

$$\overline{K}_t = \left\lfloor \frac{n_a(s_{\max\{t-W+1,0\}:t}) - \max_{a' \neq a} \left( n_{a'}(s_{\max\{t-W+1,0\}:t}) + \mathbf{1}[a' < a] \right)}{2 \min\{W, t+1\}} \right\rfloor \qquad (21)$$

*is a* tolerable poisoning threshold *at time step $t$ in Definition 1, where $W$ is the window size.*

*Proof of Corollary 10.* We let $w = \min\{t+1, W\}$ to be the actual aggregation window size at time step $t$. After poisoning with size $K$, at most $K$ subpolicies are affected and each affected policy can only make $\pm w$ changes to the aggregated action count. This implies that, after poisoning, for any action $a' \in \mathcal{A}$, the aggregated vote count $\widetilde{n}_{a'}(s_{t-w+1:t}) \in [n_{a'}(s_{t-w+1:t}) - uw, n_{a'}(s_{t-w+1:t}) + uw]$. Thus, when $K \leq \overline{K}_t$ where $\overline{K}_t$ is defined in Theorem 1, for any $a' \neq a$, we have

$$\tilde{n}_a(s_{t-w+1:t}) - \tilde{n}_{a'}(s_{t-w+1:t}) - \mathbf{1}[a' < a]$$

$$= n_a(s_{t-w+1:t}) - n_{a'}(s_{t-w+1:t}) - \mathbf{1}[a' < a] - 2Kw$$

$$\geq n_a(s_{t-w+1:t}) - n_{a'}(s_{t-w+1:t}) - \mathbf{1}[a' < a] - 2w \cdot \overline{K}_t$$

$$\geq n_a(s_{t-w+1:t}) - n_{a'}(s_{t-w+1:t}) - \mathbf{1}[a' < a] - \left( n_a(s_{t-w+1:t}) - \max_{a'' \neq a}(n_{a''}(s_{t-w+1:t}) + \mathbf{1}[a'' < a]) \right)$$

$$= \max_{a'' \neq a}(n_{a''}(s_{t-w+1:t}) + \mathbf{1}[a'' < a]) - (n_{a'}(s_{t-w+1:t}) + \mathbf{1}[a' < a]) \geq 0.$$

From the definition of TPARL protocol, the poisoned policy still chooses action $a$, which implies $\overline{K}_t$ is a tolerable poisoning threshold. $\square$

In the main text, we mention that the certification from Corollary 10 is looser than that from Theorem 1. This assertion is based on the following two facts:

1. According to Proposition 2, the certification given by Theorem 1 is theoretically tight, which means that any other certification can only be as tight as Theorem 1 or looser than it.

2. There exists examples where the computed $\overline{K}_t$ from Theorem 1 is larger than that from Corollary 10.
   For instance, suppose $W = 5$, action set $\mathcal{A} = \{a_1, a_2\}$, and there are three subpolicies. At time step $t = 4$, $\pi_i$ for $s_0$ to $s_4$ are $[a_1, a_1, a_1, a_1, a_2]$ for all subpolicies (*i.e.*, $i \in [3]$). Thus, the

benign policy $\pi_{\mathrm{T}}(s_{0:4}) = a_1$. By computation, the $\overline{K}_t$ from Theorem 1 is 1; while the $\overline{K}_t$ from Corollary 10 is 0.

Indeed, Corollary 10 can be viewed as using $2w$ to upper bound $h_{i,a,a'} = w + \sum_{j=0}^{w-1} \mathbf{1}_{i,a}(s_{t-j}) - \sum_{j=0}^{w-1} \mathbf{1}_{i,a'}(s_{t-j})$. Intuitively, Corollary 10 assumes every subpolicy can provide $2w$ vote margin shrinkage, and Theorem 1 uses $h_{i,a,a'}$ to capture the precise worse-case margin shrinkage and thus provides a tighter certification.

### F.5 PER-STATE ACTION CERTIFICATION IN DPARL

*Proof of Theorem 3.* Without loss of generality, we assume $W_{\max} \leq t + 1$ and otherwise we let $W_{\max} \leftarrow \min\{W_{\max}, t+1\}$. We let $t_0 = \max\{t - W_{\max} + 1, 0\}$ be the start time step of the maximum possible window. To prove the theorem, our general methodology is to enumerate all possible cases of a successful attack, and derive the tolerable poisoning threshold for each case respectively. Taking a minimum over these tolerable poisoning thresholds gives the required result.

Specifically, we denote $\mathcal{P}$ to the predicate of robustness under poisoning attack: $\mathcal{P} = [\widetilde{\pi_{\mathrm{D}}}(s_{t_0:t}) = a]$, and denote $K$ to the poisoning attack size. Therefore, we can decompose $\mathcal{P}$ as such:

$$\mathcal{P} = \mathcal{P}(W') \wedge \bigwedge_{\substack{1 \leq W^* \leq W_{\max}, W^* \neq W' \\ a' \neq a}} \neg \mathcal{Q}(W^*, a'). \tag{22}$$

Recall that $W'$ is the chosen window size by the protocol DPARL with unattacked subpolicies $\pi_{\mathrm{D}}$ (Equation (1)). In Equation (22), the predicate $\mathcal{P}(W')$ means that after poisoning attack, whether the prediction under window size $W'$ is still $a$; the predicate $\mathcal{Q}(W^*, a')$ means that after poisoning attack, whether the chosen action is $a'$ at window size $W^*$ and average vote margin is larger (or equal if $a' < a$) at window size $W^*$ compared to $W'$. Formally, let $\widetilde{n}_a$ be the aggregated action count after poisoning and $\widetilde{\Delta}_t^W$ be the average vote margin after poisoning at window $W$ (see Equation (2)),

$$\mathcal{P}(W') = \left( \arg\max_{a \in \mathcal{A}} \widetilde{n}_a(s_{t-W'+1:t}) = a \right),$$

$$\mathcal{Q}(W^*, a') = \left( \arg\max_{a \in \mathcal{A}} \widetilde{n}_a(s_{t-W^*+1:t}) = a' \right) \wedge \tag{23}$$

$$\left( \left( \left( \widetilde{\Delta}_t^{W^*} \geq \widetilde{\Delta}_t^{W'} \right) \wedge (a' < a) \right) \vee \left( \left( \widetilde{\Delta}_t^{W^*} > \widetilde{\Delta}_t^{W'} \right) \wedge (a' > a) \right) \right).$$

According to Theorem 1,

$$K \leq \overline{K}_t \implies \mathcal{P}(W')$$

where $\overline{K}_t$ is defined by Equation (3) with $W$ replaced by $W'$. The following Lemma 11 shows a sufficient condition for $Q(W^*, a')$. We then aggregate these conditions together with minimum to obtain a sufficient condition for $\mathcal{P}$:

$$K \leq \min \left\{ \overline{K}_t, \min_{1 \leq W^* \leq \min\{W_{\max}, t+1\}, W^* \neq W', a' \neq a, a'' \neq a} L_{a', a''}^{W^*, W'} \right\}$$

and thus conclude the proof. $\qquad\square$

**Lemma 11.** *Let $\mathcal{Q}(W^*, a')$, $K$, $W'$ be the same as defined in proof of Theorem 3, then*

$$K \leq \min_{a'' \neq a} L_{a', a''}^{W^*, W'} \implies \neg \mathcal{Q}(W^*, a'), \tag{24}$$

*where $L_{a', a''}^{W^*, W'}$ is defined in Definition 4.*

*Proof.* We prove the equivalent form:

$$\mathcal{Q}(W^*, a') \implies K > \min_{a'' \neq a} L_{a', a''}^{W^*, W'}. \tag{25}$$

Suppose a poisoning attack can successfully achieve $Q(W^*, a')$, we now induce the requirement on its poisoning size $K$. First, we notice that

$$\left( \left( \widetilde{\Delta}_t^{W^*} \geq \widetilde{\Delta}_t^{W'} \right) \wedge (a' < a) \right) \vee \left( \left( \widetilde{\Delta}_t^{W^*} > \widetilde{\Delta}_t^{W'} \right) \wedge (a' > a) \right)$$

$$\iff W^* W' \widetilde{\Delta}_t^{W^*} \geq W^* W' \widetilde{\Delta}_t^{W'} + \mathbf{1}[a' > a] \tag{26}$$

since $W^*W'\widetilde{\Delta}_t^{W*/W^*}$ is an integer by definition. According to the definition and $\mathcal{Q}(W^*,a')$'s assumption that $\arg\max_{a\in\mathcal{A}}\widetilde{n}_a(s_{t-W^*+1:t})=a'$,

$$W^*W'\widetilde{\Delta}_t^{W^*} \le W'\left(\widetilde{n}_{a'}(s_{t-W^*+1:t}) - \widetilde{n}_{a\#}(s_{t-W^*+1:t})\right),$$

where "$\le$" comes from the fact that the margin in $\widetilde{\Delta}_t^{W^*}$ should be with respect to the runner-up class after poisoning, and computing with respect to any other class provides an upper bound. Here we choose $a^\# = \arg\max_{a_0\neq a',a_0\in\mathcal{A}} n_{a_0}(s_{t-W^*+1:t})$ (see Definition 4), the runner-up class before poisoning, to empirically shrink the gap between the bound and actual margin. On the other hand,

$$W^*W'\widetilde{\Delta}_t^{W'} \ge W^*\left(\widetilde{n}_a(s_{t-W'+1:t}) - \max_{a''\neq a}\widetilde{n}_{a''}(s_{t-W'+1:t})\right),$$

where "$\ge$" comes from the fact that the margin in $\widetilde{\Delta}_t^{W'}$ should use the top class after poisoning, and computing with any other class provides a lower bound. Thus, from Equation (26) and the above two relaxations, we get

$Q(W^*,a')$

$$\Longrightarrow W'\left(\widetilde{n}_{a'}(s_{t-W^*+1:t}) - \widetilde{n}_{a\#}(s_{t-W^*+1:t})\right) \ge W^*\left(\widetilde{n}_a(s_{t-W'+1:t}) - \max_{a''\neq a}\widetilde{n}_{a''}(s_{t-W'+1:t})\right) + \mathbf{1}[a'>a]$$

$$\Longrightarrow \exists a''\neq a,$$

$$W'\left(\widetilde{n}_{a'}(s_{t-W^*+1:t}) - \widetilde{n}_{a\#}(s_{t-W^*+1:t})\right) \ge W^*\left(\widetilde{n}_a(s_{t-W'+1:t}) - \widetilde{n}_{a''}(s_{t-W'+1:t})\right) + \mathbf{1}[a'>a].$$

For each $a''\neq a$, now we use the last equation as the condition, and show that $K > L_{a',a''}^{W^*,W'}$ is a necessary condition. This proposition is equivalent to

$K \le L_{a',a''}^{W^*,W'}$

$$\Longrightarrow W'\left(\widetilde{n}_{a'}(s_{t-W^*+1:t}) - \widetilde{n}_{a\#}(s_{t-W^*+1:t})\right) < W^*\left(\widetilde{n}_a(s_{t-W'+1:t}) - \widetilde{n}_{a''}(s_{t-W'+1:t})\right) + \mathbf{1}[a'>a].$$

$$\tag{27}$$

Suppose a poisoning attack within poisoning size $K$ changes the subpolicies in set $B \subseteq [u]$. Note that $|B| \le K$. We inspect the objective in Equation (27):

$$W'\left(\widetilde{n}_{a'}(s_{t-W^*+1:t}) - \widetilde{n}_{a\#}(s_{t-W^*+1:t})\right) - W^*\left(\widetilde{n}_a(s_{t-W'+1:t}) - \widetilde{n}_{a''}(s_{t-W'+1:t})\right) - \mathbf{1}[a'>a]$$

$$=W'\left(n_{a'}(s_{t-W^*+1:t}) - n_{a\#}(s_{t-W^*+1:t})\right) - W^*\left(n_a(s_{t-W'+1:t}) - n_{a''}(s_{t-W'+1:t})\right) - \mathbf{1}[a'>a]$$

$$+\sum_{i\in B}\left(W'\sum_{w=0}^{W^*-1}\mathbf{1}[\widetilde{\pi}_i(s_{t-w+1})=a'] - W'\sum_{w=0}^{W^*-1}\mathbf{1}[\widetilde{\pi}_i(s_{t-w+1})=a^\#]\right.$$

$$\left.-W^*\sum_{w=0}^{W'-1}\mathbf{1}[\widetilde{\pi}_i(s_{t-w+1})=a] + W^*\sum_{w=0}^{W'-1}\mathbf{1}[\widetilde{\pi}_i(s_{t-w+1})=a'']\right)$$

$$-\sum_{i\in B}\left(W'\sum_{w=0}^{W^*-1}\mathbf{1}[\pi_i(s_{t-w+1})=a'] - W'\sum_{w=0}^{W^*-1}\mathbf{1}[\pi_i(s_{t-w+1})=a^\#]\right.$$

$$\left.-W^*\sum_{w=0}^{W'-1}\mathbf{1}[\pi_i(s_{t-w+1})=a] + W^*\sum_{w=0}^{W'-1}\mathbf{1}[\pi_i(s_{t-w+1})=a'']\right)$$

$$=W'\left(n_{a'}(s_{t-W^*+1:t}) - n_{a\#}(s_{t-W^*+1:t})\right) - W^*\left(n_a(s_{t-W'+1:t}) - n_{a''}(s_{t-W'+1:t})\right) - \mathbf{1}[a'>a]$$

$$+\sum_{i\in B}\sum_{w=0}^{\max\{W^*,W'\}}\sigma^w(\widetilde{\pi}_i(s_{t-w})) - \sigma^w(\pi_i(s_{t-w}))$$

$$\le W'\left(n_{a'}(s_{t-W^*+1:t}) - n_{a\#}(s_{t-W^*+1:t})\right) - W^*\left(n_a(s_{t-W'+1:t}) - n_{a''}(s_{t-W'+1:t})\right) - \mathbf{1}[a'>a]$$

$$+\sum_{i\in B}\underbrace{\sum_{w=0}^{\max\{W^*,W'\}}\max_{a_0\in\mathcal{A}}\sigma^w(a_0) - \sigma^w(\pi_i(s_{t-w}))}_{g_i}$$

$$\overset{(a)}{\le}W'\left(n_{a'}(s_{t-W^*+1:t}) - n_{a\#}(s_{t-W^*+1:t})\right) - W^*\left(n_a(s_{t-W'+1:t}) - n_{a''}(s_{t-W'+1:t})\right) - \mathbf{1}[a'>a] + \sum_{i=1}^{K}g^{(i)}$$

$$\overset{(b)}{\leq} W'\left(n_{a'}(s_{t-W^*+1:t}) - n_{a\#}(s_{t-W^*+1:t})\right) - W^*\left(n_a(s_{t-W'+1:t}) - n_{a''}(s_{t-W'+1:t})\right) - \mathbf{1}[a' > a] + \sum_{i=1}^{L_{a',a''}^{W^*,W'}} g^{(i)}$$

$$\overset{(c)}{<} 0.$$

Thus, Equation (27) is proved. Therefore, $K > L_{a',a''}^{W^*,W'}$ is a necessary condition for $Q(W^*, a')$, *i.e.*, Equation (25).

In the above deriivation, the definitions of $g^i$, $g^i$, and $\sigma^w$ are from Equation (7). $(a)$ comes from the facts that $\{g^{(i)}\}_{i=1}^u$ is a nondecreasing permutation of $\{g_i\}_{i=0}^{u-1}$, $g_i \geq 0$, and $|B| \leq K$. $(b)$ comes from the assumption $K \leq L_{a,a''}^{W^*,W'}$ and also $g^{(i)} \geq 0$. $(c)$ comes from the definition in Equation (6). $\qquad\square$

### F.6   Possible Action Set in PARL and Comparison

#### F.6.1   Certification

*Proof of Theorem 4.* According to the definition of possible action set, we only need to prove the contrary: for any $a \in \mathcal{A} \setminus A^T(K)$, within poisoning size $K$, the poisoned policy cannot choose $a$: $\widetilde{\pi}_{\mathrm{P}}(s_t) \neq a$.

According to Equation (8), any $a \in \mathcal{A} \setminus A^T(K)$ satisfies
$$\sum_{a' \in \mathcal{A}} \max\{n_{a'}(s_t) - n_a(s_t) - K + \mathbf{1}[a' < a], 0\} > K. \tag{28}$$
Given poisoning size $K$, since each poisoning size can affect only one subpolicy, we know
$$\widetilde{n}_a(s_t) \leq n_a(s_t) + K$$
where $\widetilde{n}_a$ denotes to the poisoned aggregated action count. We suppose the attack could be successful, then for $a' < a$, $\widetilde{n}_{a'}(s_t) \leq \widetilde{n}_a(s_t) - 1$, and thus $n_{a'}(s_t) - \widetilde{n}_{a'}(s_t) \geq n_{a'}(s_t) - n_a(s_t) - K + 1$. Similarly, for $a' > a$, $\widetilde{a'}(s_t) \leq \widetilde{n}_a(s_t)$, and thus $n_{a'}(s_t) - \widetilde{n}_{a'}(s_t) \geq n_{a'}(s_t) - n_a(s_t) - K$. Also, for any $a' \neq a$ after poisoning $n_{a'}(s_t) - \widetilde{n}_{a'}(s_t) \geq 0$; otherwise deviating the difference subpolicies' decisions' from $a'$ to $a$ is strictly no-worse. Given these facts, the amount of votes that need to be reduced is the LHS of Equation (28) which is larger than $K$. However, we only have $K$ poisoning size, *i.e.*, $K$ votes that can be reduced. As a result, our assumption that the attack could be successful is falsified and the poisoned policy cannot choose $a$. $\qquad\square$

**Corollary 12** (Loose PARL Action Set)**.** *Under the condition of Definition 5, suppose the aggregation protocol is PARL as defined in Definition 7, then the* possible action set *at step $t$*
$$A^L(K) = \left\{ a \in \mathcal{A} \;\middle|\; \max_{a' \in \mathcal{A}} n_{a'}(s_t) - n_a(s_t) \leq 2K - \mathbf{1}[a > \arg\max_{a' \in \mathcal{A}} n_{a'}(s_t)] \right\}. \tag{29}$$

*Proof of Corollary 12.* Again, we prove the contrary, for any $a \in \mathcal{A} \setminus A^L(K)$, within poisoning size $K$, the poisoned policy cannot choose $a$: $\widetilde{\pi}_{\mathrm{P}}(s_t) \neq a$.

According to Equation (29), let $a_m = \arg\max_{a' \in \mathcal{A}} n_{a'}(s_t)$, then any $a \in \mathcal{A} \setminus A^L(K)$ satisfies
$$n_{a_m}(s_t) - n_a(s_t) > 2K - \mathbf{1}[a > a_m].$$
After poisoning, we thus have
$$n_{a_m}(s_t) - n_a(s_t) > -\mathbf{1}[a > a_m] \implies n_{a_m}(s_t) - n_a(s_t) \geq 1 - \mathbf{1}[a > a_m]$$
From the definition, $a \notin A^L(K)$ so $a \neq a_m$. If $a_m < a$, $n_{a_m}(s_t) \geq n_a(s_t)$; if $a_m > a$, $n_{a_m}(s_t) > n_a(s_t)$. In both cases, $a_m$ has higher priority to be chosen than $a$, and thus the poisoned policy cannot choose $a$. $\qquad\square$

#### F.6.2   Comparison

**Theorem 13.** *Under the condition of Definition 5, suppose the aggregation protocol is PARL as defined in Definition 7, $A^T(K)$, $A^L(K)$ are defined according to Equations* (8) *and* (29) *accordingly, then*

*1. $A^T(K) \subseteq A^L(K)$; and there are subpolicies $\{\pi_i\}_{i=0}^{u-1}$ and state $s_t$ such that $A^T(K) \subsetneq A^L(K)$.*

2. *Given subpolicies $\{\pi_i\}_{i=0}^{u-1}$ and state $s_t$, for any $a \in A^T(K)$, there exists a poisoned training set $\widetilde{D}$ whose poisoning size $|D \ominus \widetilde{D}| \leq K$ and some RL training mechanism, such that $\widetilde{\pi_P}(s_t) = a$ where $\widetilde{\pi_P}$ is the poisoned PARL policy trained on $\widetilde{D}$.*

*Proof of Theorem 13.* We prove the two arguments separately.

1. We first prove $A^T(K) \subseteq A^L(K)$. For any $a \in A^T(K)$, let $a_m = \arg\max_{a' \in \mathcal{A}} n_{a'}(s_t)$, then

$$\sum_{a' \in \mathcal{A}} \max\{n_{a'}(s_t) - n_a(s_t) - K + \mathbf{1}[a' < a], 0\} \leq K$$

$$\Longrightarrow n_{a_m}(s_t) - n_a(s_t) - K + \mathbf{1}[a_m < a] \leq K$$

$$\Longrightarrow n_{a_m}(s_t) - n_a(s_t) \leq 2K - \mathbf{1}[a > a_m]$$

where the last proposition is exactly the set selector of $A^L(K)$ so $a \in A^L(K)$.

We then prove that $A^T(K) \subsetneq A^L(K)$ can happen by construction. Suppose that there are three actions in the action space: $\mathcal{A} = \{a_1, a_2, a_3\}$. We construct subpolicies for current state $s_t$ such that the aggregated action counts are

$$n_{a_1}(s_t) = 10, n_{a_2}(s_t) = 9, n_{a_3}(s_t) = 1.$$

Given poisoning size $K = 5$, we find that

$$n_{a_1}(s_t) - n_{a_3}(s_t) = 9 \leq 10 - \mathbf{1}[a_3 \geq a_1] = 9 \qquad \Longrightarrow a_3 \in a^L(K),$$

$$(n_{a_1}(s_t) - n_{a_3}(s_t) - K + \mathbf{1}[a_1 < a_3]) +$$

$$(n_{a_2}(s_t) - n_{a_3}(s_t) - K + \mathbf{1}[a_2 < a_3]) = 9 > K = 5 \quad \Longrightarrow a_3 \notin a^T(K).$$

Therefore, $A^T(K) \subsetneq A^L(K)$ for these subpolicies and state $s_t$.

2. We prove by construction. For any $a \in A^T(K)$, we construct the set of subpolicies to poison $B^a \subseteq [u]$ such that $|B^a| \leq K$, then describe the corresponding poisoned dataset $\widetilde{D}^a$, and finally prove that the poisoned policy $\widetilde{\pi_P}^a(s_t) = a$.

For $a \in A^T(K)$, by definition (Equation (8)), we know

$$\sum_{a' \in \mathcal{A}} \max\{\underbrace{n_{a'}(s_t) - n_a(s_t) - K + \mathbf{1}[a' < a]}_{:=t_{a,a'}}, 0\} \leq K. \tag{30}$$

We now define a set of actions $C^a \subseteq \mathcal{A}$ such that

$$C^a = \{a' \in \mathcal{A} \mid t_{a,a'} > 0\}.$$

According to this definition, $a \notin C^a$ since $t_{a,a} \leq 0$.

**Fact F.1.** *For $a \in A^T(K)$ and any $a' \in \mathcal{A}$, $n_a(s_t) + K - \mathbf{1}[a' < a] \geq 0$.*

*Proof of Fact F.1.* Suppose $n_a(s_t) + K - \mathbf{1}[a' < a] \leq 0$, then $n_a(s_t) = K = 0$ and $a' < a$. Then

$$\sum_{a' \in \mathcal{A}} \max\{n_{a'}(s_t) - n_a(s_t) - K + \mathbf{1}[a' < a], 0\} \geq \sum_{a' \in \mathcal{A}} n_{a'}(s_t) - n_a(s_t) - K = \sum_{a' \in \mathcal{A}} n_{a'}(s_t) = u > 0$$

which contradicts the requirement that the LHS of the above inequality should be $\leq K = 0$. □

Give Fact F.1, for any $a' \in C^a$, $n_{a'}(s_t) \geq t_{a,a'}$. Notice that $n_{a'}(s_t)$ is the number of subpolicies that vote for action $a'$ at state $s_t$. Therefore, we can pick an arbitrary subset of those subpolicies whose cardinality is $t_{a,a'}$. We denote $B_{a'}^a$ to such subset:

$$B_{a'}^a \subseteq \{i \in [u] \mid \pi_i(s_t) = a'\}, |B_{a'}^a| = t_{a,a'}.$$

Now define $B_\alpha^a$: $B_\alpha^a = \bigcup_{a' \in C^a} B_{a'}^a$. We construct $B_\beta^a$ to be an arbitrary subset of those subpolicies whose prediction is not $a$ and who are not in $B_\alpha^a$, and limit the $B_\beta^a$'s cardinality:

$$B_\beta^a \subseteq \{i \in [u] \mid \pi_i(s_t) \neq a\} \setminus B_\alpha^a, |B_\beta^a| = \min\{K - |B_\alpha^a|, u - n_a(s_t) - |B_\alpha^a|\}.$$

Such $B_\beta^a$ can be selected, because:

- From definition, $B_\alpha^a \subseteq \{i \in [u] \mid \pi_i(s_t) \neq a\}$, where the cardinality of $\{i \in [u] \mid \pi_i(s_t) \neq a\}$ is $u - n_a(s_t)$. So $0 \leq u - n_a(s_t) - |B_\alpha^a|$.
- Since

$$|B_\alpha^a| \leq \sum_{a' \in C^a} |B_{a'}^a| = \sum_{a' \in C^a} t_{a,a'} = \sum_{a' \in \mathcal{A}, t_{a,a'} > 0} t_{a,a'} \overset{(*)}{\leq} K,$$

$K - |B_\alpha^a| \geq 0$, and thus $|B_\beta^a| \geq 0$. Here $(*)$ is due to Equation (30).

- The superset $\{i \in [u] \mid \pi_i(s_t) \neq a\} \setminus B_\alpha^a$ has cardinality $u - n_a(s_t) - |B_\alpha^a|$ and $|B_\beta^a| \leq u - n_a(s_t) - |B_\alpha^a|$.

To this point, we can define the set of subpolicies to poison:
$$B^a := B_\alpha^a \cup B_\beta^a,$$
and $|B^a| = |B_\alpha^a| + |B_\beta^a| \leq K$.

In a similar fashion as the attack construction in proof of Proposition 9, for each $i \in B^a$, we locate its corresponding partitioned dataset $D_i$ for training subpolicy $\pi_i^a$. We inset one trajectory $p_i^a$ to $D_i$ such that our chosen learning algorithm $\mathcal{M}_0$ can train a subpolicy $\tilde{\pi}_i^a = \mathcal{M}_0(D_i \cup \{p_i^a\})$ such that $\tilde{\pi}_i^a(s_t) = a$. For example, the trajectory could be $p_i^a = \{(s_t, a, s', \infty)\}$ where $s'$ is an arbitrary state that guarantees $p_i^a$ is hashed to partition $i$; and $\mathcal{M}_0$ learns the action with maximum reward for the memorized nearest state. Then, the poisoned dataset
$$\widetilde{D}^a = \left( \bigcup_{i \in B^a} D_i \cup \{p_i^a\} \right) \cup \left( \bigcup_{i \in [u] \setminus B'} D_i \right).$$

Thus, $\widetilde{D}^a$ is $|D \ominus \widetilde{D}^a| = |B^a| \leq K$, *i.e.*, the constructed attack's poisoning size is within $K$.

We now analyze the action prediction of the poisoned policy $\widetilde{\pi_P}$. For action $a$, after poisoning, $\tilde{n}_a(s_t) = n_a(s_t) + |B^a| = n_a(s_t) + \min\{K, u - n_a(s_t)\} = \min\{n_a(s_t) + K, u\}$. If $\tilde{n}_a(s_t) = u$, then all subpolicies vote for $a$, apparently $\widetilde{\pi_P}^a(s_t) = a$; otherwise, $\tilde{n}_a(s_t) = n_a(s_t) + K$. In this case, for any action $a' \in C^a$, since we at least choose subpolicies in $B_{a'}^a$, and change their action prediction to $a$, the aggregated action count after poisoning is $\tilde{n}_{a'}(s_t) \leq n_{a'}(s_t) - |B_{a'}^a| = n_{a'}(s_t) - t_{a,a'} = n_{a'}(s_t) - n_{a'}(s_t) + n_a(s_t) + K - \mathbf{1}[a' < a] = n_a(s_t) + K - \mathbf{1}[a' < a] = \tilde{n}_a(s_t) - \mathbf{1}[a' < a]$. Thus, $a'$ has lower priority to be chosen than $a$. For any action $a' \notin C^a$ and $a' \neq a$, by the definition of $C^a$, $t_{a,a'} = n_{a'}(s_t) - n_a(s_t) - K + \mathbf{1}[a' < a] \leq 0$. After poisoning, the vote of $a'$ does not increase, *i.e.*, $\tilde{n}_{a'}(s_t) \leq n_{a'}(s_t) \leq n_a(s_t) + K - \mathbf{1}[a' < a] = \tilde{n}_a(s_t) - \mathbf{1}[a' < a]$. Thus, $a'$ also has lower priority to be chosen than $a$. In conclusion, we have $\widetilde{\pi_P}^a(s_t) = a$.

To this point, for any $a \in A^T(K)$, we successfully construct the corresponding poisoned dataset $\widetilde{D}^a$ within poisoning size $K$ such that $\widetilde{\pi_P}^a(s_t) = a$, thus concludes the proof.

$\square$

### F.7 POSSIBLE ACTION SET IN TPARL

*Proof of Theorem 5.* For ease of notation, we let $w = \min\{W, t + 1\}$, so $w$ is the actual window size used at step $t$. We let $t_0 = t - w + 1$, *i.e.*, $t_0$ is the actual start time step for TPARL aggregation at our current step $t$. Now we can write chosen action at step $t$ without poisoning as $\pi_T(s_{t_0:t})$.

We only need to prove the contrary: for any $a \in \mathcal{A} \setminus A(K)$, within poisoning size $K$, the poisoned policy cannot choose $a$: $\widetilde{\pi_T}(s_{t_0:t}) \neq a$.

According to Equation (9), for such $a$, there exists $a' \neq a$ such that
$$\sum_{i=1}^{K} h_{a',a}^{(i)} \leq \delta_{a',a} = n_{a'}(s_{t_0:t}) - n_a(s_{t_0:t}) - \mathbf{1}[a < a']. \tag{31}$$
Suppose there exists such poisoning attack that lets $\widetilde{\pi_T}(s_{t_0:t}) = a$. This implies that for $a'$, after poisoning, we have
$$\tilde{n}_{a'}(s_{t_0:t}) - \tilde{n}_{a'}(s_{t_0:t}) - \mathbf{1}[a' < a] < 0, \tag{32}$$
where $\tilde{n}_a$ is the aggregated action count after poisoning. Since the poisoning size is within $K$, it can affect at most $K$ subpolicies. We let $B \subseteq [u], |B| \leq K$ to represent the affected subpolicy set. Therefore,
$$\tilde{n}_{a'}(s_{t_0:t}) - \tilde{n}_a(s_{t_0:t}) - \mathbf{1}[a' < a]$$
$$= \underbrace{n_{a'}(s_{t_0:t}) - n_a(s_{t_0:t}) - \mathbf{1}[a' < a]}_{\delta_{a',a}} + (\tilde{n}_{a'}(s_{t_0:t}) - n_{a'}(s_{t_0:t})) - (\tilde{n}_a(s_{t_0:t}) - n_a(s_{t_0:t}))$$

$$=\delta_{a',a} + \sum_{i\in B}\left(\sum_{j=0}^{w-1}\mathbf{1}[\tilde{\pi}_i(s_{t-j})=a',\pi_i(s_{t-j})\neq a'] - \sum_{j=0}^{w-1}\mathbf{1}[\tilde{\pi}_i(s_{t-j})\neq a',\pi_i(s_{t-j})=a']\right)$$

$$-\sum_{i\in B}\left(\sum_{j=0}^{w-1}\mathbf{1}[\tilde{\pi}_i(s_{t-j})=a,\pi_i(s_{t-j})\neq a] - \sum_{j=0}^{w-1}\mathbf{1}[\tilde{\pi}_i(s_{t-j})\neq a,\pi_i(s_{t-j})=a]\right)$$

$$\geq\delta_{a',a} - \sum_{i\in B}\left(\sum_{j=0}^{w-1}\mathbf{1}[\tilde{\pi}_i(s_{t-j})\neq a',\pi_i(s_{t-j})=a'] + \sum_{j=0}^{w-1}\mathbf{1}[\tilde{\pi}_i(s_{t-j})=a,\pi_i(s_{t-j})\neq a]\right)$$

$$\geq\delta_{a',a} - \sum_{i\in B}\left(\sum_{j=0}^{w-1}\mathbf{1}[\pi_i(s_{t-j})=a'] + \sum_{j=0}^{w-1}\mathbf{1}[\pi_i(s_{t-j})\neq a]\right)$$

$$=\delta_{a',a} - \sum_{i\in B}\left(\sum_{j=0}^{w-1}\mathbf{1}_{i,a'}(s_{t-j}) + w - \sum_{j=0}^{w-1}\mathbf{1}_{i,a}(s_{t-j})\right)$$

$$=\delta_{a',a} - \sum_{i\in B}h_{i,a',a} \overset{(a)}{\geq} \delta_{a',a} - \sum_{i=1}^{|B|}h_{a',a}^{(i)} \overset{(b)}{\geq} \delta_{a',a} - \sum_{i=1}^{K}h_{a',a}^{(i)} \overset{(c)}{\geq} 0.$$

This contradicts with Equation (32), and thus the assumption is falsified, *i.e.*, there is no such poisoning attack that let $\widetilde{\pi_{\mathrm{T}}}^{(i)}(s_{t_0:t}) = a$. In the above equations, $(a)$ is due to the fact that $h_{a',a}^{(i)}$ is a nonincreasing permutation of $\{h_{i,a',a}\}_{i=0}^{u-1}$. $(b)$ is due to the facts that $|B| \leq K$ and $h_{a',a}^{(i)} \geq 0$. $(c)$ comes from Equation (31). $\qquad\square$

### F.8 Hardness for Computing Tight Possible Action Set in TPARL

*Proof of Theorem 6.* By Definition 5, the possible action set with minimum cardinality (called minimal possible action set hereinafter) is unique. Otherwise, suppose $A$ and $B$ are both minimal possible action set, but $A \neq B$, then $A \cap B$ is a smaller and valid possible action set. Therefore, the oracle that returns the minimal possible action set, denoted by MINSET, can tell whether any action $a \in$ MINSET and thus whether any action $a$ can be chosen by some poisoned policy $\tilde{\pi}$ whose poisoning size is within $K$. In other words, the problem of determining whether an action $a$ can be chosen by some poisoned policy $\tilde{\pi}$ whose poisoning size is within $K$, denoted by ATTKACT, is polynomially equivalent to MINSET: MINSET $\equiv_P$ ATTKACT. Now, we show a polynomial reduction from the set cover problem to ATTKACT, which implies that our MINSET problem is an NP-complete problem.

The decision version of the set cover problem (Karp, 1972), denoted by SETCOVER, is a well-known NP-complete problem and is defined as follows. The inputs are

1. a universal set of $n$ elements: $\mathcal{U} = \{u_1, u_2, \cdots, u_n\}$;

2. a set of subsets of $\mathcal{U}$: $\mathcal{V} = \{V_1, \cdots, V_m\}, V_i \subseteq \mathcal{U}, 1 \leq i \leq m, \bigcup_{i=1}^{m} V_i = \mathcal{U}$;

3. a positive number $K \in \mathbb{R}_+$.

The output is a boolean variable $b$, indicating that whether there exists a subset $\mathcal{W} \subseteq \mathcal{V}, |\mathcal{W}| \leq K$, such that $\forall u_i \in \mathcal{U}, \exists V \in \mathcal{W}, u_i \in V$. Given an oracle to ATTKACT, we need to show SETCOVER can be solved in polynomial time, *i.e.*, SETCOVER $\leq_P$ ATTKACT.

- If $K \geq n$:
  We scan all sets $V_i \in \mathcal{V}$. To being with, we have a record set $S \leftarrow \emptyset$, and an answer set $\mathcal{W} \leftarrow \emptyset$. Whenever we encounter a set that contains a new element $u_i \notin S$, we put $\mathcal{W} \leftarrow \mathcal{W} \cup \{V_j\}$, and record this element $S \leftarrow S \cup \{u_i\}$. After one scan pass, $\mathcal{W}$ covers all elements of $\mathcal{U}$ (since $\bigcup_{j=1}^{m} V_j = \mathcal{U}$), and $|\mathcal{W}| \leq n \leq K$. Therefore, $\mathcal{W}$ is a valid set cover. Since we can always find such $\mathcal{W}$, we can directly answer true.

- If $K \geq m$:
  We can directly return $\mathcal{V}$ as a valid set cover, since $\bigcup_{j=1}^{m} V_j = \mathcal{U}$ and $|\mathcal{V}| \leq m \leq K$. Thus, we can answer `true`.

- If $K < \min\{n, m\}$:
  This is the general case which we need to handle. Now we construct the $\mathsf{ATTKACT}(K)$ problem so that we can trigger its oracle to solve $\mathsf{SETCOVER}$.

  1. The poisoning size is $K$.
  2. The action space $\mathcal{A} = \mathcal{U} \cup \{b\} \cup \Gamma$ where $\Gamma := \{\#_1, \ldots, \#_{m^2 n}\}$. ($|\mathcal{A}| = n + 1 + m^2 n$.) The sorting of actions is $u_1 < u_2 < \cdots < u_n < b < \#_1 < \cdots < \#_{m^2 n}$.
  3. The subpolicies are $\{\pi_j^a\}_{j=1}^{m} \cup \{\pi_i^b, \pi_{i,j}^c \mid 1 \leq i \leq n, 1 \leq j \leq K - 1\}$, where $\pi_j^a$ corresponds to $V_j \in \mathcal{V}$, and $\pi_i^b, \pi_{i,j}^c$ correspond to $u_i \in \mathcal{U}$. (Number of subpolicies $u = m + Kn \leq m + n^2$.)
  4. The current time step is $t = nm$, and the window size $W = nm$.
  5. The input action is $b$, *i.e.*, asking whether $b$ can be chosen by some poisoned TPARL policy, *i.e.*, $\widetilde{\pi_{\mathrm{T}}}(s_{1:t}) = b$, if the poisoning size is within $K$.

  Now, we construct the states at each step $t$ ($1 \leq t \leq nm$) so that the subpolicies' action predictions at these steps are as follows.

  1. Count the appearing time of each $u_i$ in $\mathcal{V}$, and denote it by $c_i$: $c_i = \sum_{j=1}^{m} \mathbf{1}[u_i \in V_j]$. For each $u_i$, select a $V_{j_0}$ that contains $u_i$, and in the corresponding $\pi_{j_0}^a$, assign $m - c_i + 1$ steps to predict $u_i$; for all other $V_j$ that contains $u_i$, in the corresponding $\pi_j^a$, assign one step to predict $u_i$.
  2. After this process, each $\pi_j^a$ at least has one time step whose action prediction is $u_i$ for each $u_i \in V_j$. Among all $\{\pi_j^a\}_{j=1}^{m}$ and all time steps $1 \leq t \leq nm$, $mn$ step-action cells are filled, and the remaining $(m^2 n - mn)$ cells are filled by $\#_l \in \Gamma$ sequentially.
  3. For each $\pi_i^b$, arbitrarily select $W - m$ time steps to assign action prediction as $u_i$; and fill in other $m$ time steps by remaining $\#_l \in \Gamma$ sequentially.
  4. For each $\pi_{i,j}^c$, for all time steps, let the action prediction be $u_i$.

  As we can observe, the number of actions, the number of subpolicies, and the window size are all bounded by a polynomial of $n$ and $m$. Therefore, such construction can be done in polynomial time.

  We then show $\mathsf{SETCOVER} = \text{true} \iff \exists K', b \in \mathsf{ATTKACT}(K'), 1 \leq K' \leq K$.

  - $\Longrightarrow$:
    Suppose the covering set is $\mathcal{W} \subseteq \mathcal{V}$, we denote $K'$ to $|\mathcal{W}|$, and construct the $\mathsf{ATTKACT}$ problem with poisoning size $K'$ as described above.
    We can construct a poisoning strategy to let $\widetilde{\pi_{\mathrm{T}}}(s_{1:nm}) = b$, The poisoning strategy is to find out $\pi_j^a$ for each $V_j \in \mathcal{W}$, and to let them predict action $b$ throughout all time steps: $\widetilde{\pi}_j^a(s_t') = b, 1 \leq t' \leq nm$.
    After poisoning, the aggregated action count $\widetilde{n}_b(s_{1:nm}) = |\mathcal{W}| \times nm = K'nm$. Since $\mathcal{W}$ covers every $u_i \in \mathcal{U}$, for each $u_i \in \mathcal{U}$ there exists a set $V_j \ni u_i$, whose corresponding $\widetilde{\pi}_j^a$ is poisoned to predict $b$. Thus, $\widetilde{n}_{u_i}(s_{1:nm}) < n_{u_i}(s_{1:nm}) = m + (W - m) + W \times (K' - 1) = WK' = K'nm$. For any $\#_l \in \Gamma$, $\widetilde{n}_{\#_l}(s_{1:nm}) \leq n_{\#_l}(s_{1:nm}) = 1$. In summary,
    $$\widetilde{n}_{u_i}(s_{1:nm}) < K'nm, \widetilde{n}_b(s_{1:nm}) = K'nm, \widetilde{n}_{\#_l}(s_{1:nm}) = 1.$$
    Thus, after TPARL aggregation, the poisoned policy $\widetilde{\pi_{\mathrm{T}}}(s_{1:nm}) = b$, and therefore $b \in \mathsf{ATTKACT}(K')$.

  - $\Longleftarrow$:
    Suppose it is $K'$ that let $b \in \mathsf{ATTKACT}(K')$, which implies that there exists such a poisoning attack within size $K'$ that misleads the poisoned policy to $b$: $\widetilde{\pi_{\mathrm{T}}}(s_{1:nm}) = b$. Since the poisoning size is $K'$, after poisoning the aggregated action count
    $$\widetilde{n}_b(s_{1:nm}) \leq K'W = K'nm.$$
    For each $u_i \in \mathcal{U}$, since $n_{u_i}(s_{1:nm}) = m + (W - m) + W \times (K' - 1) = K'nm$, we always have
    $$\widetilde{n}_{u_i}(s_{1:nm}) \overset{(*)}{<} \widetilde{n}_b(s_{1:nm}) \leq n_{u_i}(s_{1:nm}), \tag{33}$$

where $(*)$ is due to the condition of successful attack. We denote the set of poisoned subpolicies by $\Pi$ ($|\Pi| \leq K'$). Therefore, Equation (33) implies that for each $u_i \in \mathcal{U}$, there exists at least one subpolicy

$$\pi'_{u_i} \in \Pi, \pi'_{u_i} \in \{\pi_j^a \mid u_i \in V_j\} \cup \{\pi_i^b\} \cup \{\pi_{i,j}^c \mid 1 \leq j \leq K-1\} \qquad (34)$$

that is poisoned by the attack, otherwise the aggregated vote $\widetilde{n}_{u_i}$ cannot change.

We partition $\Gamma$ by $\Gamma^a$ and $\Gamma^{bc}$, where

$$\Gamma^a = \Gamma \cap \{\pi_j^a\}_{j=1}^m, \Gamma^b = \Gamma \cap \{\pi_i^b, \pi_{i,j}^c \mid 1 \leq i \leq n, 1 \leq j \leq K-1\}.$$

We construct additional poisoning set $\Gamma_+^a$ following this process: In the beginning, $\Gamma_+^a \leftarrow \emptyset$. For each $u_i \in \mathcal{U}$, if $\Gamma^a \cap \{\pi_j^a \mid u_i \in V_j\}$ is not empty, skip. Otherwise, according to Equation (34), $\Gamma^b \cap \{\pi_i^b, \pi_{i,j}^c \mid 1 \leq j \leq K-1\}$ is not empty. In this case, we find an arbitrary covering set of $u_i$, namely $V_{j_0} \ni u_i$, and put $\pi_{j_0}^a$ into $\Gamma_+^a$. When the process terminates, we find that for each $u_i \in \mathcal{U}$,

$$(\Gamma^a \cup \Gamma_+^a) \cap \{\pi_j^a \mid u_i \in V_j\} \neq \emptyset. \qquad (35)$$

Following the mapping $\{\pi_j^a\}_{j=1}^m \longleftrightarrow \{V_j \mid 1 \leq j \leq m\} = \mathcal{V}$, the subset $(\Gamma^a \cup \Gamma_+^a) \subseteq \{\pi_j^a\}_{j=1}^m$ can be mapped to $(\mathcal{W}^a \cup \mathcal{W}_+^a) \subseteq \mathcal{V}$. From Equation (35), $(\mathcal{W}^a \cup \mathcal{W}_+^a)$ is a valid set cover for $\mathcal{U}$. We now study the cardinality of this set cover. From the process, we know that every $V_{j_0} \in \mathcal{W}_+^a$ corresponds to a different set in $\Gamma^b$. Thus, $|\mathcal{W}_+^a| \leq |\Gamma^b|$, which implies that

$$|\mathcal{W}^a \cup \mathcal{W}_+^a| \leq |\mathcal{W}^a| + |\mathcal{W}_+^a| \leq |\Gamma^a| + |\Gamma^b| = |\Gamma| \leq K'.$$

To this point, we successfully construct a set cover within cardinally $K' \leq K$ that covers $\mathcal{U}$, so SETCOVER $=$ true.

Therefore, we can check whether SETCOVER $=$ true by iterating $K'$ from 1 to $K$ ($<$ $\min\{n, m\}$ iterations), constructing the ATTKACT($K'$) problem, and querying the oracle. The whole process can be done in polynomial time assuming $O(1)$ computation time of the ATTKACT($K'$) oracle.

To this point, we have shown SETCOVER $\leq_P$ ATTKACT. On the other hand, an undeterminisitic Turing machine can try different poisoning strategies by branching on whether to poison current subpolicy and what actions to be assigned to each poisoned subpolicy. The decision of whether the poisoning is successful can be done in polynomial time and $b \in$ ATTKACT($K$) corresponds to the existence of successfully attacked branches. Thus, ATTKACT $\in$ NP. Given that SETCOVER is an NP-complete problem, so does ATTKACT and MINSET (since MINSET $\equiv_P$ ATTKACT). $\qquad \square$

### F.9   POSSIBLE ACTION SET IN DPARL

*Proof of Theorem 7.* For ease of notation, we assume that $W_{\max} \leq t+1$, and otherwise we let $W_{\max} \leftarrow t+1$. We let $t_0 = \max\{t - W_{\max} + 1, 0\}$ be the start time step of the maximum possible window.

We only need to prove the contrary: for any $a \in \mathcal{A} \setminus A(K)$, within poisoning size $K$, the poisoned policy cannot choose $a$: $\widetilde{\pi}_D(s_{t_0:t}) \neq a$. We prove by contradiction: we assume that there exists such a poisoning attack within poisoning size $K$ that lets $\widetilde{\pi}_D(s_{t_0:t}) = a$. From the expression of $A(K)$ (Equation (10)), $a \neq a_t = \pi_D(s_{t_0:t})$. Suppose the selected time window before the attack is $W'$ (selected according to Equation (1) based on $\Delta_t^W$ and $n_a$), and the selected time window after the attack is $\widetilde{W}'$ (selected according to Equation (1) based on $\widetilde{\Delta}_t^W$ and $\widetilde{n}_a$).

- If $\widetilde{W}' = W'$:
  Suppose we use the TPARL aggregation policy with window size $W = W'$ instead of current DPARL aggregation policy, then we will have $\pi_T(s_{t-W'+1:t}) = a_t$ and $\widetilde{\pi}_T(s_{t-W'+1:t}) = a$. Thus, according to the definition of possible action set (Definition 5), $a \in A(K)$ where $A(K)$ is defined by Equation (9) in Theorem 5. This implies that $a \in A(K)$ where $A(K)$ is defined by Equation (10), which contradicts the assumption that $a \in \mathcal{A} \setminus A(K)$.

- If $\widetilde{W}' \neq W'$:
  According to the definition in Equation (10),

$$\min_{1 \leq W^* \leq W_{\max}, W^* \neq W', a'' \neq a_t} L_{a,a''}^{W^*, W'} > K.$$

We define $a_t^\# = \arg\max_{a_0 \neq a_t} \widetilde{n}_{a_0}(s_{t-W'+1:t})$. Then, the above equation implies that

$$L_{a,a_t^\#}^{\widetilde{W}',W'} > K$$

and thus

$$\sum_{i=1}^{L_{a,a_t^\#}^{\widetilde{W}',W'}} g^{(i)} + W'(n_a^{\widetilde{W}'} - n_{a^\#}^{\widetilde{W}'}) - \widetilde{W}'(n_{a_t}^{W'} - n_{a_t^\#}^{W'}) - \mathbf{1}[a > a_t] < 0.$$

Since $g^{(i)} \geq 0$ by definition (Equation (7)),

$$\sum_{i=1}^{K} g^{(i)} + W'(n_a^{\widetilde{W}'} - n_{a^\#}^{\widetilde{W}'}) - \widetilde{W}'(n_{a_t}^{W'} - n_{a_t^\#}^{W'}) - \mathbf{1}[a > a_t] < 0, \tag{36}$$

where $a^\# = \arg\max_{a_0 \neq a', a_0 \in \mathcal{A}} n_{a_0}(s_{t-\widetilde{W}'+1:t})$ and $n_a^w$ is a shorthand of $n_a(s_{t-w+1:t})$.

Following the derivation from Equation (27) to $(a)$, we have

$$\sum_{i=1}^{K} g^{(i)} + W'(n_a^{\widetilde{W}'} - n_{a^\#}^{\widetilde{W}'}) - \widetilde{W}'(n_{a_t}^{W'} - n_{a_t^\#}^{W'}) - \mathbf{1}[a > a_t] \geq W'(\widetilde{n}_a^{\widetilde{W}'} - \widetilde{n}_{a^\#}^{\widetilde{W}'}) - \widetilde{W}'(\widetilde{n}_{a_t}^{W'} - \widetilde{n}_{a_t^\#}^{W'}) - \mathbf{1}[a > a_t].$$

Combined with Equation (36),

$$W'(\widetilde{n}_a^{\widetilde{W}'} - \widetilde{n}_{a^\#}^{\widetilde{W}'}) - \widetilde{W}'(\widetilde{n}_{a_t}^{W'} - \widetilde{n}_{a_t^\#}^{W'}) - \mathbf{1}[a > a_t] < 0. \tag{37}$$

On the other hand, the successful attack assumption, *i.e.*, $\widetilde{\pi_{\mathrm{D}}}(s_{t_0:t}) = a$ and $\pi_{\mathrm{D}}(s_{t_0:t}) = a_t$, implies that

$$\widetilde{\Delta}_t^{\widetilde{W}'} = \frac{\widetilde{n}_a^{\widetilde{W}'} - \widetilde{n}_{a^{(2)}}^{\widetilde{W}'}}{\widetilde{W}'} > \frac{\widetilde{n}_{a_{W'}}^{W'} - \widetilde{n}_{a_{W'}^{(2)}}^{W'}}{W'} = \widetilde{\Delta}_t^{W'}, \tag{38}$$

where ">" is "$\geq$" if $a < a_t$. In the above equation,

$$a^{(2)} = \arg\max_{a_0 \neq a, a_0 \in \mathcal{A}} \widetilde{n}_{a_0}^{\widetilde{W}'}, a_{W'} = \arg\max_{a_0 \in \mathcal{A}} \widetilde{n}_{a_0}^{W'}, a_{W'}^{(2)} = \arg\max_{a_0 \neq a_{W'}, a_0 \in \mathcal{A}} \widetilde{n}_{a_0}^{W'}.$$

Intuitively, after poisoning, $a^{(2)}$ is the runner-up action at window $\widetilde{W}'$, $a_{W'}$ is the action at window $W'$, and $a_{W'}^{(2)}$ is the runner-up action at window $W'$. We rewrite Equation (38) to

$$W'(\widetilde{n}_a^{\widetilde{W}'} - \widetilde{n}_{a^{(2)}}^{\widetilde{W}'}) \geq \widetilde{W}'(\widetilde{n}_{a_{W'}}^{W'} - \widetilde{n}_{a_{W'}^{(2)}}^{W'}) + \mathbf{1}[a > a_t]. \tag{39}$$

We have the following two observations:

1. $\widetilde{n}_a^{\widetilde{W}'} - \widetilde{n}_{a^\#}^{\widetilde{W}'} \geq \widetilde{n}_a^{\widetilde{W}'} - \widetilde{n}_{a^{(2)}}^{\widetilde{W}'}$, since $a$ is the top action and $a^{(2)}$ is the runner-up action and their margin should be the smallest.

2. $\widetilde{n}_{a_{W'}}^{W'} - \widetilde{n}_{a_{W'}^{(2)}}^{W'} \geq \widetilde{n}_{a_t}^{W'} - \widetilde{n}_{a_t^\#}^{W'}$, because 1) if $a_t = a_{W'}$, LHS equals to RHS; 2) if $a_t \neq a_{W'}$, LHS $\geq 0$ and RHS $\leq 0$.

Plugging these two observations to two sides of Equation (39), we get

$$W'(\widetilde{n}_a^{\widetilde{W}'} - \widetilde{n}_{a^\#}^{\widetilde{W}'}) \geq \widetilde{W}'(\widetilde{n}_{a_t}^{W'} - \widetilde{n}_{a_t^\#}^{W'}) + \mathbf{1}[a > a_t]. \tag{40}$$

This contradicts with Equation (37).

Since in both cases, we find contradictions. Now we can conclude that for any $a \in \mathcal{A} \setminus A(K)$, within poisoning size $K$, the poisoned policy cannot choose $a$: $\widetilde{\pi_{\mathrm{D}}}(s_{t_0:t}) \neq a$. $\qquad\square$

## G   ADDITIONAL EXPERIMENTAL DETAILS

### G.1   DETAILS OF THE OFFLINE RL ALGORITHMS AND IMPLEMENTATIONS

We experimented with three offline RL algorithms: DQN (Mnih et al., 2013), QR-DQN (Dabney et al., 2018), and C51 (Bellemare et al., 2017). The first one is the standard baseline, while the latter two are distributional RL algorithms which show SOTA results in offline RL tasks. We first briefly introduce the algorithm ideas, followed by the implementation details.

**Algorithm Ideas.** The core of Q-learning (Watkins & Dayan, 1992) is the Bellman optimality equation (Bellman, 1966)

$$Q^\star(s, a) = \mathbb{E}R(s, a) + \gamma \mathbb{E}_{s' \sim P} \max_{a' \in \mathcal{A}} Q^\star(s', a'), \tag{41}$$

where a parameterized $Q^\theta$ is adopted to approximate the optimal $Q^\star$ and iteratively improved. In DQN (Mnih et al., 2013) specifically, the parameterization is achieved by using a convolutional neural network (LeCun et al., 1998). In contrast to estimating the mean action value $Q^\pi(s, a)$ in DQN, distributional RL algorithms estimate a density over the values of the state-action pairs. The distributional Bellman optimality can be expressed as follows:

$$Z^\star(s, a) \stackrel{D}{=} r + \gamma Z^\star(s', \text{argmax}_{a' \in \mathcal{A}} Q^\star(s', a')) \text{ where } r \sim R(s, a), s' \sim P(\cdot \mid s, a). \tag{42}$$

Concretely, QR-DQN (Dabney et al., 2018) approximates the density $D^\star$ with a uniform mixture of $K$ Dirac delta functions, while C51 (Bellemare et al., 2017) approximates the density using a categorical distribution over a set of anchor points.

**Implementation Details.** For training the subpolicies using offline RL training algorithms, we use the code base of Agarwal et al. (2020). The configuration files containing the detailed hyperparameters can be found at their public repository `https://github.com/google-research/batch_rl/tree/master/batch_rl/fixed_replay/configs`. For the three methods DQN, QR-DQN, and C51, the names of the configuration files are `dqn.gin`, `quantile.gin`, and `c51.gin`, respectively. On each partition, we train the subpolicy for 50 epochs.

## G.2 Concrete Experimental Procedures

We conduct experiments on two Atari 2600 games, Freeway and Breakout from OpenAI Gym (Brockman et al., 2016), and one autonomous driving environment Highway (Leurent, 2018), following Section 4.

Concretely, in the **training** stage, we first partition the training dataset into $u$ partitions ($u = 30, 50,$ or $100$) by using the hash function $h(\tau)$ pre-defined in each environment. Let $s^i$ be the $i$-th value in the state representation and $d$ be the dimensionality of the state. In *Atari games* where the state is the game frame, $h(\tau)$ is defined as the sum of all pixel values of all frames in the trajectory $\tau$, *i.e.*, $h(\tau) = \sum_{s \in \tau} \sum_{i \in [d]} s^i$. In *Highway* where the state is a floating point number scalar containing the positions and velocities of all vehicles, we define $h(\tau) = \sum_{s \in \tau} \sum_{i \in [d]} f(s^i)$ where $f : \mathbb{R} \to \mathbb{Z}$ is a deterministic function that maps the given float value to integer space. Concretely, we take $f(x)$ as the the sum of the higher 16 bits and the lower 16 bits of its 32-bit representation under IEEE 754 standard (group of the Microprocessor Standards Subcommittee & Institute, 1985). We then train the subpolicies on the partitions with offline RL training algorithm $\mathcal{M}_0 \in \{$DQN (Mnih et al., 2013), QR-DQN (Dabney et al., 2018), C51 (Bellemare et al., 2017)$\}$. The detailed description of the algorithms can be found in Appendix G.1.

In the **aggregation** stage, we apply the three proposed aggregation protocols (PARL (Theorem 8), TPARL (Theorem 1), and DPARL (Theorem 3)) on the trained $u$ subpolicies and derive the aggregated policies $\pi_\text{P}$, $\pi_\text{T}$, and $\pi_\text{D}$ accordingly.

Finally, in the **certification** stage, for each aggregated policy, we provide the per-state action and cumulative reward certification following our theorems. When certifying the *per-state action*, we constrain the maximum trajectory length $H = 1000$ for Atari games and $H = 30$ for Highway (which is the full length in Highway) and report results averaged over 20 runs; for the *cumulative reward* certification, we adopt the trajectory length $H = 400$ for evaluating Freeway, $H = 75$ for Breakout, and $H = 30$ for Highway.

**Configuration of Trajectory Length $H$.** For *Atari games*, we do not evaluate the full episode length (up to tens of thousands steps), since our goal is to compare the relative certified robustness of different RL algorithms, and the evaluation on relatively short trajectory is sufficient under affordable computation cost. Moreover, different episodes in Atari games are oftentimes of different lengths; thus it is necessary that we restrict the episode length to enable a fair comparison. For *Highway*, we evaluate on the full episode ($H = 30$) where we can efficiently achieve effective comparisons.

Table 4: **Benign empirical performance** of three *aggregation protocols* (PARL, TPARL, and DPARL) applied on subpolicies trained using three *offline RL algorithms* (DQN, QR-DQN, and C51), with the number of sub-policies (*i.e.*, #partitions) $u$ equal to 30 or 50. We report results averaged over 20 runs of varying randomness in the environment.

| Freeway | | $u = 30$ | | | | | $u = 50$ | | | |
|---|---|---|---|---|---|---|---|---|---|---|
| | PARL | TPARL $(W=2)$ | TPARL $(W=3)$ | TPARL $(W=4)$ | DPARL $(W_{\max}=5)$ | PARL | TPARL $(W=2)$ | TPARL $(W=3)$ | TPARL $(W=4)$ | DPARL $(W_{\max}=5)$ |
| DQN | $10.90_{\pm 0.77}$ | $11.65_{\pm 0.57}$ | $11.25_{\pm 0.54}$ | $12.00_{\pm 0.45}$ | $11.45_{\pm 0.74}$ | $10.95_{\pm 0.97}$ | $11.60_{\pm 1.02}$ | $12.40_{\pm 0.58}$ | $11.65_{\pm 0.65}$ | $11.90_{\pm 0.89}$ |
| QR-DQN | $11.60_{\pm 0.66}$ | $11.85_{\pm 1.15}$ | $11.25_{\pm 0.62}$ | $11.80_{\pm 0.87}$ | $12.10_{\pm 0.99}$ | $11.50_{\pm 0.92}$ | $11.60_{\pm 1.02}$ | $12.15_{\pm 0.57}$ | $12.80_{\pm 0.51}$ | $11.90_{\pm 0.77}$ |
| C51 | $11.20_{\pm 0.60}$ | $12.45_{\pm 0.50}$ | $12.55_{\pm 0.50}$ | $11.40_{\pm 0.66}$ | $12.40_{\pm 0.49}$ | $11.70_{\pm 1.27}$ | $11.80_{\pm 0.75}$ | $12.70_{\pm 0.46}$ | $11.50_{\pm 0.87}$ | $11.95_{\pm 0.92}$ |

| Breakout | | $u = 30$ | | | | | $u = 50$ | | | |
|---|---|---|---|---|---|---|---|---|---|---|
| | PARL | TPARL $(W=2)$ | TPARL $(W=3)$ | TPARL $(W=4)$ | DPARL $(W_{\max}=5)$ | PARL | TPARL $(W=2)$ | TPARL $(W=3)$ | TPARL $(W=4)$ | DPARL $(W_{\max}=5)$ |
| DQN | $58.65_{\pm 40.83}$ | $38.05_{\pm 10.22}$ | $26.00_{\pm 12.79}$ | $13.50_{\pm 11.41}$ | $36.00_{\pm 13.39}$ | $60.25_{\pm 31.99}$ | $37.90_{\pm 11.55}$ | $25.90_{\pm 9.55}$ | $14.95_{\pm 7.24}$ | $45.00_{\pm 13.44}$ |
| QR-DQN | $76.50_{\pm 79.76}$ | $45.25_{\pm 42.70}$ | $22.05_{\pm 9.13}$ | $19.05_{\pm 6.91}$ | $42.05_{\pm 15.60}$ | $62.80_{\pm 28.71}$ | $32.10_{\pm 9.27}$ | $40.25_{\pm 52.97}$ | $17.30_{\pm 7.89}$ | $41.00_{\pm 13.87}$ |
| C51 | $51.80_{\pm 10.74}$ | $37.60_{\pm 11.38}$ | $24.75_{\pm 12.77}$ | $13.55_{\pm 7.37}$ | $34.75_{\pm 10.91}$ | $60.55_{\pm 20.44}$ | $34.85_{\pm 11.92}$ | $26.15_{\pm 9.98}$ | $19.65_{\pm 7.30}$ | $39.90_{\pm 14.69}$ |

| Highway | | $u = 30$ | | | | | $u = 50$ | | | |
|---|---|---|---|---|---|---|---|---|---|---|
| | PARL | TPARL $(W=2)$ | TPARL $(W=3)$ | TPARL $(W=4)$ | DPARL $(W_{\max}=5)$ | PARL | TPARL $(W=2)$ | TPARL $(W=3)$ | TPARL $(W=4)$ | DPARL $(W_{\max}=5)$ |
| DQN | $28.07_{\pm 3.86}$ | $19.16_{\pm 10.26}$ | $16.83_{\pm 8.35}$ | $12.69_{\pm 6.17}$ | $23.55_{\pm 9.01}$ | $29.05_{\pm 0.62}$ | $16.63_{\pm 9.93}$ | $15.85_{\pm 8.52}$ | $13.43_{\pm 8.29}$ | $22.51_{\pm 8.01}$ |
| QR-DQN | $28.52_{\pm 1.22}$ | $18.63_{\pm 8.07}$ | $15.53_{\pm 7.46}$ | $12.77_{\pm 6.39}$ | $23.11_{\pm 7.94}$ | $27.64_{\pm 3.01}$ | $20.59_{\pm 7.70}$ | $14.94_{\pm 7.76}$ | $14.13_{\pm 7.54}$ | $23.49_{\pm 7.95}$ |
| C51 | $28.86_{\pm 1.08}$ | $20.20_{\pm 9.21}$ | $13.30_{\pm 8.65}$ | $9.36_{\pm 5.96}$ | $21.51_{\pm 10.04}$ | $27.44_{\pm 4.32}$ | $15.12_{\pm 9.36}$ | $16.13_{\pm 8.46}$ | $10.61_{\pm 6.52}$ | $19.94_{\pm 11.14}$ |

# H ADDITIONAL EVALUATION RESULTS AND DISCUSSIONS

## H.1 BENIGN EMPIRICAL PERFORMANCE

We present the benign empirical cumulative rewards of the three aggregation protocols (PARL, TPARL, and DPARL) applied on subpolicies trained using three offline RL algorithms (DQN, QR-DQN, and C51) in Table 4. The cumulative reward is obtained by running a trajectory of maximum length $H$ and accumulating the reward achieved at each time step.

We discuss the conclusions on multiple levels. We choose $H = 1000$ for Atari games and $H = 30$ for Highway environment and report results averaged over 20 runs. On the **RL algorithm** level, we see that QR-DQN achieves the highest score in most cases. On the **aggregation protocol** level, temporal aggregation enhances the performance on Freeway while dampens the performance on Breakout and Highway. We particularly note that the results obtained by using temporal aggregation (TPARL and DPARL) gives significantly smaller variance compared to the single-step aggregation PARL, especially in Breakout. This implies that temporal aggregation can help stabilize the policy. However, the conclusion does not hold in Highway, which is an interesting phenomenon that is worth investigating. On the **partition number** level, a larger partition number can give slightly better results. On the **environment** level, Freeway is simpler and more stable than Breakout and Highway.

## H.2 COMPARISON OF COPA WITH STANDARD TRAINING

We provide additional experimental results on the comparison between the empirical performance of our COPA and the standard training in terms of the *convergence speed* (Figure 3) and *policy quality* (Table 5).

In Figure 3, we aim to show the comparison of the convergence speed. For *our proposed training*, we plot all training curves of the sampled 5 subpolicies in blue, where each subpolicy is trained on one partition of the dataset. (We do not plot all 50 curves for visual clarity.) For *standard training*, we plot the training curve of the standard policy in red, where the single policy is trained on the entire dataset. We see that on **Freeway**, most subpolicies converge slower than the standard policy, but will reach similar convergence value as the standard training one; while there also exist a few subpolicies that fail to be trained. On **Breakout**, within 50 epochs, we observe substantial fluctuation for the training curves of all the policies, as well as large variance for the achieved reward at the last

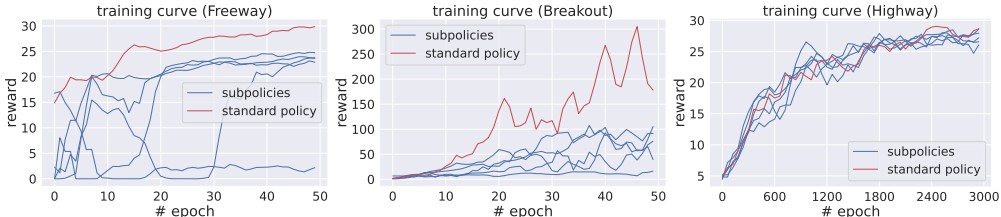

Figure 3: **The *convergence speed* of the proposed partition-based training compared with the standard training.** For *our proposed training*, we plot all training curves of the sampled 5 subpolicies in **blue**, where each subpolicy is trained on one partition of the dataset. (We do not plot all 50 curves for visual clarity). For *standard training*, we plot the training curve of the standard policy in **red**, where the single policy is trained on the entire dataset. In Atari games, we train each policy (or subpolicy) for a fixed number of 50 epochs, where each epoch consumes 1M randomly sampled training data. In Highway environment, we train each policy (or subpolicy) for a fixed number of 3000 epochs, where each epoch consumes 1K randomly sampled training data. The offline RL algorithm used is DQN.

Table 5: **The *policy quality* measured by *empirical cumulative reward* of the proposed aggregation protocols (PARL, TPARL, and DPARL) compared with the standard training.** In our aggregation, we aggregate over $u$ subpolicies with $u$ equal to 30 or 50 . We report results averaged over 20 runs of varying randomness in the environment, where each run is an episode of length at most 1000 for Atari games and 30 for Highway environment. The offline RL algorithm used is DQN.

| | standard trained policy | $u = 30$ | | | $u = 50$ | | |
|---|---|---|---|---|---|---|---|
| | | PARL | TPARL $(W = 4)$ | DPARL $(W_{\max} = 5)$ | PARL | TPARL $(W = 4)$ | DPARL $(W_{\max} = 5)$ |
| Freeway | $12.00_{\ \pm 0.89}$ | $10.90_{\ \pm 0.77}$ | $12.00_{\ \pm 0.45}$ | $11.45_{\ \pm 0.74}$ | $10.95_{\ \pm 0.97}$ | $11.65_{\ \pm 0.65}$ | $11.90_{\ \pm 0.89}$ |
| Breakout | $97.60_{\ \pm 117.28}$ | $58.65_{\ \pm 40.83}$ | $13.50_{\ \pm 11.41}$ | $36.00_{\ \pm 13.39}$ | $60.25_{\ \pm 31.99}$ | $14.95_{\ \pm 7.24}$ | $45.00_{\ \pm 13.44}$ |
| Highway | $28.90_{\ \pm 0.87}$ | $28.07_{\ \pm 3.86}$ | $12.69_{\ \pm 6.17}$ | $23.55_{\ \pm 9.01}$ | $29.05_{\ \pm 0.62}$ | $13.43_{\ \pm 8.29}$ | $22.51_{\ \pm 8.01}$ |

epoch. Thus, on these two **Atari games**, we cannot draw conclusions w.r.t. the convergence. Given more computational resources to run more epochs, we would be able to draw more informative conclusions. In contrast, on **Highway**, all subpolicies display similarly good convergence properties, showing comparable convergence speed and value with the standard training on the entire dataset.

In Table 5, we aim to compare the policy quality of the aggregated policy derived in our COPA framework (*i.e.*, PARL, TPARL, and DPARL) with the policy obtained from standard training. We see that on **Freeway**, our three protocols achieve comparable results with the policy obtained by standard training on the entire dataset; on **Breakout**, although the quality of our obtained policies is lower, our policies are much more stable with significantly lower variance than the standard training; on **Highway**, only PARL obtains comparable results to the standard training policy, indicating the lack of temporal continuity in this environment.

Table 6: **Average window size** (*i.e.*, $\sum_{t=1}^{T} W_t/T$) for the *aggregation protocol* DPARL ($W_{\max} = 5$) applied on subpolicies trained using three *offline RL algorithms* (DQN, QR-DQN, and C51), with the number of subpolicies (*i.e.*, #partitions) $u$ equal to 30 or 50. We report results averaged over all time steps in 20 runs.

| | $u = 30$ | | | $u = 50$ | | |
|---|---|---|---|---|---|---|
| | DQN | QR-DQN | C51 | DQN | QR-DQN | C51 |
| Freeway | $2.69_{\ \pm 1.73}$ | $2.59_{\ \pm 1.71}$ | $2.64_{\ \pm 1.74}$ | $2.79_{\ \pm 1.73}$ | $2.70_{\ \pm 1.73}$ | $2.72_{\ \pm 1.73}$ |
| Breakout | $2.28_{\ \pm 1.46}$ | $2.44_{\ \pm 1.55}$ | $2.32_{\ \pm 1.48}$ | $2.33_{\ \pm 1.48}$ | $2.40_{\ \pm 1.52}$ | $2.39_{\ \pm 1.52}$ |
| Highway | $1.97_{\ \pm 0.44}$ | $2.21_{\ \pm 0.27}$ | $2.16_{\ \pm 0.37}$ | $2.23_{\ \pm 0.32}$ | $2.26_{\ \pm 0.24}$ | $2.18_{\ \pm 0.37}$ |

Table 7: **Runtime (unit: seconds)** of the *aggregation protocol* DPARL ($W_{\max} = 5$) applied on subpolicies trained using *offline RL algorithm* DQN, with the number of subpolicies (*i.e.*, partition number) $u$ equal to 30 or 50. We compare with the *standard testing* which tests the runtime of a single trained DQN policy without using our framework. We report results averaged over 20 runs of varying randomness in the environment, where each run is an episode of length at most 1000.

|  | standard testing | DPARL ($u = 30$) | DPARL ($u = 50$) |
|---|---|---|---|
| Freeway | $2.53 _{\pm 0.50}$ | $56.37 _{\pm 1.75}$ | $79.05 _{\pm 4.27}$ |
| Breakout | $2.00 _{\pm 0.56}$ | $72.05 _{\pm 13.09}$ | $108.05 _{\pm 8.71}$ |

### H.3 MORE ANALYTICAL STATISTICS FOR DPARL

**Selected Window Size.** We provide the mean and variance for the selected window sizes in DPARL in Table 6. In our experiments, the *maximum window size* $W_{\max} = 5$, while the *average selected window sizes* are all around half of the maximum value. Thus, the average certification time is expected to be much smaller compared with the worst-case time complexity.

**Running Time.** We provide the running time in Table 7. Specifically, we compare the running time of DPARL ($W_{\max} = 5$) applied on $u = 30$ or 50 subpolicies trained using *offline RL algorithm* DQN with the *normal testing* which tests the runtime of a single trained DQN policy without using our framework. As we have shown in the remark of Theorem 3 in Section 4.3, the time complexity of DPARL is $O\left(W_{\max}^2 |\mathcal{A}|^2 u + W_{\max} |\mathcal{A}|^2 u \log u\right)$. In Table 7, we also see that the runtime of DPARL scales roughly quasilinearly with the number of subpolicies $u$, and quadratically with the action set size $|\mathcal{A}|$, which is 3 for Freeway and 4 for Breakout. We omit the runtime for Highway, since the horizon length, the environment type, and the neural network size are all different for Highway and Atari games.

### H.4 MAXIMUM TOLERABLE POISONING THRESHOLD VS. TOTAL TRAJECTORY NUMBER

We provide the total number of trajectories in the offline dataset used for training the environments Freeway, Breakout and Highway below, as well as the corresponding ratio of the maximum tolerable poisoning threshold w.r.t. the total number of trajectories. For **Freeway**, the total number of trajectories in the entire offline dataset is 121,892, and the ratio ($\max \overline{K}_t$ / # total trajectories) is 0.0002; for **Breakout**, the number is 209,049, and the ratio is 0.0001. We emphasize that, as shown in previous work on probable defense against poisoning in supervised learning (Levine & Feizi, 2020), the number of instances they can certify is also not high especially for challenging tasks (*e.g.*, certifying 20 instances on the dataset GTSRB in Levine & Feizi (2020), attaining the ratio 0.0005), which may imply the intrinsic difficulty of certifying against poisoning attacks. Given such difficulty, we already achieve reasonable certification as the first work on certified robust RL against poisoning, and we hope future works can further improve upon our results.

### H.5 COMPARISON OF FREEWAY, BREAKOUT, AND HIGHWAY

Freeway and Breakout are two typical types of games with distinctive game properties. Freeway is a simple game where the agent aims to cross the road and avoid the traffic. In most of the time steps, the agent would take the "forward" action; only when there is a need to avoid the traffic will the agent stop and wait. In comparison, the game Breakout involves more complicated interactions between the paddle and the environment. The paddle position and velocity both play important roles in order to catch and bounce the ball at an appropriate angle, which requires a frequent switch of the paddle moving direction. Similar to Breakout, as an autonomous driving environment, Highway requires the agent to accurately analyze the rapidly changing environment around it and react quickly, leading to frequent changes in vehicle's actions.

Given the properties of the environments, we note that in Freeway, adjacent time steps may share consistent action selections, which is not necessarily true in the Breakout and Highway. Thus, it would be expected to be beneficial to consider the past history for making the current decision in Freeway, while the past history may interfere with the action selection in Breakout and Highway.

Table 8: **Action change ratio (in percentage)** of three *aggregation protocols* (PARL, TPARL, and DPARL) applied on subpolicies trained in Highway environment using three *offline RL algorithms* (DQN, QR-DQN, and C51), with the number of subpolicies (*i.e.*, #partitions) $u$ equal to 30, 50, or 100. We report results averaged over 20 runs of varying randomness in the environment.

| $u$ | RL Algorithm | Aggregation Protocol | | |
|---|---|---|---|---|
| | | PARL | TPARL $(W = 4)$ | DPARL $(W_{\max} = 5)$ |
| 30 | DQN | $59.79_{\pm 8.99}$ | $24.89_{\pm 10.47}$ | $33.82_{\pm 8.75}$ |
| | QR-DQN | $61.00_{\pm 12.74}$ | $31.14_{\pm 6.43}$ | $37.80_{\pm 7.16}$ |
| | C51 | $59.00_{\pm 10.39}$ | $18.17_{\pm 11.52}$ | $35.44_{\pm 8.65}$ |
| 50 | DQN | $55.50_{\pm 8.84}$ | $26.05_{\pm 11.69}$ | $40.07_{\pm 13.74}$ |
| | QR-DQN | $60.37_{\pm 10.31}$ | $28.48_{\pm 6.81}$ | $37.78_{\pm 12.15}$ |
| | C51 | $59.39_{\pm 12.34}$ | $28.04_{\pm 10.71}$ | $38.21_{\pm 11.23}$ |
| 100 | DQN | $48.67_{\pm 5.91}$ | $22.17_{\pm 7.65}$ | $33.45_{\pm 9.70}$ |
| | QR-DQN | $61.17_{\pm 8.38}$ | $24.12_{\pm 9.85}$ | $36.76_{\pm 7.87}$ |
| | C51 | $61.83_{\pm 10.41}$ | $22.91_{\pm 13.34}$ | $34.06_{\pm 9.19}$ |

**Comparisons of Bottleneck states in Atari Games.** In Breakout, the game goal is to control the paddle so that the ball is bounced to hit the brick. There are clearly very different stages in the Breakout game, *e.g.*, when the ball is flying towards the paddle and when it is bounced back. An example of the *bottleneck state* is when the ball is approaching the paddle. The poisoning behavior may lead to the disastrous effect that the policy learns to control the paddle to slide in the opposite direction of the ball at such bottleneck states, while other states may not be as vulnerable. In comparison, in Freeway, the game goal is to control the agent to cross the road while avoiding the traffic. The agent is faced with similar traffic conditions everywhere on the road, and can make stable choices regardless of the complex situation. Thus there are very few *bottleneck states*.

**A Study on the Frequency of Action Change in Highway.** In Table 8, we present the action change ratio (# action changes / trajectory length) in Highway environment. Comparing among the aggregation protocols, we first note that the single-step aggregation policy $\pi_P$ frequently alters actions as the reaction to the rapidly changing driving environment, while the temporal aggregation protocol $\pi_T$ changes actions less frequently. This is because in TPARL, the agent is explicitly forced to take stable actions based on a fixed window size. Second, comparing the action change ratio with the benign empirical performance in Table 4, we observe that $\pi_P$ and $\pi_T$ achieves the highest and lowest empirical performance respectively, which indicates the correlation between action change ratio with the achieved empirical reward. Specifically, in TPARL, the action change is limited as a result of the enforced window size, leading to the limited benign performance.

### H.6 FULL RESULTS OF ROBUSTNESS CERTIFICATION FOR PER-STATE ACTION STABILITY

As a complete set of results for per-state action certification apart from Figure 1 in Section 5.1, we present the entire cumulative histogram of *tolerable poisoning thresholds* in Figure 4 and Figure 5, together with the *average tolerable poisoning thresholds* in Table 9. The definitions of the two metrcis can be found in Section 5.1.

Basically, the conclusions in terms of the comparisons on the RL algorithm level, the aggregation protocol level, the partition number level, and the game level are all similar to that derived in Section 5.1.

### H.7 FULL RESULTS OF ROBUSTNESS CERTIFICATION FOR CUMULATIVE REWARD BOUND

In addition to the evaluation results provided in Figure 2 in section 5.2, we provide a more comprehensive set of evaluation results with more settings of trajectory length $H$ here in Figure 6.

We draw similar conclusions as discussed in Section 5.2. Specifically, in Highway, C51 achieves higher certified lower bounds than other two when the poisoning size is large, which can be explained by the larger portion of states that can tolerate large poisoning sizes as shown in Figure 5.

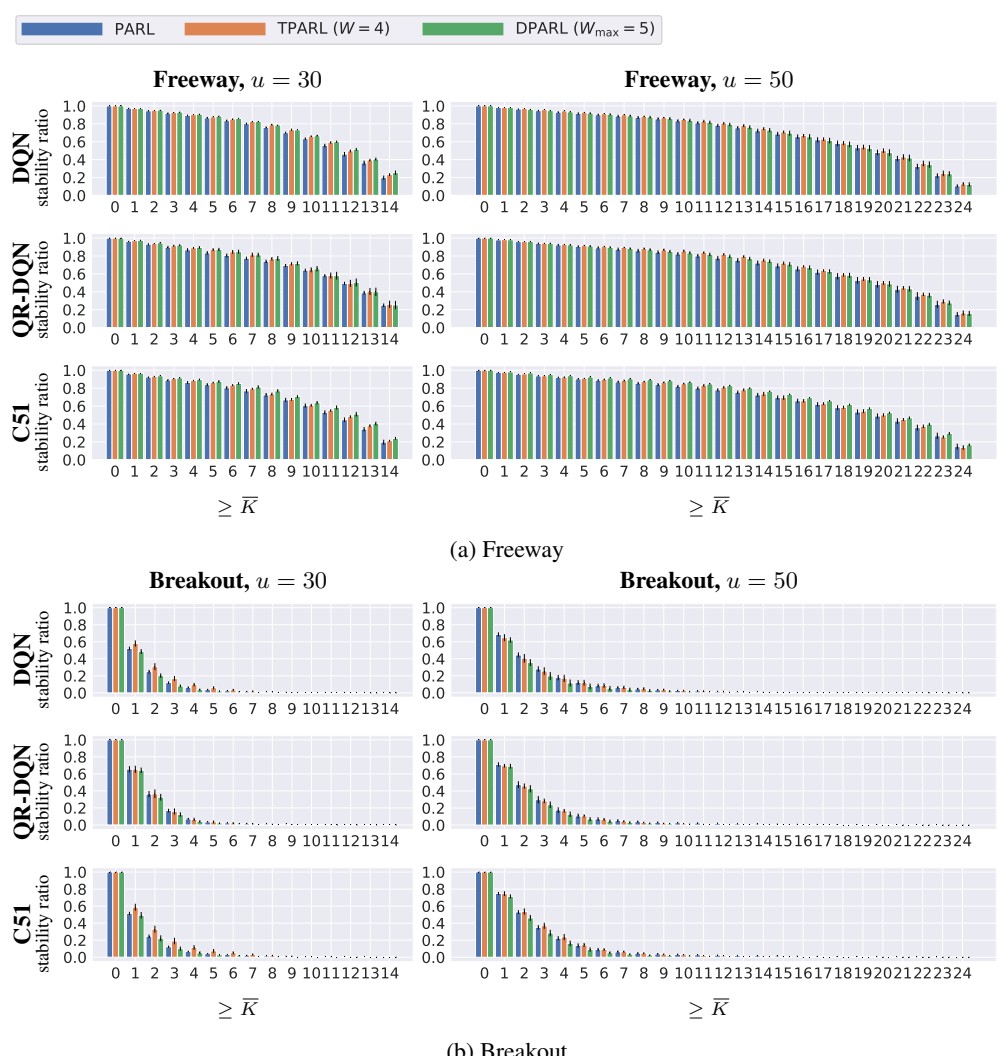

(a) Freeway

(b) Breakout

Figure 4: **Robustness certification for per-state action stability on Atari games (full results).** We plot the cumulative histogram of the *tolerable poisoning size* $\overline{K}$ for all time steps in one trajectory. We provide results on two games (Freeway and Breakout), two partition numbers ($u = 30$ and $u = 50$), and a comparison of three certification methods (PARL, TPARL, and DPARL). The results are averaged over 20 runs considering the randomness in the game environment, and the short vertical bar on top of each bar represents the standard deviation.

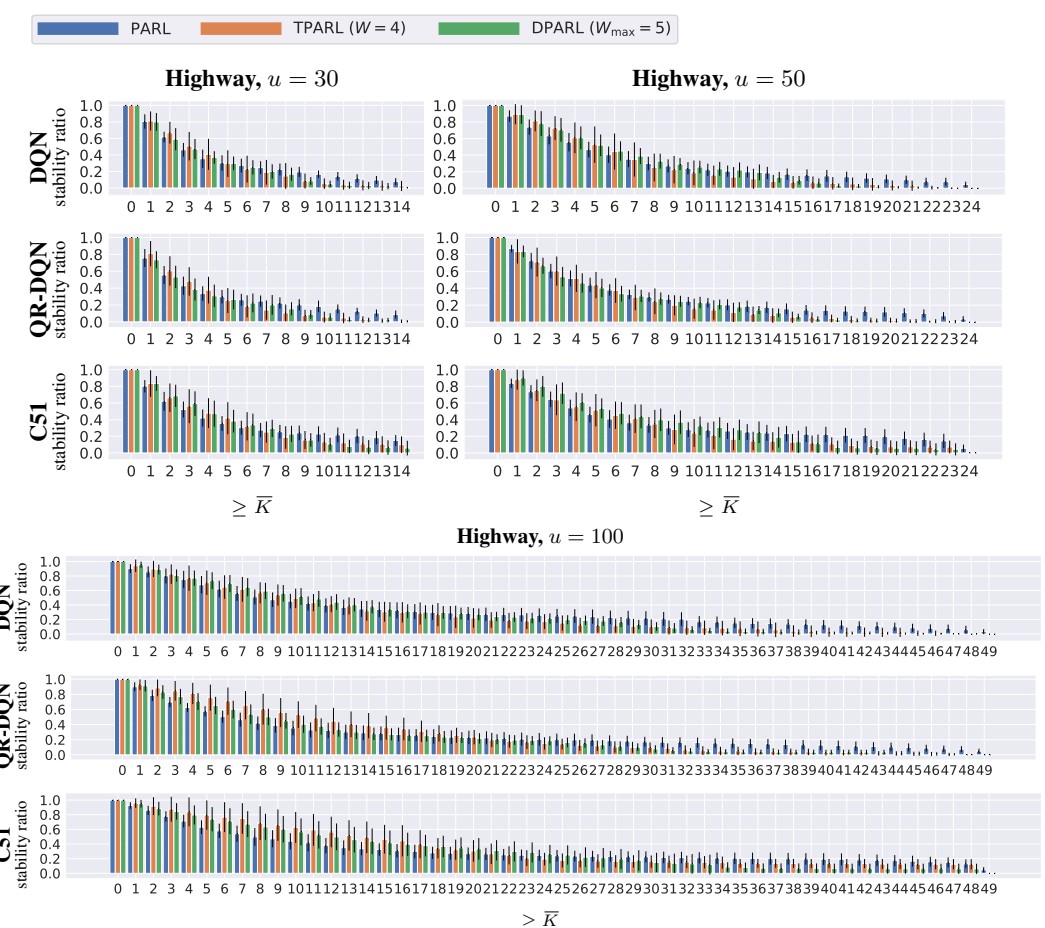

Figure 5: **Robustness certification for per-state action stability on Highway environment (full results).** We plot the cumulative histogram of the *tolerable poisoning size* $\overline{K}$ for all time steps in one trajectory. We provide results on Highway environment, three partition numbers ($u = 30$, $u = 50$ and $u = 100$), and a comparison of three certification methods (PARL, TPARL, and DPARL). The results are averaged over 20 runs considering the randomness in the environment, and the short vertical bar on top of each bar represents the standard deviation.

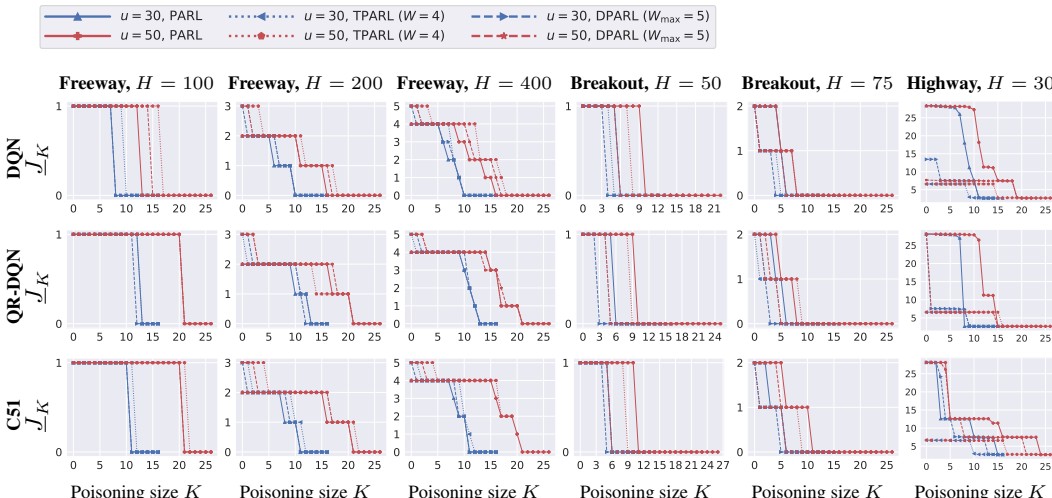

Figure 6: **Robustness certification for cumulative reward (full results).** We plot the *lower bound of cumulative reward* $\underline{J}_K$ w.r.t. *poisoning size* $K$ under three different certification methods (PARL, TPARL ($W = 4$), DPARL ($W_{\max} = 5$)) with two partition numbers ($u \in \{30, 50\}$). Each row corresponds to one RL algorithm, and each column corresponds to one setting of trajectory length $H$.

Table 9: **Average tolerable poisoning thresholds** of three *aggregation protocols* (PARL, TPARL, and DPARL) applied on subpolicies trained using three *offline RL algorithms* (DQN, QR-DQN, and C51), with the number of subpolicies (*i.e.*, #partitions) $u$ equal to 30 or 50. We report results averaged over 20 runs of varying randomness in the environment.

| Freeway | $u = 30$ | | | $u = 50$ | | |
|---|---|---|---|---|---|---|
| | PARL | TPARL $(W = 4)$ | DPARL $(W_{\max} = 5)$ | PARL | TPARL $(W = 4)$ | DPARL $(W_{\max} = 5)$ |
| DQN | $9.90 \pm 0.14$ | $10.21 \pm 0.09$ | $10.30 \pm 0.09$ | $16.78 \pm 0.42$ | $17.16 \pm 0.29$ | $16.88 \pm 0.45$ |
| QR-DQN | $9.87 \pm 0.20$ | $10.11 \pm 0.30$ | $10.16 \pm 0.34$ | $16.79 \pm 0.52$ | $17.31 \pm 0.29$ | $17.03 \pm 0.40$ |
| C51 | $9.57 \pm 0.24$ | $9.83 \pm 0.15$ | $10.11 \pm 0.19$ | $16.82 \pm 0.40$ | $17.08 \pm 0.34$ | $17.61 \pm 0.15$ |

| Breakout | $u = 30$ | | | $u = 50$ | | |
|---|---|---|---|---|---|---|
| | PARL | TPARL $(W = 4)$ | DPARL $(W_{\max} = 5)$ | PARL | TPARL $(W = 4)$ | DPARL $(W_{\max} = 5)$ |
| DQN | $1.08 \pm 0.09$ | $1.28 \pm 0.15$ | $0.85 \pm 0.09$ | $2.07 \pm 0.22$ | $1.92 \pm 0.30$ | $1.55 \pm 0.28$ |
| QR-DQN | $1.38 \pm 0.09$ | $1.33 \pm 0.18$ | $1.18 \pm 0.10$ | $2.12 \pm 0.27$ | $1.93 \pm 0.14$ | $1.71 \pm 0.19$ |
| C51 | $1.10 \pm 0.08$ | $1.42 \pm 0.19$ | $0.93 \pm 0.13$ | $2.43 \pm 0.21$ | $2.37 \pm 0.19$ | $1.90 \pm 0.19$ |

| Highway | $u = 30$ | | | $u = 50$ | | |
|---|---|---|---|---|---|---|
| | PARL | TPARL $(W = 4)$ | DPARL $(W_{\max} = 5)$ | PARL | TPARL $(W = 4)$ | DPARL $(W_{\max} = 5)$ |
| DQN | $4.38 \pm 0.78$ | $2.99 \pm 1.08$ | $3.75 \pm 1.85$ | $7.16 \pm 1.58$ | $6.27 \pm 1.61$ | $6.32 \pm 0.98$ |
| QR-DQN | $4.18 \pm 0.91$ | $3.35 \pm 1.44$ | $2.84 \pm 0.55$ | $6.52 \pm 1.17$ | $5.02 \pm 1.60$ | $4.61 \pm 1.20$ |
| C51 | $4.97 \pm 1.09$ | $4.31 \pm 1.45$ | $4.12 \pm 0.95$ | $7.95 \pm 1.46$ | $6.55 \pm 2.20$ | $6.69 \pm 1.47$ |

We additionally point out that the value $\underline{J}_K$ achieved at poisoning size $K = 0$ (*e.g.*, $\underline{J}_K = 4$ for Freeway, 2 for Breakout, and 28.31 for Highway under $\pi_P$ over $u = 50$ DQN subpolicies) corresponds to the case where there is no poisoning at all on the training set. Our successive bounds under larger $K$ are non-vacuous compared to this value.

# I  A BROADER DISCUSSION ON RELATED WORK

## I.1  POISONING ATTACKS IN RL

Below, we provide a brief discussion on the related works on policy poisoning and reward poisoning (Ma et al., 2019; Sun et al., 2021; Huang & Zhu, 2019).

Ma et al. (2019) and Huang & Zhu (2019) study reward poisoning where the attacker can modify the rewards in an offline dataset or the reward signals during the online interaction. The attacker's goal is to force learning a particular target policy, or minimize the agent's reward in the original task. Under this framework, Ma et al. (2019) considers two victim learners with specific assumptions on the learner structure and develops attacks for them, while Huang & Zhu (2019) provides a theoretical analysis on the conditions for successful attacks against a Q-learning agent. In comparison to only reward poisoning in Ma et al. (2019) and Huang et al. (2017), Sun et al. (2021) focuses on general policy poisoning attacks for policy gradient learners in the online RL setting by using the vulnerability-awareness metric to decide when to poison, and the adversarial critic to guide the poisoning. Sun et al. (2021) achieve successful poisoning against policy-based agents in various complex environments in online RL, but it remains an interesting open problem as to how to poison the offline RL dataset which is agnostic to the learning algorithm.

## I.2  EMPIRICALLY ROBUST RL

**Robust RL against Evasion Attacks.** We briefly review several categories of RL methods that demonstrate empirical robustness against evasion attacks.

*Randomization methods* (Tobin et al., 2017; Akkaya et al., 2019) were first proposed to encourage exploration. This type of method was later systematically studied for its potential to improve model robustness. NoisyNet (Fortunato et al., 2017) adds parametric noise to the network's weight during training, providing better resilience to both training-time and test-time attacks (Behzadan & Munir, 2017; 2018), also reducing the transferability of adversarial examples, and enabling quicker recovery with fewer number of transitions during phase transition.

Under the *adversarial training* framework, Kos & Song (2017) and Behzadan & Munir (2017) show that re-training with random noise and FGSM perturbations increases the resilience against adversarial examples. Pattanaik et al. (2018) leverage attacks using an engineered loss function specifically designed for RL to significant increase the robustness to parameter variations. RS-DQN (Fischer et al., 2019) is an *imitation learning* based approach that trains a robust student-DQN in parallel with a standard DQN in order to incorporate the constrains such as SOTA adversarial defenses (Madry et al., 2017; Mirman et al., 2018).

SA-DQN (Zhang et al., 2020a) is a *regularization* based method that adds regularizers to the training loss function to encourage the top-1 action to stay unchanged under perturbation.

Built on top of the *neural network verification* algorithms (Gowal et al., 2018; Weng et al., 2018), Radial-RL (Oikarinen et al., 2020) proposes to minimize an adversarial loss function that incorporates the upper bound of the perturbed loss, computed using certified bounds from verification algorithms. CARRL (Everett et al., 2021) aims to compute the lower bounds of action-values under potential perturbation and select actions according to the worst-case bound, but it relies on linear bounds (Weng et al., 2018) and is only suitable for low-dimensional environments.

These robust RL methods only provide empirical robustness against perturbed state inputs during test time, but cannot provide theoretical guarantees for the performance of the trained models under any bounded perturbations.

**Robust RL against Poisoning Attacks.** Banihashem et al. (2021) considers the *threat model* of reward poisoning attacks proposed by Ma et al. (2019); Rakhsha et al. (2020); Zhang et al. (2020b), where the attacker aims to force learning a target policy while minimizing the cost of reward manipulation. As shown in Ma et al. (2019); Rakhsha et al. (2020); Zhang et al. (2020b), this optimization problem is feasible and always has a unique optimal solution. Leveraging this property, Banihashem et al. (2021) formulates the defense task as another optimization problem which aims to optimize the agent's worst case performance among the set of plausible candidates of the true reward function. They specifically consider two settings regarding the agent's knowledge of the attack parameter, and provide lower bounds on the performance of the defense policy. In contrast to Banihashem et al. (2021) which focuses on a specific type of attack, our paper considers the *threat model* of general poisoning attacks, where the attacker has the power to manipulate the training trajectories arbitrarily. Given limited knowledge regarding the attack, our proposed COPA framework is general and applicable to any potential attack. Our robustness is derived from the aggregation over both sub-policy level and temporal level. One similarity between Banihashem et al. (2021) and our work is that we both aim to provide lower bounds of the cumulative reward for our proposed method as the provable guarantee / certification criteria.

### I.3 ROBUSTNESS CERTIFICATION FOR RL

Wu et al. (2022) provide the first robustness certification for RL against *test-time evasion attacks* following the line of work on randomized smoothing (Cohen et al., 2019; Salman et al., 2019). Concretely, they apply per-state smoothing to achieve the certification for per-state action stability, as well as trajectory smoothing to obtain the certification for cumulative reward. Notably, Wu et al. (2022) propose an adaptive tree search algorithm to explore all possible trajectories and thus derive the robustness guarantee for cumulative reward. The relationship between our COPA-SEARCH and their Algorithm 3 is discussed in detail in Appendix E.3. In comparison, robustness in their work is derived from *smoothing*, while our robustness comes from *aggregation*. We propose aggregation protocols and certification methods that leverage the temporal information by dynamically aggregating over the past time steps, which is not covered in Wu et al. (2022).

