# OpenReview forum: "COPA: Certifying Robust Policies for Offline Reinforcement Learning against Poisoning Attacks"
_ICLR.cc/2022/Conference — ICLR 2022 Poster_

### Official Review · Reviewer_zvac · 2021-10-31

**Correctness:** 4
**Technical Novelty And Significance:** 3
**Empirical Novelty And Significance:** 4
**Recommendation:** 5
**Confidence:** 2

**Main Review:**

Strengths:

(1). This paper is the first to study policy certification for offline reinforcement learning. Given that "poisoning attacks in RL" is a popular recent topic, this paper is able to push the frontier of this research line even more and open potential future research questions.

(2). The paper provided a systematic theoretical study of policy certification based on two criteria --- per-state action stability and cumulative reward lower bound. Both criteria are sensible since they characterize the objective of general RL tasks. The theoretical results can provide guidance for designing robust RL algorithms in the future research.

(3). Experiments consolidated the theoretical findings in this paper.

Weaknesses:

(1). The idea of splitting the training set into subsets and then apply aggregation on sub-policies trained on each subset is not surprising, and has been studied a lot in previous works, e.g., in [1]. Therefore, the technical contribution of this paper seems a little bit weak, although it studied a new problem.

(2). The applicability of the COPA certification framework is limited in the sense that it only applies to poisoning attacks where the attacker poisons a small fraction of training trajectories. However, an attacker may be able to slightly perturb all trajectories without incurring much effort. For example, the attacker just need to change the data point of a single step (e.g. the first step) in each trajectory, and that already breaks the assumption made in this paper. With all that, I mean the attacker ability is very weak and the certification against weak attackers is not a strong certification.

(3). Another disadvantage of the paper is that the theorems involves very intensive symbols, which can be confusing to readers. The authors should consider adjusting how the results are stated to make it super clear.

(4). Very related papers are missing in the reference. To list a few, please see [2-4]. The authors should discuss them in the related work.

[1] Certified Robustness of Nearest Neighbors against Data Poisoning Attacks
[2] Policy poisoning in batch reinforcement learning and control
[3] Vulnerability-aware poisoning mechanism for online rl with unknown dynamics
[4] Deceptive reinforcement learning under adversarial manipulations on cost signals

**Summary Of The Paper:**

This paper studied how to certificate a policy when the training dataset is partially corrupted in the offline reinforcement learning scenario. Two criteria including per-state action stability certificate and cumulative reward lower bound certificate are proposed. Based on these two criteria, the paper designed the COPA, a general certificate framework, which provably achieved certain level of certificate. The COPA went through two phases during training, where in the first phase, the trajectories are split into subsets of equal size, then sub-policies are trained on each subset, and finally the sub-policies are aggregated to form a single learned policy. Experiments demonstrate the effectiveness of the proposed COPA certification framework.

**Summary Of The Review:**

Overall, I think the paper studied a very important and interesting problem, but the paper needs to be improved in writing, related work, and discussions on potential weakness of the COPA framework.

---

> ### Author Response · Authors · 2021-11-21
> **Official Response to Reviewer zvac**
>
> We thank the reviewer for the thoughtful comments and provide our response below.
>
> > 1. Technical contributions compared with related work Jia et al. [1]
>
> We thank the reviewer for pointing out the related work Jia et al. [1]. Indeed, the high-level idea of partitioning the training set and taking the majority vote among the multiple trained models is studied supervised learning [1,2]. However, we would clarify that our most important contribution is not adapting the partition-based training protocol to RL, but proposing the aggregation protocols (especially the temporal aggregation protocols leveraging the property of RL) and the corresponding non-trivial certification for these aggregation protocols under two certification criteria for robustness in RL. We emphasize that previous works such as [1] do not have the temporal dimension in RL which makes the problem more involved; while we do exploit the temporal continuity with temporal aggregation to enhance our certification which cannot be done in previous works.
>
> > 2. Applicability of the COPA certification framework is limited / The attacker ability is weak.
>
> We thank the reviewer for the thoughtful threat model. In the case of the attacker slightly perturbing all trajectories without incurring much effort, one can extend randomized smoothing based certification (e.g., RAB [3]) to directly provide certification. However, the threat model considered in our work (select a bounded number of trajectories and arbitrarily change them) cannot be practically handled by randomized smoothing. We also emphasize that the threat model in our paper (extending [4] and the $\epsilon$-contamination model) is of severe concern, and is more practical since it is more for attackers to manipulate a subset of trajectories (note that here we allow the manipulation to be arbitrary). In the future, we will extend our framework to cover both the current threat model and the threat model proposed by the reviewer.
>
> > 3. Theorems involve intensive symbols.
>
> We thank the reviewer for the valuable comment. We have simplified the notations in Theorem 1 (Eqn. 3) and Definition 4 (Eqn. 6) in our revision. Hope these modifications help to improve readability.
>
> > 4. Missing related works [5,6,7]
>
> We thank the reviewer for complementing the related works on policy poisoning and reward poisoning [5,6,7]. We provide a brief discussion below, and have accordingly added the references and discussions in Appendix I.3 in our revision.
>
> [5] and [7] study reward poisoning where the attacker can modify the rewards in an offline dataset or the reward signals during the online interaction. The attacker’s goal is to force learning a particular target policy, or minimize the agent’s reward in the original task. Under this framework, [5] considers two victim learners with specific assumptions on the learner structure and develops attacks for them, while [7] provides a theoretical analysis on the conditions for successful attacks against a Q-learning agent. In comparison to only reward poisoning in [5,7], [6] focuses on general policy poisoning attacks for policy gradient learners in the online RL setting by using the vulnerability-awareness metric to decide when to poison, and the adversarial critic to guide the poisoning. [6] achieves successful poisoning against policy-based agents in various complex environments in online RL, but it remains an interesting open problem as to how to poison the offline RL dataset which is agnostic to the learning algorithm.
>
> **References**
>
> [1] Jia, Jinyuan, Xiaoyu Cao, and Neil Zhenqiang Gong. "Certified robustness of nearest neighbors against data poisoning attacks." arXiv preprint arXiv:2012.03765 (2020).
>
> [2] ​​Levine, Alexander, and Soheil Feizi. "Deep Partition Aggregation: Provable Defenses against General Poisoning Attacks." International Conference on Learning Representations. 2020.
>
> [3] Weber, Maurice, et al. "RAB: Provable robustness against backdoor attacks." arXiv preprint arXiv:2003.08904 (2020).
>
> [4] Zhang, Xuezhou, et al. "Corruption-Robust Offline Reinforcement Learning." arXiv preprint arXiv:2106.06630 (2021).
>
> [5] Ma, Yuzhe, et al. "Policy poisoning in batch reinforcement learning and control." arXiv preprint arXiv:1910.05821 (2019).
>
> [6] Sun, Yanchao, Da Huo, and Furong Huang. "Vulnerability-Aware Poisoning Mechanism for Online RL with Unknown Dynamics." International Conference on Learning Representations. 2020.
>
> [7] Huang, Yunhan, and Quanyan Zhu. "Deceptive reinforcement learning under adversarial manipulations on cost signals." International Conference on Decision and Game Theory for Security. Springer, Cham, 2019.

---

> ### Author Response · Authors · 2021-11-29
> **Further Follow-up to Reviewer zvac**
>
> Dear Reviewer zvac,
>
> As the end of the discussion is approaching, we would like to kindly ask you to consider our responses to your concerns. We are very thankful to your comments and suggestions that helped improve our paper, and have reflected them in our revision. (We posted a “Paper Revision Summary” for your reference.) We also provided explanations and clarifications that we hope can address your concerns. Please let us know if you have any final questions that we can address. Thanks!
>
> Best regards,
>
> Paper4064 Authors

---

### Official Review · Reviewer_6rjC · 2021-11-01

**Correctness:** 3
**Technical Novelty And Significance:** 2
**Empirical Novelty And Significance:** 2
**Recommendation:** 5
**Confidence:** 4

**Details Of Ethics Concerns:**

I do not have concerns about ethics.

**Main Review:**

Strengths:

1. The studied problem, certifiable defense against poisoning in offline RL, is important to the community. As far as I know, this is the first method in certifying both per-state action stability and cumulative reward against offline poisoning attacks in RL.
2. The problem setup is clearly stated. The authors provide analysis for 3 aggregation protocols, which could be helpful for understanding the  partition-aggregation based methods in RL.
3. The experiment compares several deep RL methods in terms of their intrinsic vulnerability, which is interesting.

Weaknesses:

1. For writing:

(1) Many details and definitions that are essential for understanding the method are omitted to the Appendix, which makes the flow of the paper not very smooth, especially in Section 3.

(2) I don't see the necessity of separately discussing 3 aggregation protocols in the main paper. The 3 methods do not differ by a lot. The empirical results in Figure 1 and 2 also show that they do not have significant differences. Discussing all 3 protocols separately results in the weakness mentioned above -- too many details are omitted in the main paper. I would like to see the authors focus on the most interesting one in the main paper and provide detailed explanations, then put the other two variants to the Appendix.

(3) Some notations are not very clear. For example, $K_t$ is not defined in or before Theorem 1. Does it mean the poisoning threshold such that the action in the $t$-th step does not change? What is $p$ in Theorem 1? What is the argmax operation in Equation 3 maxing over? This is the most important result in the paper, but the presentation is really confusing.

2. For the proposed algorithm:

(1) The temporal continuity assumption does not seem to be a standard one in RL literature. Can the authors justify how realistic this assumption is?

(2) I am very confused by the COPA-SEARCH algorithm. Why can the learner search in the MDP in the offline setting where the agent is only supposed to have access to a set of trajectories? For example, in Algorithm 5 Line 24, the learner directly queries the transition function P for the next state of $s_t,a$. The Input of Algorithm 5 also requires the environment $\mathcal{E}$. If the algorithm requires to query the environment arbitrarily, the algorithm does not make any sense. If you already know the MDP dynamics, the poisoning in trajectories do not matter at all. This is the biggest concern I have. Please let me know if I missed any important statement.

3. For the evaluation:

(1) The authors compute the certified poisoning threshold $\bar{K}$, but do not present the number of total trajectories.

(2) The experiment results are not convincing. Deep RL training requires a large number of trajectories (millions of interactions), but the certified $\bar{K}$ provided by the proposed method is too small (around 10) based on Figure 1. That is, the defense method is too weak. In the cumulative certification, there is no comparison between the performance of the proposed method and the vanilla training method under practical poisoning attacks, so that we do not know how good the method is in terms of practical defending. There is only the result of the computed reward lower bound in Figure 2. More importantly, the certified lower bounds are too low, and become vacuous (lower bound=0) when $K>10$ or $K>5$. A normal DQN agent in Breakout should get >300 rewards. But the lower bound is 0~2, which does not make much sense, because the reward in Breakout is no smaller than 0.

4. Some related works are missing. For example, [1] also proposes a defense method agains poisoning attacks, although in a different setting. It is worth mentioning this work and discussing the differences between the settings of this paper and [1].

[1] Banihashem, Kiarash, Adish Singla, and Goran Radanovic. "Defense Against Reward Poisoning Attacks in Reinforcement Learning." arXiv preprint arXiv:2102.05776 (2021).

**Summary Of The Paper:**

This paper proposes a defense method against poisoning attacks in offline RL. The authors propose a framework named COPA to certify the number of tolerable poisoned trajectories. The proposed defense includes a partitioning stage which separate the (possibly poisoned trajectories), a training stage where $u$ sub-policies are trained with $u$ partitions,  and an aggregation stage where the sub-policies are aggregated to produce the final policy. Three aggregation protocols are proposed with corresponding certification results.

**Summary Of The Review:**

This paper studies an interesting and important topic, and the proposed method makes intuitive sense. However there are many flaws in terms of writing, algorithm assumption, and empirical results. So I do no think this paper can be accepted before a major modification is made.

---

> ### Author Response · Authors · 2021-11-21
> **Official Response to Reviewer 6rjC (4/4)**
>
> > 4. Missing related work
>
> We thank the reviewer for pointing out the related work Banihashem et al. [6]. We provide a comparison between our COPA framework with [6] below, and have updated the references in Section 5 Related Work and added more discussions in Appendix I.2 in our revision.
>
> Banihashem et al. [6] considers the *threat model* of reward poisoning attacks proposed in [7,8,9], where the attacker aims to force learning a target policy while minimizing the cost of reward manipulation. As shown in [7,8,9], this optimization problem is feasible and always has a unique optimal solution. Leveraging this property, [6] formulates the defense task as another optimization problem which aims to optimize the agent’s worst case performance among the set of plausible candidates of the true reward function. They specifically consider two settings regarding the agent’s knowledge of the attack parameter, and provide lower bounds on the performance of the defense policy. In contrast to [6] which focuses on a specific type of attack, our paper considers the *threat model* of general poisoning attacks, where the attacker has the power to manipulate the training trajectories arbitrarily. Given limited knowledge regarding the attack, our proposed COPA framework is general and applicable to any potential attack. Our robustness is derived from the aggregation over both sub-policy level and temporal level. One similarity between [6] and our work is that we both aim to provide lower bounds of the cumulative reward for our proposed method as the provable guarantee / certification criteria.
>
> **References**
>
> [1] Veerapaneni, Rishi, et al. "Entity abstraction in visual model-based reinforcement learning." Conference on Robot Learning. PMLR, 2020.
>
> [2] Legenstein, Robert, Niko Wilbert, and Laurenz Wiskott. "Reinforcement learning on slow features of high-dimensional input streams." PLoS computational biology 6.8 (2010): e1000894.
>
> [3] Agarwal, Rishabh, Dale Schuurmans, and Mohammad Norouzi. "An optimistic perspective on offline reinforcement learning." International Conference on Machine Learning. PMLR, 2020.
>
> [4] ​​Levine, Alexander, and Soheil Feizi. "Deep Partition Aggregation: Provable Defenses against General Poisoning Attacks." International Conference on Learning Representations. 2020.
>
> [5] Rosenfeld, Elan, et al. "Certified robustness to label-flipping attacks via randomized smoothing." International Conference on Machine Learning. PMLR, 2020.
>
> [6] Banihashem, Kiarash, Adish Singla, and Goran Radanovic. "Defense Against Reward Poisoning Attacks in Reinforcement Learning." arXiv preprint arXiv:2102.05776 (2021).
>
> [7] Ma, Yuzhe, et al. "Policy poisoning in batch reinforcement learning and control." Advances in Neural Information Processing Systems (2019).
>
> [8] Rakhsha, Amin, et al. "Policy teaching via environment poisoning: Training-time adversarial attacks against reinforcement learning." International Conference on Machine Learning. PMLR, 2020.
>
> [9] Zhang, Xuezhou, et al. "Adaptive reward-poisoning attacks against reinforcement learning." International Conference on Machine Learning. PMLR, 2020.

---

> ### Author Response · Authors · 2021-11-21
> **Official Response to Reviewer 6rjC (3/4)**
>
> > 3. For the evaluation
>
> **(1) The number of total trajectories**
>
> We thank the reviewer for the question. We provide the total number of trajectories in the offline dataset used for training the games Freeway and Breakout below, as well as the corresponding ratio of the maximum tolerable poisoning threshold w.r.t. the total number of trajectories. For **Freeway**, the total number of trajectories in the entire offline dataset is 121892, and the ratio ($\max \bar K_t$ / # total trajectories) is 0.0002; for **Breakout**, the number is 209049, and the ratio is 0.0001. We have included the numbers in Appendix H.4 in our revision.
>
> **(2) Unconvincing experimental results**
>
> We thank the reviewer for the detailed comments and address each of them below.
>
> *Small certified $\bar K$*: As presented in the above point, for Freeway, the total number of trajectories used for training is 121892 and the ratio of the tolerable poisoning threshold w.r.t. the total number of trajectories is 0.0002. We emphasize that, as shown in previous work on probable defense against poisoning in supervised learning [4], the number of instances they can certify is not high especially for challenging tasks (e.g., certifying 20 instances on the dataset GTSRB in [4], attaining the ratio 0.0005), which may imply the intrinsic difficulty of certifying against poisoning attacks. Given such difficulty, we already achieve reasonable certification as the first work on certified robust RL against poisoning attacks, and we hope future works can further improve upon our results.  We have included the discussions in Appendix H.4 in our revision.
>
> *Comparison between the performance of the proposed method and the vanilla training method under practical poisoning attacks*: Thanks for the helpful comment. We note that in this paper, we focus on providing the certified robustness for RL algorithms with theoretical guarantees, which means our certification always holds as long as the attacks are under certain constraints (e.g., number of poisoning instances here), and it is agnostic to actual attack algorithms. Thus, as all the existing certified robustness work for different ML algorithms, it is important to compare the certified robustness for different learning algorithms while it is not very helpful here to perform empirical attacks [4,5].
>
> *Too low lower bounds in Figure 2*: We explain that the low lower bounds presented in Figure 2 is due to the short trajectory length $H=75$ adopted in our experiments. A normal episode in Breakout would contain thousands or tens of thousand steps, which gives a cumulative reward of over $300$. When the trajectory length is constrained to $75$, the achieved points will be low; but since our goal is to compare the relative certified robustness of different RL algorithms, such evaluation on short trajectory is sufficient with efficient computation cost. We additionally emphasize that the score achieved when the poisoning size $K=0$ is exactly the result under clean testing, and thus our successive bounds under larger $K$ are non-vacuous compared to this value.

---

> ### Author Response · Authors · 2021-11-21
> **Official Response to Reviewer 6rjC (2/4)**
>
> > 2. For the proposed algorithm
>
> **(1) Temporal continuity assumption**
>
> We thank the reviewer for the question and clarify that the temporal continuity assumption is in fact not uncommon in reinforcement learning, especially in model-based RL. For example, [1] leverages temporal continuity and interactive feedback to help with entity abstraction in scene modeling in real-world RL tasks, and [2] also relies on the temporal continuity of real-world stimuli (modeled as state variables) to extract salient information. In our work, we similarly leverage this property to increase robustness at bottleneck states. Moreover, our empirical results on Freeway imply that some RL tasks do exhibit such property and our TPARL and DPRAL work well under these tasks.
>
> **(2) COPA-Search algorithm**
>
> We are sorry about not making it clear in the paper. The offline RL setting in our paper refers to the *training* setting, i.e., the policy training is conducted on pre-collected offline RL datasets without accessing the environment, so the training procedure cannot just query the environment and ignore the poisoned trajectories. To evaluate and certify the learned policy, we follow the standard approach [3] which deploys the learned policy in an interactive online environment and tests its performance per episode. During testing, we do need to access the environment (otherwise there is no meaningful way to test), but importantly, we do not retrain the policies using the testing environment; the policies have been trained using poisoned data and are kept fixed. Particularly, Algorithm 5 is not for the learner--it is involved during testing, not during training. This testing setting also makes practical sense. For example, we first train an autonomous vehicle using a pre-collected large scale dataset, and then deploy the trained AV in the real world to test its performance. We hope this clarifies the reviewer’s concern. Please let us know if there is additional unclarity and we really look forward to further improving our paper based on the suggestions.

---

> > ### Comment · Reviewer_6rjC · 2021-11-24
> > **Still not convinced**
> >
> > Thank the authors for the detailed response. The revised paper looks much better now. But I still have concerns on the COPA-Search algorithm. If it is not designed for the trainer, then the certification does not make much sense. In my understanding, certification should be evaluated or computed by the learner, so that it knows the worst-case performance of itself. If we can only know the certification during test time, then certifying the method does not help us know how good the policy is before deploying it in a real-world environment.
> >
> > The AV example is also not very clear. How can you know the real dynamics P and R in practical environments? Testing an AV in a real world can be expensive and risky. It is realistic to "test" a policy in a real world just to get a sense of how bad it is.
> >
> > Therefore, I do not think this is a useful certification if it can only be evaluated during test time.

---

> > > ### Author Response · Authors · 2021-11-25
> > > **Response to the Follow-up of Reviewer 6rjC (2/2)**
> > >
> > > > *“The AV example is also not very clear. How can you know the real dynamics P and R in practical environments? Testing an AV in a real world can be expensive and risky. ”*
> > >
> > > We are sorry for the confusion. We provide more detailed explanations below.
> > >
> > > For the AV example, we aimed to provide a real-world example to motivate the setting of training using a large-scale pre-collected offline RL dataset, but by no means we aim to address the AV testing problem completely and we will make this clear in our revision. As we can see, offline RL training is widely studied [1,2,3], aiming to deploy a well trained policy to an unknown testing environment based on its generalization ability. Similarly, though we cannot prove the generalization of the certified robustness in new environments with different initial states as described above, we indeed aim to infer the certified robustness in real-world testing environments, just as current empirical evaluation on simulated environments to infer the safety in unknown real-world scenarios. In addition, our certification provides a relative robustness comparison between different RL algorithms with theoretical guarantees, and such relative robustness comparison should still hold for other testing environments with similar distributions. How to exactly certify or even empirically test the unknown real-world scenarios is a challenging and interesting direction, and we agree that empirical testing in real-world scenarios is definitely important, while we believe our work providing certified robustness evaluation for different RL algorithms in a given environment (which could be of similar distribution with the real one based on existing sim-to-real work) would also provide an important evaluation perspective. We will add these discussions in our revision and tune down the tone as we take the AV testing as an example but there are definitely lots of other challenges that need to be addressed.
> > >
> > > Thanks again for the insightful questions, and please let us know if you have further questions and concerns. Thanks!
> > >
> > > **References**
> > >
> > > [1] Levine, Sergey, et al. "Offline reinforcement learning: Tutorial, review, and perspectives on open problems." arXiv preprint arXiv:2005.01643 (2020).
> > >
> > > [2] Agarwal, Rishabh, Dale Schuurmans, and Mohammad Norouzi. "An optimistic perspective on offline reinforcement learning." International Conference on Machine Learning. PMLR, 2020.
> > >
> > > [3] Fu, Justin, et al. "D4rl: Datasets for deep data-driven reinforcement learning." arXiv preprint arXiv:2004.07219 (2020).
> > >
> > > [4] Rosenfeld, Elan, et al. "Certified robustness to label-flipping attacks via randomized smoothing." International Conference on Machine Learning. PMLR, 2020.​​
> > >
> > > [5] Weber, Maurice, et al. "RAB: Provable robustness against backdoor attacks." arXiv preprint arXiv:2003.08904 (2020).
> > >
> > > [6] Steinhardt, Jacob, Pang Wei Koh, and Percy Liang. "Certified defenses for data poisoning attacks." Proceedings of the 31st International Conference on Neural Information Processing Systems. 2017.
> > >
> > > [7] Jia, Jinyuan, Xiaoyu Cao, and Neil Zhenqiang Gong. "Certified robustness of nearest neighbors against data poisoning attacks." arXiv preprint arXiv:2012.03765 (2020).

---

> > > > ### Comment · Reviewer_6rjC · 2021-11-28
> > > > **Following up**
> > > >
> > > > Thank you for the further clarification. I understand that a strict and tight certification is not easy to obtain in offline RL. But the interpretation given by authors are still not very convincing to me.
> > > >
> > > > > The main difference between the certification setting suggested by the reviewer and ours boils down to the following question: Do we need to consider certification for a specific initial state of the agent during test time (this is our setting), or consider all possible worst case initial states in real-world scenarios (this is the reviewer’s setting)?
> > > >
> > > > 1. The certification is not only about initial state, but also about the dynamics of the MDP.
> > > >
> > > > 2. It is okay to make assumptions about the initial state distribution in the MDP. Why is it necessary to directly interacting with the environment?
> > > >
> > > > > Our paper studies a more realistic type of certification, which is also used by most previous certification works [4,5,6,7].
> > > >
> > > > It is not clear what is shared between this paper and the cited papers. I do not think the settings are similar. The proposed method requires interacting with the test-time environment, which is analogous to a supervised certification that only knows the worst-case accuracy after testing. In contrast, the bounds provided by the cited papers are more realistic. For example, the bound (5) in paper [4] is generic for any test point.
> > > >
> > > > > In COPA we can only provide certification for trajectories starting from a given initial state just like in standard image classification, the certification can only certify a given set of test images; and it is very challenging (and almost impossible) to provide certification for all possible real-world situations.
> > > >
> > > > This claim is not very clear and not precise. In many certification works, such as randomized smoothing (although it is for test-time robustness), the robust radius is agnostic to test data distribution, so it is not true that standard image classification can only certify a given set of test images. For poisoning attacks, it is true that no one can certify a totally out-of-distribution test dataset. However, in most cases, the test dataset and the training dataset come from the same distribution, so there exists certification for in-distribution test data points. For example, the bound in Levine et al. is not for a specific test set. In the studied RL problem, the test MDP is the same as the training-time MDP. So I do not agree that it is impossible to obtain a certificate. If the training data is sufficient and the amount of poison is limited, one can build an estimated MDP model with uncertainty intervals. Then it is easy to obtain a lower bound of the policy value in the real MDP. When data is insufficient, the bound may be loose, but at least should exist.
> > > >
> > > > ---
> > > >
> > > > In summary, the certification algorithm is not very practical, or there is not enough analysis on this problem in the paper. But based on the revised paper and author responses, some of my other concerns are addressed, so I have raised my score to 5.
> > > >
> > > > Alexander Levine, and Soheil Feizi. "Deep partition aggregation: Provable defense against general poisoning attacks."

---

> > > > > ### Author Response · Authors · 2021-11-29
> > > > > **Response to Following up of Reviewer 6rjC (2/2)**
> > > > >
> > > > > > *“If the training data is sufficient and the amount of poison is limited, one can build an estimated MDP model with uncertainty intervals. Then it is easy to obtain a lower bound of the policy value in the real MDP. When data is insufficient, the bound may be loose, but at least should exist.”*
> > > > >
> > > > > We thank the reviewer for proposing this potential approach. We do agree this can be a valid approach, although it can be even less practical than our algorithm because estimating the MDP dynamics for a realistic high dimensional environment (like Atari) using a reasonable amount of offline data is almost impossible. In fact, if this MDP can be estimated relatively easily, then offline RL would be as simple as online RL. We greatly appreciate the reviewer for providing this insight and we will discuss it in our revision. Additionally, an advantage of our approach is that we can provide a *deterministic bound*, while using an estimated MDP one can only obtain a *probabilistic* bound. Sure enough, if we directly obtain the approximated MDP model from the training set, it would serve more functionalities; but our proposed approach is more targeted to solving the problem of providing cumulative reward lower bound certification given a specific initial state. The estimation and generalization of MDP given enough training data would be an interesting future direction and we will add related discussion in our revision.
> > > > >
> > > > > ---
> > > > > Nevertheless, we agree with the reviewer that existing certification methods, including ours, still have many limitations and the problem of giving worst-case performance guarantees under any real-world inputs without any test environment or instances remains open. However, we do believe our work is an important step towards giving certification for offline RL agents. The discussions with you are very helpful and insightful but we feel it can be inappropriate to reject our paper due to the limitations which are commonly shared by existing works and very challenging to solve in a single paper.
> > > > >
> > > > > We hope our responses clarify the reviewer’s concerns and are happy to discuss further. Thank you again for the inspiring questions!

---

> > > > > > ### Comment · Reviewer_6rjC · 2021-11-29
> > > > > > **Clarification**
> > > > > >
> > > > > > Thank you for the further clarification. But I would like to clarify my concern and my suggestions as below.
> > > > > >
> > > > > > > Their certifications require a given set of test data points and are not agnostic to test data distribution.
> > > > > >
> > > > > > Thank you for the clarification. I understand this point, but my last response may not be described precisely. Let me rephrase my concern: existing certification usually have a closed-form expression of the lower bound, whose values depend on properties of the test points. The closed-form solution makes it easier for analyzing the robustness of a defense and for comparing different certification methods. However, the proposed certification in this paper is found by an algorithm. Then, when there are other defending algorithms, how can people compare the certifications provided by different methods?
> > > > > >
> > > > > > Moreover, I am wondering why there is a need for a specific algorithm if we already know a policy and MDP dynamics in the test time. It is not hard to directly compute the expected return of any policy in a known MDP. Can you justify the challenges and the necessarity of designing such a worst-case searching algorithm?
> > > > > >
> > > > > > > We do agree this can be a valid approach, although it can be even less practical than our algorithm because estimating the MDP dynamics for a realistic high dimensional environment (like Atari) using a reasonable amount of offline data is almost impossible.
> > > > > >
> > > > > > First, what I propose not not necessarily the best solution. There could be other methods that do not require that much data. Second, I think COPA-SEARCH is less realistic than the MDP estimation method I described, because COPA-SEARCH requires to know the transition probability P. Isn't it a more unrealistic assumption in Atari games?

---

> > > > > > > ### Author Response · Authors · 2021-11-29
> > > > > > > **Response to Clarification of Reviewer 6rjC**
> > > > > > >
> > > > > > > We thank the reviewer for the reply and provide more clarifications below.
> > > > > > >
> > > > > > > > *”The closed-form solution makes it easier for analyzing the robustness of a defense and for comparing different certification methods. However, the proposed certification in this paper is found by an algorithm.”* and *”how can people compare the certifications provided by different methods?”*
> > > > > > >
> > > > > > > Our certification algorithm can be seen as a blackbox function with the policy, initial states, and an environment as the inputs, and it produces a per-instance certificate for the given initial state.
> > > > > > > A closed-form cannot be practically obtained because the environment transition $P$ is usually not explicit (e.g., in Atari environment it is almost impossible to give $P$ explicitly). Instead, we must interact with the environment (e.g., the game simulator) to compute the certification. However, given fixed inputs (policy, initial states, and environment) to our algorithm, our algorithm produces deterministic results, and we believe it should be comparable to different certification methods.
> > > > > > >
> > > > > > > In addition, previous works in the classification setting involve only *one-step prediction*; while our work studies a sequential decision problem where the decision at each step will influence the future trajectories, and such influence is jointly determined by the *policy* and the *environment*. When only considering the *policy* (just as in per-state certification in our Section 4), we can also provide closed-form solutions (Theorems 8, 1, and 3). But when the *environment* is also involved (which is inevitable in certifying the cumulative rewards for the trajectories), we have to rely on the interactions with the environment (since we do not know the ground truth MDP), and therefore have to run the search algorithm to account for the cumulative influences of the *policy* and *environment* in this multi-step decision problem.
> > > > > > >
> > > > > > > We further clarify that whether a certification has closed-form solution or not does not influence whether it can be compared with other certification methods or not. The point is, the certification is computed on a per-instance basis; as long as the certification is empirically computable following either a closed-form expression or a practical algorithm, it is comparable with other methods.
> > > > > > >
> > > > > > > > *”It is not hard to directly compute the expected return of any policy in a known MDP”* and *” Can you justify the challenges and the necessarity of designing such a worst-case searching algorithm?”*
> > > > > > >
> > > > > > > We would like to clarify that we do not know the *exact policy* that will be evaluated due to the *unknown poisoning* on the training dataset, so it is impossible to *“directly compute the expected return”* of such unknown policies even *“in a known MDP”*. The way we address this problem is through considering the *worst-case* influence of the poisoning on the policy, i.e., what are the possible actions (Possible Action Set, Definition 5) given certain magnitudes of poisoning. We present the results for such worst-case influences in Theorems 4, 5, and 7 in Section 3.4, and leverage these theorems to expand the worst-case tree branches in line 19 Algorithm 5 (COPA-Search). Since we want to provide a deterministic lower bound for the cumulative reward of **all possible policies** under poisoning, we must explicitly consider the worst-case influence per step and devise a search algorithm based upon it.
> > > > > > >
> > > > > > >
> > > > > > > > *”COPA-SEARCH requires to know the transition probability P. Isn't it a more unrealistic assumption in Atari games?”*
> > > > > > >
> > > > > > > We clarify that we do not need to know the groundtruth transition *probability* $P$; instead, we only need to interact with the environment, so that we can follow the transition *function* as a **feedback** from the environment. This is a *standard* assumption in Atari games, and our evaluations in Section 4 are all conducted on the standard platform of OpenAI Gym which only allows the interaction but does not provide the groundtruth transition probability $P$. In fact, it is almost impossible to explicitly write the full transition probability $P$ for a complex environment like Atari, so practically, we interact with the environment and observe the actual transition for a particular trajectory. This is also done by most modern DRL algorithms on Gym environments, which do not require an explicit formulation of $P$ but learn from interactions with the environment.
> > > > > > >
> > > > > > > ---
> > > > > > > Again, we thank the reviewer for the swift reply and are happy to discuss more!

---

> > > > > > > > ### Comment · Reviewer_6rjC · 2021-11-30
> > > > > > > > **thank you for the clarification**
> > > > > > > >
> > > > > > > > Thank you for the clarification. I indeed had some misunderstanding on the algorithm requirements. Algorithm 5 says $s_{t+1}\leftarrow P(s_t,a_t)$, which looks like you require to know the exact $P$ function. So I would suggest not to write it in this way. It is good to know that the environment dynamics are not required. However, this leads to a new concern that this algorithm only applies to deterministic MDPs and is not easy to extend to stochastic MDPs for getting a lower bound.
> > > > > > > >
> > > > > > > > In addition, the experimental results are still not very convincing as the derived bounds are too low. Even if it is the first proposed certification, if it does not improve the native lower bound (the lowest possible reward in the environment), it is not very helpful in practice. Therefore, I tend to maintain my score for now.

---

> > > > > > > > > ### Author Response · Authors · 2021-11-30
> > > > > > > > > **Response to the further comment of Reviewer 6rjC**
> > > > > > > > >
> > > > > > > > > We thank the reviewer for the prompt feedback and we are glad that our clarifications were effective. We thank the reviewer’s suggestion for rewriting the Algorithm 5 and will definitely do that in our revision to avoid future confusion. However, we would like to further clarify that **it is a complete misunderstanding that our method does not improve the native lower bound**; maybe some numbers are misinterpreted here and we provide details below.
> > > > > > > > >
> > > > > > > > > > *“the experimental results are still not very convincing as the derived bounds are too low.*” and “*if it does not improve the native lower bound (the lowest possible reward in the environment), it is not very helpful in practice”*
> > > > > > > > >
> > > > > > > > > The seemingly low lower bounds presented in Figure 2 is due to the relatively short trajectory length (400 for Freeway and 75 for Breakout) we adopted in our experiments. Under the same trajectory length, **a normal agent under clean testing achieves a reward of 5 for Freeway and 2 for Breakout**, not 300 as mentioned by the reviewer; this is shown as the score with poisoning size $K=0$ in Figure 2, and thus our successive bounds under larger $K$ are non-vacuous compared to this value. We did not evaluate on the full episode length (up to tens of thousands steps), since our goal is to compare the relative certified robustness of different RL algorithms, and such evaluation on short trajectory is sufficient with reasonable computation cost. We will further clarify this in our final revision.
> > > > > > > > >
> > > > > > > > > > *"However, this leads to a new concern that this algorithm only applies to deterministic MDPs and is not easy to extend to stochastic MDPs for getting a lower bound."*
> > > > > > > > >
> > > > > > > > > We thank the reviewer for the suggestion of considering the case of stochastic MDPs. We will discuss the potential way to extend our COPA-Search algorithm to stochastic MDPs. In the current version of COPA search, the exhaustive search is enabled by the deterministic MDP assumption; without this assumption, we can leverage *sampling* (i.e., by repeatedly taking the same action at the same state) to obtain the set of high probability next state transitions with high confidence. In this way, our COPA-Search will be able to yield a *probabilistic bound*. Since we are the first certification work on poisoning attacks for offline RL agents, we cannot cover all these aspects in a single paper and they would be interesting future directions.
> > > > > > > > >
> > > > > > > > > ---
> > > > > > > > >
> > > > > > > > > We hope our response addresses your concern especially on the seemingly impractical lower bound, which was caused by a misunderstanding. Please kindly let us know if you have any further concerns, thank you.

---

> > > > > ### Author Response · Authors · 2021-11-29
> > > > > **Response to Following up of Reviewer 6rjC (1/2)**
> > > > >
> > > > > We thank the reviewer for the follow-up comments as well as the encouraging score increase, and we would like to make further clarifications on the following points. Especially, from the comments there may be some misunderstanding on existing certification methods such as randomized smoothing: their certifications require a *given set of test data points* and are not agnostic to test data distribution.
> > > > >
> > > > > > *“1. The certification is not only about initial state, but also about the dynamics of the MDP.”*
> > > > >
> > > > > We further explain our analogy to the classification setting. In classification, the certification is provided for certain *given* test points, rather than *any test data*; in RL, comparably, the instance we certify is not a single point, but a whole trajectory. A trajectory is determined by its *initial state* and the *environmental dynamics* (when the dynamics and the policy are deterministic, as stated in our assumption). Thus, certifying the RL algorithm in our case can be viewed as certifying a given *initial state* and the MDP dynamics have been considered in our tree search algorithm, which is analogous to certifying a *given* test data in classification case. We did not include the MDP dynamics in our original analogy because it is the same regardless of initial states, and only the initial state can be changed just as input test images in classification. Maybe it is more clear to say “a trajectory” rather than just “an initial state” here and we will make it clear in the revision discussion.
> > > > >
> > > > > > *“2. It is okay to make assumptions about the initial state distribution in the MDP. Why is it necessary to directly interacting with the environment?”*
> > > > >
> > > > > Thank you for the suggestion. As explained above, our certification algorithm is “point-wise” (per initial state), like many existing certification algorithms. So an assumption of initial state distribution is not directly helpful, and it is also hard to make a reasonable assumption for a complex environment. The interaction is needed to know the states and rewards in a trajectory. Using a specific test trajectory is indeed a limitation of our certification, but it is also shared by most existing certification approaches (e.g., in randomized smoothing, one must use a specific test image, add noise to it and sample the classifiers multiple times to get a point-wise certificate for that specific test image).
> > > > >
> > > > > > *“The proposed method requires interacting with the test-time environment, which is analogous to a supervised certification that only knows the worst-case accuracy after testing.”*
> > > > >
> > > > > This analogy is correct, and actually it is the same situation in standard supervised certification without any conflict: previous works must run certification on a specific test set (with a finite number of known test instances, which can be seen as the “test environment”), and **the worst-case accuracy on this specific test set is only known after testing**. Without a specific test set, existing work such as [4] also cannot give any certification.
> > > > >
> > > > > > *“the bound (5) in paper [4] is generic for any test point”* and *“in randomized smoothing, the robust radius is per-test sample based and is agnostic to test data distribution, so it is not true that standard image classification can only certify a given set of test images.”*
> > > > >
> > > > > We clarify that bound (5) in paper [4] is of similar flavor as our obtained lower bounds. Bound (5) is not providing a generic bound for *any* test instance; instead, it is also applied on *per* test instance (i.e., you have to run their algorithm for every instance in the test set), and the criterion relies on the information $p$ (i.e., the majority vote) of the given instance.
> > > > >
> > > > > A similar restriction applies to randomized smoothing: its certification is also **test data dependent**, and the robust radius is point-wise and not for a test distribution. (That’s also why randomized smoothing randomly samples 500 test data in ImageNet to perform certification.) Similarly, our work also provides a per test instance bound, where the test instance here means a specific trajectory (determined by its initial state and environment dynamics).
> > > > >
> > > > > > *“For example, the bound in Levine et al. is not for a specific test set. In the studied RL problem, the test MDP is the same as the training-time MDP.”*
> > > > >
> > > > > We thank the reviewer for bringing this interesting point. We believe that our COPA-search shares the same utility as the bound in (Levine et al.): Following the reviewer’s view that the bound in (Levine et al.) as “not for a specific test set”, our COPA-search is not for a specific MDP --- we can also apply our COPA-search to certify the poisoned policy’s worst-case performance on any given test-time MDPs, as long as interaction with that MDP is allowed (same as access to clean test image is allowed in Levine et al.).

---

> > > ### Author Response · Authors · 2021-11-25
> > > **Response to the Follow-up of Reviewer 6rjC (1/2)**
> > >
> > > We thank the reviewer for appreciating our response and the revision, as well as the thoughtful follow-up. Below, we provide more clarifications for the reviewer’s concern on the assumption for our COPA-Search algorithm.
> > >
> > > > *”If we can only know the certification during test time, then certifying the method does not help us know how good the policy is before deploying it in a real-world environment.”*
> > >
> > > The certification setting suggested by the reviewer is a valid but much more difficult type of certification which can be impractical in many scenarios. Our paper studies a more realistic type of certification, which is also used by most previous certification works [4,5,6,7].
> > >
> > > The main difference between the certification setting suggested by the reviewer and ours boils down to the following question: Do we need to consider certification for a specific initial state of the agent during test time (this is our setting), or consider all possible worst case initial states in real-world scenarios (this is the reviewer’s setting)?
> > >
> > > As an analogy for the image classification setting (which people are probably more familiar with): do we consider certification for specific images in a test set (as an analogy to our setting), or do we consider certification for any worst case input image without accessing the test set?
> > >
> > > The second type of certification (as suggested by the reviewer) is often very hard, since the number of possible initial states (or input images in the classification setting) is infinite. And since it is almost impossible for an agent (or a classifier) to always work correctly (e.g., 100% accuracy) under any initial states (or any input images, even nonsense ones), the second type of certification is almost impractical.
> > >
> > > Our certification algorithm needs to start from a **given initial state** at test time, and we give the certified reward **under any possible trajectories** that starts from this initial state and **under any worst case poisoned agent policy**. We acknowledge that it is a limitation of our certification method, but it is also shared by almost all existing works on certification. In COPA we can only provide certification for trajectories starting from a given initial state just like in standard image classification, the certification can only certify a given set of test images; and it is very challenging (and almost impossible) to provide certification for all possible real-world situations. We will discuss this limitation in our paper.
> > >
> > > To tackle the challenge mentioned by the reviewer (“know how good the policy is before deploying it in a real-world environment”), one must know the distribution of initial states (or input image distributions in the classification setting) to exclude the potentially nonsense inputs, and certify against all possible inputs from the specified input distribution. This problem is largely open even for the simpler setting of image classification - in fact, even modeling the distribution of real-world inputs is hard, and it is beyond the scope of our paper. We leave it as a future work.

---

> ### Author Response · Authors · 2021-11-21
> **Official Response to Reviewer 6rjC (1/4)**
>
> We thank the reviewer for the valuable comments and suggestions. We apologize for the confusions in our paper; especially, the biggest concern on accessing the environment during learning is a misunderstanding. We address the comments below.
>
> > 1. For writing
>
> **(1) Omitted details and definitions**
>
> We thank the reviewer for the helpful comment. We will adjust our structure and move some important details and definitions back to the main paper to make the flow of the paper smooth, and we will only focus on the analysis of one protocol (DPARL) and put the related details of other protocols in appendix.
>
> **(2) Separately discussing 3 aggregation protocols**
>
> We thank the reviewer for the insightful comment. We clarify the motivation for separately discussing three aggregation protocols below. The three protocols are developed progressively considering additional flexibility based on the previous one. We will make the discussion clear in our final version.
>
> First of all, PARL is a relatively direct extension of the deep partition aggregation method [1] in supervised learning which only performs aggregation on the sub-policy level. Then, TPARL takes into consideration the sequential decision making nature of RL and leverages the temporal continuity to increase the tolerable poisoning threshold by additionally aggregating on the temporal level. Furthermore, DPARL circumvents the problem of requiring prior knowledge regarding accurate specifications for the window size $W$ at each time step by using dynamically computed window size per time step. We emphasize that each protocol intends to address the shortcomings of the previous one. Although the reviewer pointed out that the three methods provide empirically similar results in Figures 1 and 2, we clarify that protocols with temporal aggregation (TPARL and DPARL) are overall better than the single-step aggregation PARL, and we may expect that DPARL will provide better certification for games where the variance of window size is much larger across time steps.
>
> We thank the reviewer for the valuable suggestion, and will adjust our presentation to focus more on one protocol (DPARL) with more details in the main paper for higher clarity and readability.
>
> **(3) Unclear notations**
>
> We are sorry about the unclear notations and we thank the reviewer for pointing them out. We provide clarifications below and will revise our paper accordingly to make them clear.
>
> *$K_t$*: The reviewer’s understanding is correct. It refers to the tolerable poisoning threshold at time step $t$. We mentioned $K_t$ in the last sentence in the paragraph below Definition 1. In revision, we made it more clear by involving the definition directly in Definition 1.
>
> *Theorem 1*: We believe Equation 3 is mathematically strict, and we improve the clarity by reformulating $\arg\max_{\sum_{i=1}^p h_{a,a’}^{(i)} \le \delta_{a,a’}} p$ by $\max \\{ p: \sum_{i=1}^p h_{a,a’}^{(i)} \le \delta_{a,a’} \\}$. As you can see, $p$ parameterizes the constraint and it is what we are maxing over. It corresponds to the maximum tolerable number of flipped sub-policies. We are sorry about the confusion and have updated our Equation 3 in our revision to reflect the changes.

---

### Official Review · Reviewer_xuEG · 2021-11-02

**Correctness:** 3
**Technical Novelty And Significance:** 3
**Empirical Novelty And Significance:** 3
**Recommendation:** 6
**Confidence:** 4

**Main Review:**

Strengths:

1. The paper is well-written. It clearly states its main contributions and the intuition of the techniques used in  COPA.

2. The paper proposes a certification framework against poisoning attacks in deep reinforcement learning, which is non-trivial. It advocates two certification criteria: per-state action stability and lower bound of cumulative reward.

3. For each certification criteria, it provides bounds for different proposed aggregation protocols.

4. In addition to theoretical results, the paper also presents numerical results.


Weaknesses:

1. I appreciate the authors' providing numerical results and the theoretical studies. However, it is unclear to me to what extent the introduction of the proposed method degrades the training of the RL algorithms (in terms of convergence speed and policy quality).  In particular, how does the proposed method perform compared with vanilla DRL algorithms when there are no adversarial attacks?

2. While the paper claims that "there is no robust RL method that is able to provide practically computable certified robustness against poisoning attacks,"  it might not be accurate.  There are a few related works on certified robustness in DRL.  For instance, "Certifiable Robustness to Adversarial State Uncertainty in Deep Reinforcement Learning" by Everett et al., and "CROP: Certifying Robust Policies for Reinforcement Learning through Functional Smoothing" by Wu et al..  Though the exact settings could be different, the authors might want to compare the proposed method with the existing literature.


**Summary Of The Paper:**

This paper proposes a certification method against poisoning attacks in offline reinforcement learning, where attackers can manipulate a subset of the training trajectories. It presents two certification criteria: per-state action stability and cumulative reward bound. It also introduces different aggregation protocols to train the policies and provides some bounds regarding the certification. In addition to theoretical results, it also has ablation studies to identify the implications of different parameters.

**Summary Of The Review:**

Solid work with minor issues that can be fixed.

---

> ### Author Response · Authors · 2021-11-21
> **Official Response to Reviewer xuEG (2/2)**
>
> **References**
>
> [1] Everett, Michael, Bjorn Lutjens, and Jonathan P. How. "Certifiable Robustness to Adversarial State Uncertainty in Deep Reinforcement Learning." IEEE transactions on neural networks and learning systems.
>
> [2] Wu, Fan, et al. "CROP: Certifying Robust Policies for Reinforcement Learning through Functional Smoothing." arXiv preprint arXiv:2106.09292 (2021).
>
> [3] ​​Weng, Lily, et al. "Towards fast computation of certified robustness for relu networks." International Conference on Machine Learning. PMLR, 2018.
>
> [4] Cohen, Jeremy, Elan Rosenfeld, and Zico Kolter. "Certified adversarial robustness via randomized smoothing." International Conference on Machine Learning. PMLR, 2019.
>
> [5] Salman, Hadi, et al. "Provably robust deep learning via adversarially trained smoothed classifiers." Proceedings of the 33rd International Conference on Neural Information Processing Systems. 2019.

---

> ### Author Response · Authors · 2021-11-21
> **Official Response to Reviewer xuEG (1/2)**
>
> We thank the reviewer for the constructive comments and we provide our responses below.
>
> > 1. Does the proposed method degrade the training of the RL algorithms?
>
> We thank the reviewer for the insightful question. We provide additional experimental results on the comparison between the empirical performance of our proposed partition-based training and the standard training in terms of the *convergence speed* and *policy quality* following the suggestion. The results are in Figure 3 and Table 5 in Appendix H.2 in our revision.
>
> In Figure 3, we aim to show the comparison of the convergence speed. For *our proposed training*, we plot all training curves of the sampled $5$ subpolicies in blue, where each subpolicy is trained on one partition of the dataset. (We do not plot all $50$ curves for visual clarity). For *standard training*, we plot the training curve of the standard policy in red, where the single policy is trained on the entire dataset. We see that on **Freeway**, most subpolicies converge slower than the standard policy, but will reach similar convergence value as the standard training one; while there also exist a few subpolicies that fail to be trained. On **Breakout**, all subpolicies converge slower and reach lower convergence value compared with the standard policy. The degradation of single subpolicy quality is reasonable given that 1) the size of the training set of each subpolicy (roughly 5M) is much smaller than that of the standard policy (250 M); 2) Breakout is more challenging than Freeway. This result also provides insights into the selection of partition number $u$ which controls the trade-off between the subpolicy quality and the upper bound of the tolerable poisoning threshold.
>
> In Table 5, we aim to compare the policy quality of the aggregated policy derived in our COPA framework (i.e., PARL, TPARL, and DPARL) with the policy obtained from standard training. We see that in **Freeway**, our three protocols achieve comparable results with the policy obtained by standard training on the entire dataset; on **Breakout**, although the quality of our obtained policies is lower, our policies are much more stable with significantly lower variance than the standard training. Overall, we see that the proposed method achieves similar performance with DRL algorithms without considering adversaries since we only perform aggregation without adding strong regularization or anything operation that hurts the training.
>
> > 2. Comparisons with related work on certified robust RL
>
> We thank the reviewer for pointing out the interesting related works [1,2], and we provide concrete comparisons between our work and [1,2] below. We have updated Appendix I.2 and Appendix I.3 of the revision to include these references and discussions.
>
> First of all, our work is different from [1,2] in several major aspects in the problem setup. In terms of the **threat model**, [1,2] focus on certifying robust RL algorithms against *evasion attacks* during test time, while we aim to provide certifiably robust RL against *poisoning attacks* during training time where the attacker is more capable. Second, in terms of the **learning setting**, our work is set up against the *offline RL* setting which learns from the pre-collected offline dataset, compared to the *online RL* setting in [1,2] which performs data collection and policy training simultaneously.
>
> We next provide more detailed comparisons with each work.
>
> Everett et al. [1] aims to compute the lower bounds of action-values under potential perturbation and select actions according to the worst-case bound, but it relies on linear bounds [3] and is only suitable for low-dimensional environments. Moreover, although the authors associate robustness with per-state action selection, their method in essence only seeks to improve robustness, but cannot provide any practical certification. In contrast, our work not only proposes two clear criteria for robustness (i.e., per-state action stability and cumulative reward lower bound), but also provides practical methods to compute the certification.
>
> Wu et al. [2] provides robustness certification for RL against *test-time evasion attacks* following the line of work on randomized smoothing [4,5]. Concretely, they apply per-state smoothing to achieve the certification for per-state action stability, as well as trajectory smoothing to obtain the certification for cumulative reward. In comparison, robustness in their work is derived from *smoothing*, while our robustness comes from *aggregation*. We propose aggregation protocols and certification methods that leverage the temporal information by dynamically aggregating over the past time steps, which is not covered in [2].
>
> To summarize, neither [1] nor [2] provide practically computable certified robustness against poisoning attacks in RL, and to our best knowledge, our COPA framework is the first that achieves this goal.

---

### Official Review · Reviewer_RQX2 · 2021-11-08

**Correctness:** 3
**Technical Novelty And Significance:** 4
**Empirical Novelty And Significance:** 3
**Recommendation:** 6
**Confidence:** 2

**Main Review:**

Generally, the paper was well-written, although there are some organisational problems I would like to see addressed. As the authors mention, although poisoning attacks are major concern in practical settings, one may argue they are under-researched compared to test-time attacks. As far as I am aware, this is the first paper that proposes methods to certify an RL algorithm is robust to poisoning attacks, and the results look convincing. I will go through each of my main concerns / questions / misunderstandings below in no particular order:


1. What is the motivation for developing three separate aggregation protocols? From the experiments, TPARL and DTPARL provide similar certification results. I would suggest the authors choose to present one protocol and defer the other two to the appendix, it would dramatically improve the readability of the work.


2. As far as I understand, the number of sub-policies, *u*,  that need to be trained needs to be *at least* as large as the number of poisoned action-states / trajectories, *K*. Given that in offline RL one has a finite dataset to learn a policy on, what is the affect of varying *u* on the quality of these learned sub-policies? The certification method is dependent on some level of agreement between sub-policies, but if *u* needs to be very large will this not increase sub-policy variance? It seems from Appendix H.1 this is not true for *u*=30, 50, but it would be useful to show the utility and certification over a larger sweep of *u*.


3. TPARL and DTPARL are based on the insight that not all states are created equal, and bottleneck / critical states are more important from a poisoning perspective. How much variance in window size is there for DPTARL? I'm interested in this because although the worst-case complexity is based on *W_max* should one not expect average certification time to be much smaller if the average window size is << *W_max*?


4. Related to (3), it is claimed that certification time complexity *per state*  "in practice adds negligible overhead compared with standard network policy inference.". Maybe I missed this but I couldn't see any empirical results that back up this claim. The complexity looks like a bottleneck to me, being quadratic in the space of actions and linear in *u*. The latter is perhaps to be expected since this essentially says that if one wants to certify a policy is robust to a large adversarial poisoning budget, it will take more time than certifying against a smaller budget.


5. The COPA-search algorithm is not clearly presented. Could the authors comment of the time-complexity of the search as I had trouble understanding this part.


6. Could the authors comment on why QR-DQN and C51 are more certifiably robust than DQN? Initially I thought this may just be because they achieve higher utility but this is not the case (according to Appendix H.1)


7.  How were the trajectory lengths chosen in Section 4.2?


8. What is the intuition behind why Breakout is less stable than Freeway? Does it contain more critical / bottleneck states? It would have been very useful to perform a deeper analysis of robustness to poisoning attacks wrt bottleneck and non-bottleneck states.

9. Could the authors comment on the relation between this work and [1]? Although the goals are different it feels like there is a surprising amount of overlap for an uncited paper: the name and organisation are similar, algorithms for per-state and cumulative reward certification are given, and are evaluated on similar datasets.

[1] Wu et al. CROP: Certifying Robust Policies for Reinforcement Learning through Functional Smoothing. 2106.09292

**Summary Of The Paper:**

This paper studies the problem of certifying a policy learned via an offline RL algorithm is robust to poisoning attacks. The authors propose two certification criteria: per-state action stability and lower bound of cumulative reward. Three different protocols are proposed, although they all based on ensembling / aggregation of sub-policy decisions. For the certification of cumulative reward, a tree-based search approach is given that evaluates all possible actions within a set. Experiments on Freeway and Breakout for a few different offline RL algorithms are given, showing Freeway can be certified to be robust to poisoning attacks for a large poisoning threshold, while Breakout cannot.



**Summary Of The Review:**

Overall, I think this work represents a solid contribution to the field of robustness certification and RL, against what some may consider a more practical adversary than in evasion attack settings. There are a number of subtle algorithmic and experimental points that I would like the authors to clarify before raising my score, such as the complexity of COPA-search, and a more in-depth analysis of robustness to poisoning critical / bottleneck states over non-bottleneck states. I feel like this is a key point to address as it will provide stronger evidence of the deltas between the different aggregation protocols.

---

> ### Author Response · Authors · 2021-11-21
> **Official Response to Reviewer RQX2 (4/4)**
>
>
> **References**
>
> [1] ​​Levine, Alexander, and Soheil Feizi. "Deep Partition Aggregation: Provable Defenses against General Poisoning Attacks." International Conference on Learning Representations. 2020.
>
> [2] Hendrycks, Dan, et al. "Natural adversarial examples." Proceedings of the IEEE/CVF Conference on Computer Vision and Pattern Recognition. 2021.
>
> [3] Ye, Weirui, et al. "Mastering Atari Games with Limited Data." Advances in Neural Information Processing Systems 34 (2021).
>
> [4] Veerapaneni, Rishi, et al. "Entity abstraction in visual model-based reinforcement learning." Conference on Robot Learning. PMLR, 2020.
>
> [5] Legenstein, Robert, Niko Wilbert, and Laurenz Wiskott. "Reinforcement learning on slow features of high-dimensional input streams." PLoS computational biology 6.8 (2010): e1000894.
>
> [6] Wu, Fan, et al. "CROP: Certifying Robust Policies for Reinforcement Learning through Functional Smoothing." arXiv preprint arXiv:2106.09292 (2021).

---

> ### Author Response · Authors · 2021-11-21
> **Official Response to Reviewer RQX2 (3/4)**
>
> > 8. The reasons why Breakout is less stable than Freeway and a deeper analysis of robustness
>
> **Does Breakout contain more critical / bottleneck states?**
>
> In Breakout, the game goal is to control the paddle so that the ball is bounced to hit the brick. There are clearly very different stages in the Breakout game, e.g., when the ball is flying towards the paddle and when it is bounced back. An example of the *bottleneck state* is when the ball is approaching the paddle. The poisoning behavior may lead to the disastrous effect that the policy learns to control the paddle to slide in the opposite direction of the ball at such bottleneck states, while other states may not be as vulnerable. In comparison, in Freeway, the game goal is to control the agent to cross the road while avoiding the traffic. The agent is faced with similar traffic conditions everywhere on the road, and can make stable choices regardless of the complex situation. Thus there are very few *bottleneck states*. We have added such discussion about different game properties in Appendix H.5 following the suggestion.
>
> **Analysis of robustness to poisoning attacks w.r.t. bottleneck and non-bottleneck states**
>
> According to our illustration in Section 3.2, bottleneck states are those where sub-policies vote for diverse actions but the best action is the same as previous states, and non-bottleneck states are those where subpolicies mostly vote for the same action.
>
> It is a bit too abstract to analyze robustness in bottleneck and non-bottleneck states since the abstract vote assumptions are needed. Therefore, we analyze the robustness via a concrete example in Appendix B. In Appendix B, $s_0$ to $s_6$ are non-bottleneck states and $s_7$ is a bottleneck state. As we can see, with our TPARL and DPARL, for bottleneck state $s_7$, the tolerable poisoning threshold is improved from 0 to 1 or 2. For non-bottleneck states $s_0$ to $s_6$, we can easily find that three protocols will lead to similar tolerable poisoning thresholds. We explicitly state bottleneck/non-bottleneck states in Appendix B in revision.
>
> We provide additional discussions regarding why bottleneck states are vulnerable and why our temporal aggregation strategy can effectively alleviate the problem in revision Appendix B. We first explain the existence of the bottleneck states. Given the property of the bottleneck states that there is high disagreement among different sub-policies on such states, they may lie close to the decision boundary. This may be a result of poisoning, or simply due to the intrinsic difficulty of the state. On the other hand, such states naturally exist in the rollouts (like natural adversarial examples [2]), and may induce high instability of the rollouts during testing. We next discuss additional intuitions for our temporal aggregation. Essentially, temporal aggregation effectively leverages the adjacent non-bottleneck states to rectify the decisions at the bottleneck states, based on the assumption of temporal continuity which has been revealed and utilized in several previous works [3,4,5]. We have included the above discussions in Appendix B in revision and thanks for the helpful suggestion to help us improve the paper.
>
> > 9. Relationship between our work and Wu et al. [6]
>
> We thank the reviewer for the helpful comment. We clarify that our work is different from [6] in several aspects. First, in terms of the **threat model**, [6] focuses on certifying robust RL algorithms against *evasion attacks* during test time, while we aim to provide certifiably robust RL against *poisoning attacks* during training time. Second, in terms of the **learning setting**, our work is set up against the *offline RL* setting which learns from the pre-collected offline dataset, compared to the *online RL* setting in [6] which performs data collection and policy training simultaneously. Moreover, the **sources** for robustness of the two work are also different. The robustness of [6] is derived via *smoothing*, while our robustness arises from *aggregation*. We propose aggregation protocols and certification methods that leverage the temporal information by dynamically aggregating over the past time steps, which is orthogonal with techniques in [6]. Regarding the adaptive search algorithm for certifying the cumulative reward lower bound, our COPA-Search is inspired from [6] but with major distinction that COPA-Search can derive deterministic guarantee against poisoning attacks, in contrast to the probabilistic guarantee against test-time state perturbations in [6]. We have updated our Section 3.4, Appendix E.3, and Appendix I.3 to include the reference and relation discussions following the suggestion.

---

> ### Author Response · Authors · 2021-11-21
> **Official Response to Reviewer RQX2 (2/4)**
>
> > 5. Time complexity of the COPA-Search algorithm
>
> We thank the reviewer for pointing this out. We first provide a more detailed explanation of the COPA-Search algorithm below and in our revision **Appendix E.3**, followed by the analysis on the time complexity.
>
> At a high level, COPA-Search *exhaustively* explores new trajectories leveraging the theorems (Theorem 4, 5, and 7) for computing the Possible Action Set (Definition 5) at each time step, and *effectively* updates the lower bound of cumulative reward by expanding a trajectory tree dynamically. The algorithm returns a collection of pairs {$(K’,J_{K’})$}  sorted in ascending order of $K’$. For any $(K’,J_{K’})$ in the collection, our algorithm ensures that as long as the poisoning size is no larger than $K’$, the cumulative reward will be lower bounded by $J_{K’}$.
>
> Concretely, COPA-Search alternately executes the procedure of *trajectory exploration and expansion* and *poisoning threshold growth*. In *trajectory exploration and expansion*, COPA-Search organizes all possible trajectories in the form of search tree and progressively grows it. For each node (representing a game state), we leverage Theorems 4, 5, and 7 to compute the Possible Action Set. We then expand the tree branches corresponding to the actions in the derived set. In *poisoning threshold growth*, when all trajectories for the current poisoning threshold are explored, we increase $K’$ to seek for certification under larger poisoning sizes, via maintaining a priority queue of all poisoning sizes during the expansion of the tree. The iterative procedures end when the priority queue becomes empty.
>
> We next provide an analysis on the time complexity of the algorithm. The computation complexity of COPA is $O(H |S_{\mathrm{explored}}| \times (\log |S_{\mathrm{explored}}| + |A| T))$, where $|S_{\mathrm{explored}}|$ is the number of explored states throughout the search procedure, which is no larger than cardinality of state set, $H$ is the horizon length, $|A|$ is the cardinality of action set, and $T$ is the time complexity of per-state action certification. The main bottleneck of the algorithm is the large number of possible states, which is in the worst case exponential to state dimension. However, to provide a deterministic worst-case certification agnostic to game properties, exploring all possible states may be inevitable theoretically. We have included this analysis in Appendix E.3 in the revision.
>
> > 6. The reasons why QR-DQN and C51 are more certifiably robust
>
> We thank the reviewer for the thoughtful question. We first provide our understanding on why QR-DQN and C51 are more certifiably robust, and then explain the potential relationship between benign performance in Appendix H.1 and the certified robustness. As concretely described in Appendix G.1, QR-DQN and C51 are two distributional RL algorithms. Instead of learning the mean action value $Q^\pi(s,a)$, they **estimate a density** over the values of the state-action pairs, which provides more information than only the mean value compared with DQN. This main discrepancy may be the reason why they achieve higher certified robustness than DQN.
>
> Regarding the benign performance in Appendix H.1, we note that the benign performance of a policy and its certified robustness against poisoning attacks are orthogonal. Even though a policy may perform well in the clean testing environment, it does not mean that it achieves high certified robustness which depends on several other properties such as the stability of the algorithm.
>
> Thus, we can see that it is important to provide such robustness certification, which can further serve as an effective metric to measure different policies in addition to their benign performance (as we shown in H.1 that different policies perform similarly).
>
> > 7. Choice of the trajectory length
>
> We thank the reviewer for the question. The trajectory length represents a trade-off between the experiment efficiency and the demonstrated results. With longer trajectory, we may observe more different comparisons between different RL algorithms and certification methods, but the overall running time and experimental cost will accordingly increase. The configurations used in our paper ($H=400$ for Freeway and $H=75$ for Breakout) are chosen by experimentation. With these configurations, we can observe sufficient interesting comparisons with affordable experimental cost, as well as a good assessment of different policies regarding their certified robustness against poisoning attacks.

---

> ### Author Response · Authors · 2021-11-21
> **Official Response to Reviewer RQX2 (1/4)**
>
> We thank the reviewer for the insightful comments and suggestions, and we provide our responses below.
>
> > 1. Motivation for developing three separate aggregation protocols
>
> We thank the reviewer for the thoughtful question. The three protocols are developed progressively considering additional flexibility based on the previous one, and DPARL is the most flexible one and we also demonstrate that it achieves the highest certified robustness.
>
> First of all, PARL is a relatively direct extension of the deep partition aggregation method [1] in supervised learning which only performs aggregation on the sub-policy level. Then, TPARL takes into consideration the **sequential decision making** nature of RL and leverages the temporal continuity to increase the tolerable poisoning threshold by additionally aggregating on the temporal level. Furthermore, DPARL circumvents the problem of requiring prior knowledge regarding accurate specifications for the window size $W$ at each time step by using **dynamically computed window size** per time step. We emphasize that each protocol intends to address the shortcomings of the previous one. Although empirically TPARL and DPARL provide similar results in our evaluated games, the contributions of DPARL are non-trivial, and DPARL may provide better results for games where the variance of window size is much larger across time steps.
>
> We thank the reviewer for the suggestion, and we have adjusted our presentation to emphasize the goals and focus of each protocol, and made it clear that DPARL is the final and most flexible proposed protocol, while others provide important information and analysis to understand the overall certified robustness of RL against poisoning attacks.
>
> > 2. Effect of varying $u$ on the learned sub-policies.
>
> We thank the reviewer for bringing up the interesting point. The reviewer is right that the partition number $u$ represents a trade-off between the learned sub-policy quality and the upper bound for the certified tolerable poisoning threshold. That’s to say, if $u$ is too small, the upper bound of tolerable poisoning threshold ($u/2$) could be too small; if $u$ is too large, each sub-policy has too few training trajectories to learn a high-quality policy. Given larger datasets, which contain more trajectories and allow for poisoning of more trajectories proportionally, we can adopt a larger partition number $u$ and train more policies to obtain potentially higher tolerable poisoning thresholds. But for the available dataset with limited size, we empirically evaluated that 50 is an appropriate partition number that provides both reasonable performance per sub-policy and promising certification results. We find that with $u$>50, each policy would be trained with only roughly 2M data, which is hard to converge in practice.
>
> > 3. Variance in window size for DPARL
>
> We thank the reviewer for the insightful comment. We follow the suggestion and conduct additional experiments as shown in Table 6 in Appendix H.3 in our revision. We provide the mean and variance of the selected window sizes in DPARL. The results show that the *maximum window size* $W_{\rm max}=5$, while the *average selected window sizes* are all around half of the maximum value. Thus, the average certification time is expected to be much smaller compared with the worst-case time complexity.
>
> > 4. Certification time complexity
> ​​
>
> We thank the reviewer for the interesting question. We follow the suggestion and add the new results on the runtime in Table 7 in Appendix H.3 in our revision. Specifically, we compare the running time of DPARL ($W_{\rm max}=5$) applied on $u=30$ and $50$ subpolicies trained using *offline RL algorithm* DQN with the *normal testing* which tests the runtime of a single trained DQN policy without using our framework. As we have shown in the remark of Theorem 3 in Section 3.3, the time complexity of DPARL is $​​​​​​O\left(W_{\max }^{2}|\mathcal{A}|^{2} u+W_{\max }|\mathcal{A}|^{2} u \log u\right)$. In Table 7, we also see that ​​the runtime of DPARL scales roughly quasilinearly with the number of subpolicies $u$, and quadratically with the action set size $|\cal A|$, which is $3$ for Freeway and $4$ for Breakout.

---

### Author Response · Authors · 2021-11-23
**Paper Revision Summary**



We thank all the reviewers for their insightful questions and suggestions! We are glad that the reviewers found our paper well-written, studies an important problem, technically solid, novel, and contains comprehensive experimental evaluation.

The below is a summary of major paper updates:

1. **[Section 2 - Definition 1]** Add an explicit definition of $\bar{K}_t$, following Reviewer $\color{purple}\text{6rjC}$’s suggestions.
2. **[Section 3.2, Section 5, Appendix I]** Include more related work and add a separate section (Appendix I) to discuss our relation with relative work in detail, following Reviewer $\color{green}\text{RQX2}$, Reviewer $\color{blue}\text{xuEG}$, Reviewer $\color{purple}\text{6rjC}$, and Reviewer $\color{brown}\text{zvac}$’s suggestions.
3. **[Section 3.2, Appendix B]** Add a detailed illustration, analysis, and discussion about bottleneck states and non-bottleneck states following Reviewer $\color{green}\text{RQX2}$’s suggestions.
4. **[Section 3.3 - Theorem 1 and Definition 4]** Simplify the notations in the theorem and eliminate the ambiguity caused by the previous ‘$\arg\max$’, following Reviewer $\color{brown}\text{zvac}$’s suggestions.
5. **[Section 3.3, Appendix H.3]** Add a discussion and extra experiment results about actual computation overhead of per-state action certification following Reviewer $\color{green}\text{RQX2}$’s suggestions.
6. **[Section 3.4, Appendix E.3]** Revise the description for COPA-Search algorithm to improve its clarity, clearly state its difference with related work (Wu et al), and provide a theoretical computational complexity analysis of COPA-Search, following Reviewer $\color{green}\text{RQX2}$’s suggestions.
7. **[Section 4.1, Appendix H.2 - Figure 3 and Table 5]** Report benign reward of our partition-based trained policies and standard trained policy to show that overall the proposed method achieves similar performance with DRL algorithms without considering adversaries since we only perform aggregation without adding strong regularization or anything operation that hurts the training, following Reviewer $\color{blue}\text{xuEG}$’s suggestions.
8. **[Section 4.1, Appendix H.3]** Provide additional statistics for DPARL, such as the mean and variance of the selected window sizes and certification runtime, following Reviewer $\color{green}\text{RQX2}$’s suggestions.
9. **[Section 4.1, Appendix H.4]** Provide the ratio of maximum tolerable poisoning threshold w.r.t. the total number of trajectories to improve the comprehensiveness following Reviewer $\color{purple}\text{6rjC}$’s suggestions.

Please also let us know if there are other questions, and we really look forward to the discussion with the reviewers to further improve our paper. Thank you!!

---

### Decision · Program_Chairs · 2022-01-20

**Decision:**

Accept (Poster)

**Comment:**

The authors develop a novel framework for certifying the robustness of RL agents against data poisoning attacks. They obtain lower bounds on the cumulative reward for several benchmark tasks.

Reviewers had concerns about certain organizational and technical aspects of the paper, but these were addressed well in the discussion phase and author responses. Hence, I recommend acceptance. However, I would urge the authors to incorporate points from the discussion phase into the revised version, in particular the discussion with reviewers xuEG and RQX2.